# CROP: Certifying Robust Policies for Reinforcement Learning through Functional Smoothing

**Fan Wu**[1] **Linyi Li**[1] **Zijian Huang**[1] **Yevgeniy Vorobeychik**[2] **Ding Zhao**[3] **Bo Li**[1]
[1]University of Illinois at Urbana-Champaign  [2]Washington University in St. Louis
[3]Carnegie Mellon University
{fanw6,linyi2,zijianh4,lbo}@illinois.edu  yvorobeychik@wustl.edu
dingzhao@andrew.cmu.edu

## Abstract

As reinforcement learning (RL) has achieved great success and been even adopted in safety-critical domains such as autonomous vehicles, a range of empirical studies have been conducted to improve its robustness against adversarial attacks. However, how to certify its robustness with theoretical guarantees still remains challenging. In this paper, we present the first unified framework CROP (Certifying Robust Policies for RL) to provide robustness certification on both action and reward levels. In particular, we propose two robustness certification criteria: *robustness of per-state actions* and *lower bound of cumulative rewards*. We then develop a local smoothing algorithm for policies derived from Q-functions to guarantee the robustness of actions taken along the trajectory; we also develop a *global smoothing* algorithm for certifying the lower bound of a finite-horizon cumulative reward, as well as a novel *local smoothing* algorithm to perform adaptive search in order to obtain tighter reward certification. Empirically, we apply CROP to evaluate several existing empirically robust RL algorithms, including adversarial training and different robust regularization, in four environments (two representative Atari games, Highway, and CartPole). Furthermore, by evaluating these algorithms against adversarial attacks, we demonstrate that our certifications are often tight. All experiment results are available at website `https://crop-leaderboard.github.io`.

## 1 Introduction

Reinforcement learning (RL) has been widely applied to different applications, such as robotics (Kober et al., 2013; Deisenroth et al., 2013; Polydoros & Nalpantidis, 2017), autonomous driving vehicles (Shalev-Shwartz et al., 2016; Sallab et al., 2017), and trading (Deng et al., 2016; Almahdi & Yang, 2017; Ye et al., 2020). However, recent studies have shown that learning algorithms are vulnerable to adversarial attacks (Goodfellow et al., 2014; Kurakin et al., 2016; Moosavi-Dezfooli et al., 2016; Jia & Liang, 2017; Eykholt et al., 2018), and a range of attacks have also been proposed against the input states and trained policies of RL (Huang et al., 2017; Kos & Song, 2017; Lin et al., 2017; Behzdan & Munir, 2017a). As more and more safety-critical applications are being deployed in real-world (Christiano et al., 2016; Fisac et al., 2018; Cheng et al., 2019; eop), how to test and improve their robustness before massive production is of great importance.

To defend against adversarial attacks in RL, different *empirical defenses* have been proposed (Mandlekar et al., 2017; Behzadan & Munir, 2017b; Pattanaik et al., 2018; Fischer et al., 2019; Zhang et al., 2020; Oikarinen et al., 2020; Donti et al., 2020; Shen et al., 2020; Eysenbach & Levine, 2021). In particular, adversarial training (Kos & Song, 2017; Behzadan & Munir, 2017b; Pattanaik et al., 2018) and regularized RL algorithms by enforcing the smoothness of the trained models (Shen et al., 2020; Zhang et al., 2020) have been studied to improve the robustness of trained policies. However, several strong adaptive attacks have been proposed against these empirical defenses (Gleave et al., 2019; Hussenot et al., 2019; Russo & Proutiere, 2019) and it is important to provide *robustness certification* for a given learning algorithm to end such repeated game between attackers and defenders.

To provide *robustness certification*, several studies have been conducted on classification. For instance, both deterministic (Ehlers, 2017; Katz et al., 2017; Cheng et al., 2017; Tjeng et al., 2017; Weng et al., 2018; Zhang et al., 2018; Singh et al., 2019; Gehr et al., 2018; Wong & Kolter, 2018; Raghunathan et al., 2018) and probabilistic approaches (Lecuyer et al., 2019; Cohen et al., 2019; Lee et al., 2019; Salman et al., 2019; Carmon et al., 2019; Jeong & Shin, 2020) have been explored to provide a lower bound of classification accuracy given bounded adversarial perturbation. Considering the sequential decision making property of RL, which makes it more challenging to be directly certified compared to classification, in this paper we ask: *How to provide efficient and effective robustness certification for RL algorithms? What criteria should be used to certify the robustness of RL algorithms?*

Different from classification which involves one-step prediction only, RL algorithms provide both action prediction and reward feedback, making *what to certify* and *how to certify* robustness of RL challenging. In this paper we focus on Q-learning and propose two certification criteria: *per-state action stability* and *lower bound of perturbed cumulative reward*. In particular, to certify the *per-state action stability*, we propose the *local smoothing* on each input state and therefore derive the certified radius for perturbation at each state, within which the action prediction will not be altered. To certify the *lower bound of cumulative reward*, we propose both *global smoothing* over the finite trajectory to obtain the expectation or percentile bounds given trajectories smoothed with sampled noise sequences; and *local smoothing* to calculate an absolute lower bound based on our adaptive search algorithm.

We leverage our framework to test nine empirically robust RL algorithms on multiple RL environments. We show that the certified robustness depends on both the algorithm and the environment properties. For instance, RadialRL (Oikarinen et al., 2020) is the most certifiably robust method on Freeway. In addition, based on the per-state certification, we observe that for some environments such as Pong, some states are more certifiably robust and such pattern is periodic. Given the information of which states are more vulnerable, it is possible to design robust algorithms to specifically focus on these vulnerable states. Based on the lower bound of perturbed cumulative reward, we show that our certification is tight by comparing our bounds with empirical results under adversarial attacks.

**Technical Contributions.** In this paper, we take an important step towards providing robustness certification for Q-learning. We make contributions on both theoretical and empirical fronts.

- We propose a framework for certifying the robustness of Q-learning algorithms, which is notably the *first* that provides the robustness certification w.r.t. the cumulative reward.
- We propose two robustness certification criteria for Q-learning algorithms, together with corresponding certification algorithms based on global and local smoothing strategies.
- We theoretically prove the certification radius for input state and lower bound of perturbed cumulative reward under bounded adversarial state perturbations.
- We conduct extensive experiments to provide certification for nine empirically robust RL algorithms on multiple RL environments. We provide several interesting observations which would further inspire the development of robust RL algorithms.

## 2 PRELIMINARIES

**Q-learning and Deep Q-Networks (DQNs).** Markov decision processes (MDPs) are at the core of RL. Our focus is on discounted discrete-time MDPs, which are defined by tuple $(\mathcal{S}, \mathcal{A}, R, P, \gamma, d_0)$, where $\mathcal{S}$ is a set of states (each with dimensionality $N$), $\mathcal{A}$ represents a set of discrete actions, $R : \mathcal{S} \times \mathcal{A} \to \mathbb{R}$ is the reward function, and $P : \mathcal{S} \times \mathcal{A} \to \mathcal{P}(\mathcal{S})$ is the transition function with $\mathcal{P}(\cdot)$ defining the set of probability measures, $\gamma \in [0, 1]$ is the discount factor, and $d_0 \in \mathcal{P}(\mathcal{S})$ is the distribution over the initial state. At time step $t$, the agent is in the state $s_t \in \mathcal{S}$. After choosing action $a_t \in \mathcal{A}$, the agent transitions to the next state $s_{t+1} \sim P(s_t, a_t)$ and receives reward $R(s_t, a_t)$. The goal is to learn a policy $\pi : \mathcal{S} \to \mathcal{P}(\mathcal{A})$ that maximizes the expected cumulative reward $\mathbb{E}[\sum_t \gamma^t r_t]$.

Q-learning (Watkins & Dayan, 1992) learns an action-value function (Q-function), $Q^\star(s, a)$, which is the maximum expected cumulative reward the agent can achieve after taking action $a$ in state $s$: $Q^\star(s, a) = R(s, a) + \gamma \mathbb{E}_{s' \sim P(s,a)} [\max_{a'} Q^\star(s', a')]$. In deep Q-Networks (DQNs) (Mnih et al., 2013), $Q^\star$ is approximated using a neural network parametrized by $\theta$, *i.e.*, $Q^\pi(s, a; \theta) \approx Q^\star(s, a)$. Let $\rho \in \mathcal{P}(\mathcal{S} \times \mathcal{A})$ be the observed distribution defined over states $s$ and actions $a$, the network can be trained via minimizing loss function $\mathcal{L}(\theta) = \mathbb{E}_{(s,a) \sim \rho, s' \sim P(s,a)} \left[ (R(s, a) + \gamma \max_{a'} Q^\pi (s', a'; \theta) - Q^\pi(s, a; \theta))^2 \right]$. The greedy policy $\pi$ is defined as taking the action with highest $Q^\pi$ value in each state $s$: $\pi(s) = \arg\max_{a \in \mathcal{A}} Q^\pi(s, a)$.

**Certified Robustness for Classifiers via Randomized Smoothing.** Randomized smoothing (Cohen et al., 2019) has been proposed to provide probabilistic certified robustness for classification. It achieves state-of-the-art certified robustness on large-scale datasets such as ImageNet under $\ell_2$-bounded constraints (Salman et al., 2019; Yang et al., 2020). In particular, given a base model and a test instance, a smoothed model is constructed by outputting the most probable prediction over different Gaussian perturbed inputs.

## 3   ROBUSTNESS CERTIFICATION IN Q-LEARNING

In this section, we first introduce the threat model, followed by two robustness certification criteria for the Q-learning algorithm: *per-state action* and *cumulative reward*. We consider the standard adversarial setting in Q-learning (Huang et al., 2017; Kos & Song, 2017; Zhang et al., 2020), where the adversary can apply $\ell_2$-bounded perturbation $\mathcal{B}^\varepsilon = \{\delta \in \mathbb{R}^n \mid \|\delta\|_2 \leq \varepsilon\}$ to input state observations of the agent during decision (test) time to cause the policy to select suboptimal actions. The agent observes the perturbed state and takes action $a' = \pi(s + \delta)$, following policy $\pi$. Following the Kerckhoff's principle (Shannon, 1949), we consider a *worst-case* adversary who applies adversarial perturbations to *every* state at decision time. Our analysis and methods are generalizable to other $\ell_p$ norms following (Yang et al., 2020; Lecuyer et al., 2019).

### 3.1   ROBUSTNESS CERTIFICATION FOR Q-LEARNING WITH DIFFERENT CRITERIA

To provide the robustness certification for Q-learning, we propose two certification criteria: *per-state action robustness* and *lower bound of the cumulative reward*.

**Robustness Certification for Per-State Action.** We first aim to explore the robustness (stability/consistency) of the per-state action given adversarially perturbed input states.

**Definition 1** (Certification for per-state action). Given a trained network $Q^\pi$ with policy $\pi$, we define the robustness certification for per-state action as the *maximum perturbation magnitude* $\bar{\varepsilon}$, such that for any perturbation $\delta \in \mathcal{B}^{\bar{\varepsilon}}$, the predicted action under the perturbed state will be the same as the action taken in the clean environment, *i.e.*, $\pi(s + \delta) = \pi(s), \forall \delta \in \mathcal{B}^{\bar{\varepsilon}}$.

**Robustness Certification for Cumulative Reward.** Given that the cumulative reward is important for RL, here in addition to the per-state action, we also define the robustness certification regarding the cumulative reward under input state perturbation.

**Definition 2** (Cumulative reward). Let $P : \mathcal{S} \times \mathcal{A} \to \mathcal{P}(\mathcal{S})$ be the transition function of the environment with $\mathcal{P}(\cdot)$ defining the set of probability measures. Let $R, d_0, \gamma, Q^\pi, \pi$ be the reward function, initial state distribution, discount factor, a given trained Q-network, and the corresponding greedy policy as introduced in Section 2. $J(\pi)$ represents the *cumulative reward* and $J_\varepsilon(\pi)$ represents the *perturbed cumulative reward* under perturbations $\delta_t \in \mathcal{B}^\varepsilon$ at each time step $t$:

$$J(\pi) := \sum_{t=0}^{\infty} \gamma^t R(s_t, \pi(s_t)), \quad \text{and} \quad J_\varepsilon(\pi) := \sum_{t=0}^{\infty} \gamma^t R(s_t, \pi(s_t + \delta_t)), \tag{1}$$
$$\text{where } s_{t+1} \sim P(s_t, a_t), s_0 \sim d_0, \quad \text{where } s_{t+1} \sim P(s_t, \pi(s_t + \delta_t)), s_0 \sim d_0.$$

The randomness of $J(\pi)$ arises from the environment dynamics, while that of $J_\varepsilon(\pi)$ includes additional randomness from the perturbations $\{\delta_t\}$. We focus on a finite horizon $H$ in this paper, where a sufficiently large $H$ can approximate $J(\pi)$ and $J_\varepsilon(\pi)$ to arbitrary precision when $\gamma < 1$.

**Definition 3** (Robustness certification for cumulative reward). The robustness certification for cumulative reward is the *lower bound* of *perturbed cumulative reward* $\underline{J}$ such that $\underline{J} \leq J_\varepsilon(\pi)$ under perturbation in $\mathcal{B}^\varepsilon = \{\delta \in \mathbb{R}^n \mid \|\delta\|_2 \leq \varepsilon\}$ applied to all time steps.

We will provide details on the certification of per-state action in Section 4 and the certification of cumulative reward in Section 5 based on different smoothing strategies and certification methods.

## 4   ROBUSTNESS CERTIFICATION STRATEGIES FOR PER-STATE ACTION

In this section, we discuss the robustness certification for *per-state action*, aiming to calculate a lower bound of *maximum perturbation magnitude* $\bar{\varepsilon}$ in Definition 1.

### 4.1   CERTIFICATION FOR PER-STATE ACTION VIA ACTION-VALUE FUNCTIONAL SMOOTHING

Let $Q^\pi$ be the action-value function given by the trained network $Q$ with policy $\pi$. We derive a smoothed function $\widetilde{Q}^\pi$ through per-state *local smoothing*. Specifically, at each time step $t$, for each action $a \in \mathcal{A}$, we draw random noise from a Gaussian distribution $\mathcal{N}(0, \sigma^2 I_N)$ to smooth $Q^\pi(\cdot, a)$.

$$\widetilde{Q}^{\pi}(s_t, a) := \underset{\Delta_t \sim \mathcal{N}(0, \sigma^2 I_N)}{\mathbb{E}} Q^{\pi}(s_t + \Delta_t, a) \quad \forall s_t \in \mathcal{S}, a \in \mathcal{A}, \text{ and } \tilde{\pi}(s_t) := \underset{a}{\arg\max} \widetilde{Q}^{\pi}(s_t, a) \quad \forall s_t \in \mathcal{S}.$$

(2)

**Lemma 1** (Lipschitz continuity of the smoothed value function). *Given the action-value function $Q^{\pi} : \mathcal{S} \times \mathcal{A} \to [V_{\min}, V_{\max}]$, the smoothed function $\widetilde{Q}^{\pi}$ with smoothing parameter $\sigma$ is L-Lipschitz continuous with $L = \frac{V_{\max} - V_{\min}}{\sigma} \sqrt{2/\pi}$ w.r.t. the state input.*

The proof is given in Appendix A.1. Leveraging the Lipschitz continuity in Lemma 1, we derive the following theorem for certifying the robustness of per-state action.

**Theorem 1.** *Let $Q^{\pi} : \mathcal{S} \times \mathcal{A} \to [V_{\min}, V_{\max}]$ be a trained value network, $\widetilde{Q}^{\pi}$ be the smoothed function with (2). At time step $t$ with state $s_t$, we can compute the lower bound $r_t$ of maximum perturbation magnitude $\bar{\varepsilon}(s_t)$ (i.e., $r_t \leq \bar{\varepsilon}(s_t)$, $\bar{\varepsilon}$ defined in Definition 1) for locally smoothed policy $\tilde{\pi}$:*

$$r_t = \frac{\sigma}{2} \left( \Phi^{-1} \left( \frac{\widetilde{Q}^{\pi}(s_t, a_1) - V_{\min}}{V_{\max} - V_{\min}} \right) - \Phi^{-1} \left( \frac{\widetilde{Q}^{\pi}(s_t, a_2) - V_{\min}}{V_{\max} - V_{\min}} \right) \right),$$

(3)

*where $\Phi^{-1}$ is the inverse CDF function, $a_1$ is the action with the highest $\widetilde{Q}^{\pi}$ value at state $s_t$, and $a_2$ is the runner-up action. We name the lower bound $r_t$ as certified radius for the state $s_t$.*

The proof is omitted to Appendix A.2. The theorem provides a certified radius $r_t$ for per-state action given smoothed policy: As long as the perturbation is bounded by $r_t$, *i.e.*, $\|\delta_t\|_2 \leq r_t$, the action does not change: $\tilde{\pi}(s_t + \delta_t) = \tilde{\pi}(s_t)$. To achieve high certified robustness for per-state action, Theorem 1 implies a tradeoff between value function smoothness and the margin between the values of top two actions: If a larger smoothing parameter $\sigma$ is applied, the action-value function would be smoother and therefore more stable; however, it would shrink the margin between the top two action values leading to smaller certified radius. Thus, there exists a proper smoothing parameter to balance the tradeoff, which depends on the actual environments and algorithms.

## 4.2 CROP-LOACT: LOCAL RANDOMIZED SMOOTHING FOR CERTIFYING PER-STATE ACTION

Next we introduce the algorithm to achieve the certification for per-state action. Given a $Q^{\pi}$ network, we apply (2) to derive a smoothed network $\widetilde{Q}^{\pi}$. At each state $s_t$, we obtain the greedy action $\tilde{a}_t$ w.r.t. $\widetilde{Q}^{\pi}$, and then compute the certified radius $r_t$. We present the complete algorithm in Appendix B.1.

There are some additional challenges in smoothing the value function and computing the certified radius in Q-learning compared with the standard classification task (Cohen et al., 2019). **Challenge 1:** In classification, the output range of the confidence $[0, 1]$ is known a priori; however, in Q-learning, for a given $Q^{\pi}$, its range $[V_{\min}, V_{\max}]$ is unknown. **Challenge 2:** In the classification task, the lower and upper bounds of the top two classes' prediction probabilities can be directly computed via the confidence interval base on multinomial proportions (Goodman, 1965). For Q-networks, the outputs are not probabilities and calculating the multinomial proportions becomes challenging.

**Pre-processing.** To address **Challenge 1**, we estimate the output range $[V_{\min}, V_{\max}]$ of a given network $Q^{\pi}$ based on a finite set of valid states $\mathcal{S}_{\text{sub}} \subseteq \mathcal{S}$. In particular, we craft a sufficiently large set $\mathcal{S}_{\text{sub}}$ to estimate $V_{\min}$ and $V_{\max}$ for $Q^{\pi}$ on $\mathcal{S}$, which can be used later in per-state smoothing. The details for estimating $V_{\min}$ and $V_{\max}$ are deferred to Appendix B.1.

**Certification.** To smooth a given state $s_t$, we use Monte Carlo sampling (Cohen et al., 2019) to sample noise applied to $s_t$, and then estimate the corresponding smoothed value function $\widetilde{Q}^{\pi}$ at $s_t$ with (2). In particular, we sample $m$ Gaussian noise $\Delta_i \sim \mathcal{N}(0, \sigma^2 I_N)$, clip the Q-network output to ensure that it falls within the range $[V_{\min}, V_{\max}]$, and then take the average of the output to obtain the smoothed action prediction based on $\widetilde{Q}^{\pi}$. We then employ Theorem 1 to compute the certified radius $r_t$. We omit the detailed inference procedure to Appendix B.1. To address **Challenge 2**, we leverage Hoeffding's inequality (Hoeffding, 1994) to compute a lower bound of $\widetilde{Q}^{\pi}(s_t, a_1)$ and an upper bound of $\widetilde{Q}^{\pi}(s_t, a_2)$ with one-sided confidence level parameter $\alpha$ given the top two actions $a_1$ and $a_2$. When the former is higher than the latter, we can certify a positive radius for the given state $s_t$.

## 5 ROBUSTNESS CERTIFICATION STRATEGIES FOR THE CUMULATIVE REWARD

In this section, we present robustness certification strategies for the cumulative reward. The goal is to provide the lower bounds for the *perturbed cumulative reward* in Definition 2. In particular, we propose both *global smoothing* and *local smoothing* strategies to certify the perturbed cumulative reward. In the global smoothing, we view the whole state trajectory as a function to smooth, which

would lead to relatively loose certification bound. We then propose the local smoothing by smoothing each state individually to obtain the absolute lower bound.

## 5.1 CERTIFICATION OF CUMULATIVE REWARD BASED ON GLOBAL SMOOTHING

In contrast to Section 4 where we perform per-state smoothing to achieve the certification for per-state action, here, we aim to perform *global smoothing* on the state trajectory by viewing the entire trajectory as a function. In particular, we first derive the *expectation bound* of the cumulative reward based on global smoothing by estimating the Lipschitz constant for the cumulative reward w.r.t. the trajectories. Since the Lipschitz estimation in the expectation bound is algorithm agnostic and could lead to loose estimation bound, we subsequently propose a more practical and tighter *percentile bound*.

**Definition 4** ($\sigma$-randomized trajectory and $\sigma$-randomized policy). Given a state trajectory $(s_0, s_1, \ldots, s_{H-1})$ of length $H$ where $s_{t+1} \sim P(s_t, \pi(s_t))$, $s_0 \sim d_0$, with $\pi$ the greedy policy of the action-value function $Q^\pi$, we derive a $\sigma$-*randomized trajectory* as $(s'_0, s'_1, \ldots, s'_{H-1})$, where $s'_{t+1} \sim P(s'_t, \pi(s'_t + \Delta_t))$, $\Delta_t \sim \mathcal{N}(0, \sigma^2 I_N)$, and $s'_0 = s_0 \sim d_0$. We correspondingly define a $\sigma$-*randomized policy* $\pi'$ based on $\pi$ in the following form: $\pi'(s_t) := \pi(s_t + \Delta_t)$ where $\Delta_t \sim \mathcal{N}(0, \sigma^2 I_N)$.

Let the operator $\oplus$ concatenates given input states or noise that are added to each state. The sampled noise sequence is denoted by $\Delta = \oplus_{t=0}^{H-1} \Delta_t$, where $\Delta_t \sim \mathcal{N}(0, \sigma^2 I_N)$.

**Definition 5** (Perturbed return function). Let $R, P, \gamma, d_0$ be the reward function, transition function, discount factor, and initial state distribution in Definition 2. We define a bounded *perturbed return function* $F_\pi : \mathbb{R}^{H \times N} \to [J_{\min}, J_{\max}]$ representing cumulative reward with potential perturbation $\delta$:

$$F_\pi\left(\oplus_{t=0}^{H-1} \delta_t\right) := \sum_{t=0}^{H} \gamma^t R(s_t, \pi(s_t + \delta_t)), \quad \text{where } s_{t+1} \sim P(s_t, \pi(s_t + \delta_t)), s_0 \sim d_0. \quad (4)$$

We can see when there is no perturbation ($\delta_t = \mathbf{0}$), $F_\pi(\oplus_{t=0}^{H-1} \mathbf{0}) = J(\pi)$; when there are adversarial perturbations $\delta_t \in \mathcal{B}^\varepsilon$ at each time step, $F_\pi(\oplus_{t=0}^{H-1} \delta_t) = J_\varepsilon(\pi)$, *i.e.*, *perturbed cumulative reward*.

**Mean Smoothing: Expectation bound.** Here we propose to sample noise sequences $\Delta$ to perform *global smoothing* for the entire state trajectory, and calculate the lower bound of the expected perturbed cumulative reward $\mathbb{E}_\Delta [J_\varepsilon(\pi')]$ under all possible $\ell_2$-bounded perturbations within magnitude $\varepsilon$. The expectation is over the noise sequence $\Delta$ involved in the $\sigma$-randomized policy $\pi'$ in Definition 4.

**Lemma 2** (Lipschitz continuity of smoothed perturbed return function). *Let $F$ be the perturbed return function function defined in (4), the smoothed perturbed return function $\widetilde{F}_\pi$ is $\frac{(J_{\max} - J_{\min})}{\sigma} \sqrt{2/\pi}$-Lipschitz continuous, where $\widetilde{F}_\pi\left(\oplus_{t=0}^{H-1} \delta_t\right) := \mathbb{E}_{\Delta \sim \mathcal{N}(0, \sigma^2 I_{H \times N})} F_\pi\left(\oplus_{t=0}^{H-1}(\delta_t + \Delta_t)\right)$.*

**Theorem 2** (Expectation bound). *Let $\underline{J_E} = \widetilde{F}_\pi\left(\oplus_{t=0}^{H-1} \mathbf{0}\right) - L\varepsilon\sqrt{H}$, where $L = \frac{(J_{\max} - J_{\min})}{\sigma} \sqrt{2/\pi}$. Then $\underline{J_E} \leq \mathbb{E}[J_\varepsilon(\pi')]$.*

*Proof Sketch.* We first derive the equality between expected perturbed cumulative reward $\mathbb{E}[J_\varepsilon(\pi')]$ and the smoothed perturbed return function $\widetilde{F}_\pi(\oplus_{t=0}^{H-1} \delta_t)$. Thus, to lower bound the former, it suffices to lower bound the latter, which can be calculated leveraging the Lipschitz continuity of $\widetilde{F}$ in Lemma 2 (proved in Appendix A.3), noticing that the distance between $\oplus_{t=0}^{H-1} \mathbf{0}$ and the adversarial perturbations $\oplus_{t=0}^{H-1} \delta_t$ is bounded by $\varepsilon\sqrt{H}$. The complete proof is omitted to Appendix A.4.

We obtain $J_{\min}$ and $J_{\max}$ in Lemma 2 from environment specifications which can be loose in practice. Thus the Lipschitz constant $L$ estimation is coarse and mean smoothing is usually loose. We next present a method that circumvents estimating the Lipschitz constant and provides a tight percentile bound.

**Percentile Smoothing: Percentile bound.** We now propose to apply *percentile smoothing* to smooth the *perturbed cumulative reward* and obtain the lower bound of the $p$-th percentile of $J_\varepsilon(\pi')$, where $\pi'$ is a $\sigma$-randomized policy defined in Definition 4.

$$\widetilde{F}_\pi^p\left(\oplus_{t=0}^{H-1} \delta_t\right) = \sup_y \left\{y \in \mathbb{R} \mid \mathbb{P}\left[F_\pi\left(\oplus_{t=0}^{H-1}(\delta_t + \Delta_t)\right) \leq y\right] \leq p\right\}. \quad (5)$$

**Theorem 3** (Percentile bound). *Let $\underline{J_p} = \widetilde{F}_\pi^{p'}\left(\oplus_{t=0}^{H-1} \mathbf{0}\right)$, where $p' := \Phi\left(\Phi^{-1}(p) - \varepsilon\sqrt{H}/\sigma\right)$. Then $\underline{J_p} \leq$ the $p$-th percentile of $J_\varepsilon(\pi')$.*

The proof is provided in Appendix A.5 based on Chiang et al. (2020). There are several other advantages of percentile smoothing over mean smoothing. First, the certification given by percentile

smoothing is among the cumulative rewards of the sampled $\sigma$-randomized trajectories and is therefore achievable by a real-world policy, while the expectation bound is less likely to be achieved in practice given the loose Lipschitz bound. Second, for a discrete function such as perturbed return function, the output of mean smoothing is continuous w.r.t. $\sigma$, while the results given by percentile smoothing remain discrete. Thus, the percentile smoothed function preserves properties of the base function before smoothing, and shares similar interpretation, *e.g.*, the number of rounds that the agent wins. Third, taking $p = 50\%$ in percentile smoothing leads to the *median smoothing* which achieves additional properties such as robustness to outliers (Manikandan, 2011). The detailed algorithm CROP-GRE, including the inference procedure, the estimation of $\widetilde{F}_\pi$, the calculation of the empirical order statistics, and the configuration of algorithm parameters $J_{\min}, J_{\max}$, are deferred to Appendix B.2.

## 5.2 Certification of Cumulative Reward based on Local Smoothing

Though global smoothing provides efficient and practical bounds for the perturbed cumulative reward, such bounds are still loose as they involve smoothing the entire trajectory at once. In this section, we aim to provide a tighter lower bound for $J_\varepsilon(\tilde{\pi})$ by performing *local smoothing*.

Given a trajectory of $H$ time steps which is guided by the locally smoothed policy $\tilde{\pi}$, we can compute the certified radius at each time step according to Theorem 1, which can be denoted as $r_0, r_1, \ldots, r_{H-1}$. Recall that when the perturbation magnitude $\varepsilon < r_t$, the optimal action $a_t$ at time step $t$ will remain unchanged. This implies that when $\varepsilon < \min_{t=0}^{H-1} r_t$, none of the actions in the entire trajectory will be changed, and therefore the lower bound of the cumulative reward when $\varepsilon < \min_{t=0}^{H-1} r_t$ is the return of the current trajectory in a deterministic environment. Increasing $\varepsilon$ has two effects. First, the *total* number of time steps where the action is susceptible to change will increase; second, at *each* time step, the action can change from the best to the runner-up or the rest. We next introduce an extension of certified radius $r_t$ to characterize the two effects.

**Theorem 4.** *Let $(r_t^1, \ldots, r_t^{|\mathcal{A}|-1})$ be a sequence of certified radii for state $s_t$ at time step $t$, where $r_t^k$ denotes the radius such that if $\varepsilon < r_t^k$, the possible action at time step $t$ will belong to the actions corresponding to top $k$ action values of $\widetilde{Q}$ at state $s_t$. The definition of $r_t$ in Theorem 1 is equivalent to $r_t^1$ here. The radii can be computed similarly as follows:*

$$r_t^k = \frac{\sigma}{2} \left( \Phi^{-1} \left( \frac{\widetilde{Q}^\pi(s_t, a_1) - V_{\min}}{V_{\max} - V_{\min}} \right) - \Phi^{-1} \left( \frac{\widetilde{Q}^\pi(s_t, a_{k+1}) - V_{\min}}{V_{\max} - V_{\min}} \right) \right), \quad 1 \leq k < |\mathcal{A}|,$$

*where $a_1$ is the action of the highest $\widetilde{Q}$ value at state $s_t$ and $a_{k+1}$ is the $(k+1)$-th best action. We additionally define $r_t^0(s_t) = 0$, which is also compatible with the definition above.*

We defer the proof in Appendix A.6. With Theorem 4, for any given $\varepsilon$, we can compute all possible actions under perturbations in $\mathcal{B}^\varepsilon$. This allows an exhaustive search to traverse all trajectories satisfying that all certified radii along the trajectory are smaller than $\varepsilon$. Then, we can conclude that $J_\varepsilon(\tilde{\pi})$ is lower bounded by the minimum return over all these possible trajectories.

### 5.2.1 CROP-LoRe: Local smoothing for Certified Reward

Given a policy $\pi$ in a deterministic environment, let the initial state be $s_0$, we propose CROP-LoRe to certify the lower bound of $J_\varepsilon(\tilde{\pi})$. At a high-level, CROP-LoRe *exhaustively* explores new trajectories leveraging Theorem 4 with priority queue and *effectively* updates the lower bound of cumulative reward $\underline{J}$ by expanding a trajectory tree dynamically. The algorithm returns a collection of pairs $\{(\varepsilon_i, \underline{J}_{\varepsilon_i})\}_{i=1}^{|C|}$ sorted in ascending order of $\varepsilon_i$, where $|C|$ is the length of the collection. For all $\varepsilon'$, let $i$ be the largest integer such that $\varepsilon_i \leq \varepsilon' < \varepsilon_{i+1}$, then as long as the perturbation magnitude $\varepsilon \leq \varepsilon'$, the cumulative reward $J_\varepsilon(\tilde{\pi}) \geq \underline{J}_{\varepsilon_i}$. The algorithm is shown in Algorithm 3 in Appendix B.3.

**Algorithm Description.** The method starts from the base case: when perturbation magnitude $\varepsilon = 0$, the lower bound of cumulative reward $\underline{J}$ is exactly the benign reward. The method then gradually increases the perturbation magnitude $\varepsilon$ (later we will explain how a new $\varepsilon$ is determined). Along the increase of $\varepsilon$, the perturbation may cause the policy $\pi$ to take different actions at some time steps, thus resulting in new trajectories. Thanks to the local smoothing, the method leverages Theorem 4 to figure out the exhaustive list of possible actions under current perturbation magnitude $\varepsilon$, and effectively explore these new trajectories by formulating them as expanded branches of a trajectory tree. Once all new trajectories are explored, the method examines all leaf nodes of the tree and figures out the minimum reward among them, which is the new lower bound of cumulative reward $\underline{J}$ under this new $\varepsilon$. To mitigate the explosion of branches, CROP-LoRe proposes several *optimization* tricks.

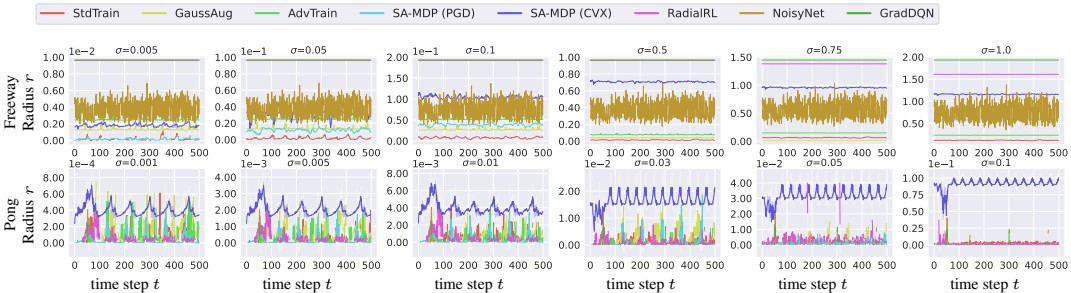

Figure 1: Robustness certification for *per-state action* in terms of certified radius $r$ at all time steps. Each column corresponds to a smoothing variance $\sigma$. The shaded area represents the standard deviation which is small. RadialRL is the most certifiably robust method on Freeway, while SA-MDP (CVX) is the most robust on Pong.

In the following, we will briefly introduce the *trajectory exploration and expansion* and *growth of perturbation magnitude* steps, as well as the *optimization* tricks. More algorithm details are deferred to Appendix B.3, where we also provide an analysis on the time complexity of the algorithm.

In **trajectory exploration and expansion**, CROP-LoRe organizes all possible trajectories in the form of search tree and progressively grows it. For each node (representing a state), leveraging Theorem 4, we compute a non-decreasing sequence $\{r^k(s)\}_{k=0}^{|\mathcal{A}|-1}$ representing the required perturbation radii for $\pi$ to choose each alternative action. Suppose the current $\varepsilon$ satisfies $r^i(s) \leq \varepsilon < r^{i+1}(s)$, we can grow $(i+1)$ branches from current state $s$ corresponding to the original action and $i$ alternative actions since $\varepsilon \geq r^j(s)$ for $1 \leq j \leq i$. We expand the tree branches using depth-first search (Tarjan, 1972). In **perturbation magnitude growth**, when all trajectories for perturbation magnitude $\varepsilon$ are explored, we increase $\varepsilon$ to seek certification under larger perturbations. This is achieved by preserving a priority queue (van Emde Boas, 1977) of the critical $\varepsilon$'s that we will expand on. Concretely, along the trajectory, at each tree node, we search for the possible actions and store actions corresponding to $\{r^k(s)\}_{k=i+1}^{|\mathcal{A}|-1}$ into the priority queue, since these actions are exactly those need to be explored when $\varepsilon$ grows. We repeat the procedure of *perturbation magnitude growth* (*i.e.*, popping out the head element from the queue) and *trajectory exploration and expansion* (*i.e.*, exhaustively expand all trajectories given the perturbation magnitude) until the priority queue becomes empty or the perturbation magnitude $\varepsilon$ reaches the predefined threshold. Additionally, we adopt a few **optimization** tricks commonly used in search algorithms to reduce the complexity of the algorithm, such as pruning and the memorization technique (Michie, 1968). More potential improvements are discussed in Appendix B.3.

So far, we have presented all our certification methods. In Appendix C, we further discuss the *advantages*, *limitations*, and *extensions* of these methods, and provide more *detailed analysis* to help with understanding.

## 6 EXPERIMENTS

In this section, we present evaluation for the proposed robustness certification framework CROP. Concretely, we apply our three certification algorithms (CROP-LoAct, CROP-GRe, and CROP-LoRe) to certify nine RL methods (**StdTrain** (Mnih et al., 2013), **GaussAug** (Kos & Song, 2017), **Adv-Train** (Behzadan & Munir, 2017b), **SA-MDP (PGD,CVX)** (Zhang et al., 2020), **RadialRL** (Oikarinen et al., 2020), **CARRL** (Everett et al., 2021), **NoisyNet** (Fortunato et al., 2017), and **Grad-DQN** (Pattanaik et al., 2018)) on two high-dimensional Atari games (Pong and Freeway), one low dimensional control environment (CartPole), and an autonomous driving environment (Highway). As a summary, we find that (1) SA-MDP (CVX), SA-MDP (PGD), and RadialRL achieve high certified robustness in different environments; (2) Large smoothing variance can help to improve certified robustness significantly on Freeway, while a more careful selection of the smoothing parameter is needed in Pong; (3) For methods that demonstrate high certified robustness, our certification of the cumulative reward is tight. We defer the detailed descriptions of the environments to Appendix D.1 and the introduction to the RL methods and implementation details to Appendix D.2. More interesting results and discussions are omitted to Appendix E and our leaderboard, including results on CartPole (Appendix E.6) and Highway (Appendix E.7). As a *foundational* work providing robustness certification for RL, we expect more RL algorithms and RL environments will be certified under our framework in future work.

### 6.1 EVALUATION OF ROBUSTNESS CERTIFICATION FOR PER-STATE ACTION

In this subsection, we provide the robustness certification evaluation for per-state action.

**Experimental Setup and Metrics.** We evaluate the *locally smoothed policy* $\tilde{\pi}$ derived in (2). We follow Theorem 1 and report the *certified radius* $r_t$ at each step $t$.

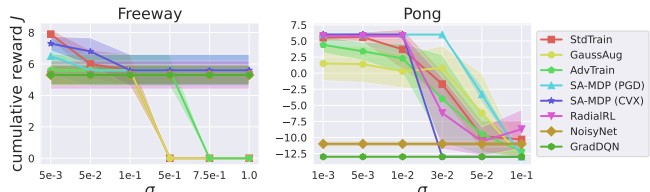

Figure 2: Benign performance of locally smoothed policy $\tilde{\pi}$ under different smoothing variance $\sigma$ with clean state observations.

We additionally compute *certified ratio $\eta$ at radius $r$*, representing the percentage of time steps that can be certified, formally denoted as $\eta_r = 1/H \sum_{t=1}^{H} \mathbb{1}_{\{r_t \geq r\}}$. More details are omitted to Appendix D.4.

**Evaluation Results of CROP-LOACT.** We present the certification comparison in Figure 1 and the benign performance under different smoothing variances in Figure 2; we omit details of certified ratio to Appendix E.1.

On *Freeway*, we evaluate with a large range of smoothing parameter $\sigma$ up to 1.0, since Freeway can tolerate large noise as shown in Figure 2. Results under even larger $\sigma$ are deferred to Appendix E.2, showing that the certified robustness can be further improved for the empirically robust methods. From Figure 1, we see that *RadialRL consistently achieves the highest certified radius across all $\sigma$'s.* This is because RadialRL explicitly optimizes over the worst-case perturbations. StdTrain, GaussAug, and AdvTrain are not as robust, and increasing $\sigma$ will not make a difference. In *Pong*, SA-MDP (CVX) is the most certifiably robust, which may be due to the hardness of optimizing the worst-case perturbation in Pong. More interesting, all methods present similar periodic patterns for the certified radius on different states, which would inspire further robust training methods to take the "confident state" (*e.g.*, when the ball is flying towards the paddle) into account. We illustrate the frames with varying certified radius in Appendix E.3. Overall, the certified robustness of these methods largely matches empirical observations (Behzadan & Munir, 2017b; Oikarinen et al., 2020; Zhang et al., 2020).

## 6.2 EVALUATION OF ROBUSTNESS CERTIFICATION FOR CUMULATIVE REWARD

Here we will discuss the evaluation for the robustness certification regarding cumulative reward in Section 5. We show the evaluation results for both CROP-GRE and CROP-LORE.

**Evaluation Setup and Metrics.** We evaluate the *$\sigma$-randomized policy $\pi'$* derived in Definition 4 for CROP-GRE and the *locally smoothed policy $\tilde{\pi}$* derived in (2) for CROP-LORE. We compute the expectation bound $\underline{J_E}$, percentile bound $\underline{J_p}$, and absolute lower bound $\underline{J}$ following Theorem 2, Theorem 3, and Section 5.2.1. To validate the tightness of the bounds, we additionally perform empirical attacks. More rationales see Appendix D.3. The detailed parameters are omitted to Appendix D.4.

**Evaluation of CROP-GRE.** We present the certification of reward in Figure 3 and mainly focus on analyzing percentile bound. We present the bound w.r.t. different attack magnitude $\varepsilon$ under different smoothing parameter $\sigma$. For each $\sigma$, there exists an upper bound of $\varepsilon$ that can be certified, which is positively correlated with $\sigma$. Details see Appendix B.2. We observe similar conclusion as in the per-state action certification that RadialRL is most certifiably robust for Freeway, while SA-MDP (CVX,PGD) are most robust for Pong although the highest robustness is achieved at different $\sigma$.

**Evaluation of CROP-LORE.** In Figure 3, we note that in *Freeway*, RadialRL is the most robust, followed by SA-MDP (CVX), then SA-MDP (PGD), AdvTrain, and GaussAug. As for *Pong*, SA-MDP (CVX) outperforms RadialRL and SA-MDP (PGD). Remaining methods are not clearly ranked.

## 6.3 DISCUSSION ON EVALUATION RESULTS

**Impact of Smoothing Parameter $\sigma$.** We draw similar conclusions regarding the impact of smoothing variance from the results given by CROP-LOACT and CROP-GRE. In *Freeway*, as $\sigma$ increases, the robustness of StdTrain, GaussAug, and AdvTrain barely increases, while that of SA-MDP (PGD), SA-MDP (CVX), and RadialRL steadily increases. In *Pong*, a $\sigma$ in the range 0.01-0.03 shall be suitable for almost all methods. More explanations from the perspective of benign performance and certified results will be provided in Appendix E.4. For CROP-LORE, there are a few different conclusions. On *Freeway*, it is clear that under larger attack magnitude $\varepsilon$, larger smoothing parameter $\sigma$ always secures higher lower bound $\underline{J}$. In *Pong*, CROP-LORE can only achieve non-zero certification with small $\sigma$ under a small range of attack magnitude, implying the difficulty of establishing non-trivial certification for these methods on Pong. We defer details on the selection of $\sigma$ to Appendix E.4.

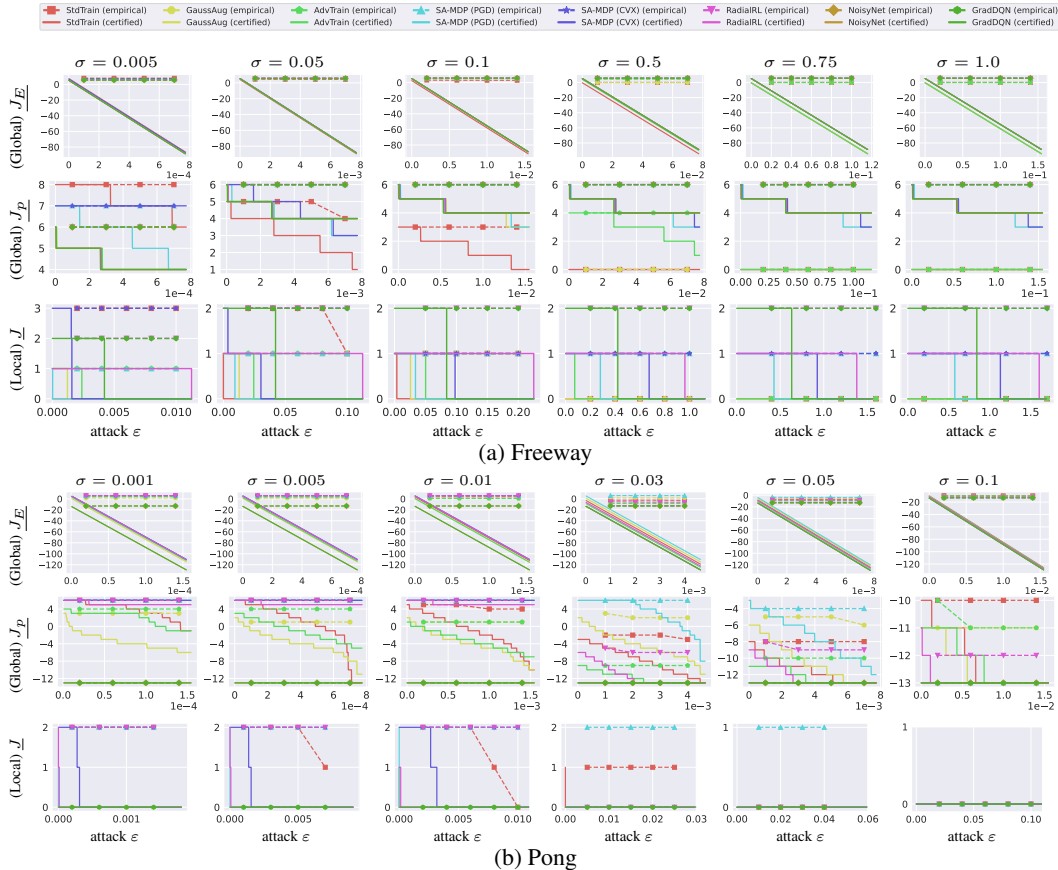

Figure 3: Robustness certification for cumulative reward, including *expectation bound* $\underline{J_E}$, *percentile bound* $\underline{J_p}$ ($p = 50\%$), and *absolute lower bound* $\underline{J}$. Each column corresponds to one smoothing variance. Solid lines represent the certified reward bounds, and dashed lines show the empirical performance under PGD.

**Tightness of the certification $\underline{J_E}$, $\underline{J_p}$, and $\underline{J}$.** We compare the empirical cumulative rewards achieved under PGD attacks with our certified lower bounds. First, the empirical results are consistently lower bounded by our certifications, validating the correctness of our bounds. Regarding the tightness, the improved $\underline{J_p}$ is much tighter than the loose $\underline{J_E}$, supported by discussions in Section 5.1. The tightness of $\underline{J}$ can be reflected by the zero gap between the certification and the empirical result under a wide range of attack magnitude $\varepsilon$. Furthermore, the certification for SA-MDP (CVX,PGD) are quite tight for Freeway under large attack magnitudes, and RadialRL demonstrates its superiority on Freeway. Additionally, different methods may achieve the same empirical results under attack, yet their certifications differ tremendously, indicating the importance of the robustness *certification*.

**Game Properties.** The certified robustness of any method on Freeway is much higher than that on Pong, indicating that Freeway is a more stable game than Pong, also shown in Mnih et al. (2015).

## 7 RELATED WORK

We briefly review several RL methods that demonstrate empirical robustness. Kos & Song (2017) and Behzadan & Munir (2017b) show that *adversarial training* can increase the agent's resilience. SA-DQN (Zhang et al., 2020) leverages *regularization* to encourage the top-1 action to stay unchanged under perturbation. Radial-RL (Oikarinen et al., 2020) minimizes an adversarial loss function that incorporates the upper bound of the perturbed loss. CARRL (Everett et al., 2021) computes the lower bounds of Q values under perturbation for action selection, but it is only suitable for low-dimensional environments. Most of these works only provide empirical robustness, with Zhang et al. (2020) and Fischer et al. (2019) additionally provide robustness certificates at state level. To the best of our knowledge, this is the *first* work providing robustness certification for the cumulative reward of RL methods. Broader discussions and comparisons of related work are in Appendix F.

**Conclusions.** To provide the robustness certification for RL methods, we propose a general framework CROP, aiming to provide certification based on two criteria. Our evaluations show that certain empirically robust RL methods are certifiably robust for specific RL environments.

ACKNOWLEDGMENTS

This work is partially supported by the NSF grant No.1910100, NSF CNS 20-46726 CAR, Alfred P. Sloan Fellowship, and Amazon Research Award.

**Ethics Statement.** In this paper, we prove the first framework for certifying the robustness of RL algorithms against evasion attacks. We aim to certify and therefore potentially improve the trustworthiness of machine learning models, and we do not expect any ethics issues raised by our work.

**Reproducibility Statement.** All theorem statements are substantiated with rigorous proofs in our Appendix. We have upload the source code as the supplementary material for reproducibility purpose.

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

## A  PROOFS

### A.1  PROOF OF LEMMA 1

We recall Lemma 1:

**Lemma 1** (Lipschitz continuity of the smoothed value function). *Given the action-value function $Q^\pi : \mathcal{S} \times \mathcal{A} \to [V_{\min}, V_{\max}]$, the smoothed function $\widetilde{Q}^\pi$ with smoothing parameter $\sigma$ is L-Lipschitz continuous with $L = \frac{V_{\max} - V_{\min}}{\sigma} \sqrt{2/\pi}$ w.r.t. the state input.*

*Proof.* To prove Lemma 1, we leverage the technique in the proof for Lemma 1 of Salman et al. (2019) in their Appendix A.

For each action $a \in \mathcal{A}$, our smoothed value function is

$$\widetilde{Q}^\pi(s, a) := \mathop{\mathbb{E}}_{\Delta \sim \mathcal{N}(0, \sigma^2 I_N)} Q^\pi(s + \Delta, a) = \frac{1}{(2\pi)^{N/2} \sigma^N} \int_{\mathbb{R}^n} Q^\pi(t, a) \exp\left(-\frac{1}{2\sigma^2} \|s - t\|^2\right) dt.$$

Taking the gradient w.r.t. $s$, we obtain

$$\nabla_s \widetilde{Q}^\pi(s, a) = \frac{1}{(2\pi)^{N/2} \sigma^N} \int_{\mathbb{R}^n} Q^\pi(t, a) \frac{1}{\sigma^2} (s - t) \exp\left(-\frac{1}{2\sigma^2} \|s - t\|^2\right) dt.$$

For any unit direction $u$, we have

$$u \cdot \nabla_s \widetilde{Q}^\pi(s, a) \leq \frac{1}{(2\pi)^{N/2} \sigma^N} \int_{\mathbb{R}^n} \frac{V_{\max} - V_{\min}}{\sigma^2} |u(s - t)| \exp\left(-\frac{1}{2\sigma^2} \|s - t\|^2\right) dt$$

$$= \frac{V_{\max} - V_{\min}}{\sigma^2} \cdot \int_{\mathbb{R}^n} \frac{1}{(2\pi)^{1/2} \sigma} |s_i - t| \exp\left(-\frac{1}{2\sigma^2} |s_i - t|^2\right) dt$$

$$\cdot \prod_{j \neq i} \int_{\mathbb{R}^n} \frac{1}{(2\pi)^{1/2} \sigma} \exp\left(-\frac{1}{2\sigma^2} |s_j - t|^2\right) dt$$

$$= \frac{V_{\max} - V_{\min}}{\sigma} \sqrt{\frac{2}{\pi}}$$

Thus, $\widetilde{Q}^\pi$ is $L$-Lipschitz continuous with $L = \frac{V_{\max} - V_{\min}}{\sigma} \sqrt{2/\pi}$ w.r.t. the state input. $\qquad \square$

## A.2 PROOF OF THEOREM 1

We recall Theorem 1:

**Theorem 1.** *Let $Q^\pi : \mathcal{S} \times \mathcal{A} \to [V_{\min}, V_{\max}]$ be a trained value network, $\widetilde{Q}^\pi$ be the smoothed function with (2). At time step $t$ with state $s_t$, we can compute the lower bound $r_t$ of maximum perturbation magnitude $\bar{\varepsilon}(s_t)$ (i.e., $r_t \leq \bar{\varepsilon}(s_t)$, $\bar{\varepsilon}$ defined in Definition 1) for locally smoothed policy $\tilde{\pi}$:*

$$r_t = \frac{\sigma}{2} \left( \Phi^{-1} \left( \frac{\widetilde{Q}^\pi(s_t, a_1) - V_{\min}}{V_{\max} - V_{\min}} \right) - \Phi^{-1} \left( \frac{\widetilde{Q}^\pi(s_t, a_2) - V_{\min}}{V_{\max} - V_{\min}} \right) \right), \tag{6}$$

*where $\Phi^{-1}$ is the inverse CDF function, $a_1$ is the action with the highest $\widetilde{Q}^\pi$ value at state $s_t$, and $a_2$ is the runner-up action. We name the lower bound $r_t$ as certified radius for the state $s_t$.*

We first present a lemma that can help in the proof of Theorem 1.

**Lemma 3.** *Let $\Phi$ be the CDF of a standard normal distribution, the mapping $\eta_a(s) := \sigma \cdot \Phi^{-1} \left( \frac{\widetilde{Q}^\pi(s,a) - V_{\min}}{V_{\max} - V_{\min}} \right)$ is 1-Lipschitz continuous.*

The lemma can be proved following the same technique as the proof for Lemma 1 in Appendix A.1. The detailed proof can be referred to in the proof for Lemma 2 of Salman et al. (2019) in their Appendix A. We next show how to leverage Lemma 3 to prove Theorem 1.

*Proof for Theorem 1.* Let the perturbation be $\delta_t$, based on the Lipschitz continuity of the mapping $\eta$, we have

$$\eta_{a_1}(s_t) - \eta_{a_1}(s_t + \delta_t) \leq \|\delta_t\|_2, \tag{7}$$

$$\eta_{a_2}(s_t + \delta_t) - \eta_{a_2}(s_t) \leq \|\delta_t\|_2. \tag{8}$$

Suppose that under perturbation $\delta_t$, the action selection would be misled in the sense that the smoothed value for the original action $a_1$ is lower than that of another action $a_2$, *i.e.*, $\widetilde{Q}^\pi(s_t + \delta_t, a_1) \leq \widetilde{Q}^\pi(s_t + \delta_t, a_2)$. Then, based on the monotonicity of $\eta$, we have

$$\eta_{a_1}(s_t + \delta) \leq \eta_{a_2}(s_t + \delta). \tag{9}$$

Summing up (7), (8), and (9), we obtain

$$\|\delta_t\|_2 \geq \frac{1}{2} \left( \eta_{a_1}(s_t) - \eta_{a_2}(s_t) \right)$$

$$= \frac{\sigma}{2} \left( \Phi^{-1} \left( \frac{\widetilde{Q}^\pi(s_t, a_1) - V_{\min}}{V_{\max} - V_{\min}} \right) - \Phi^{-1} \left( \frac{\widetilde{Q}^\pi(s_t, a_2) - V_{\min}}{V_{\max} - V_{\min}} \right) \right)$$

which is a lower bound of the *maximum perturbation magnitude* $\bar{\varepsilon}(s_t)$ that can be tolerated at state $s_t$. Hence, when $r_t$ takes the value of the computed lower bound, it satisfies the condition that $r_t \leq \bar{\varepsilon}(s_t)$. □

## A.3 PROOF OF LEMMA 2

We recall Lemma 2:

**Lemma 2** (Lipschitz continuity of smoothed perturbed return function)**.** *Let $F$ be the perturbed return function function defined in (4), the smoothed perturbed return function $\widetilde{F}_\pi$ is $\frac{(J_{\max} - J_{\min})}{\sigma} \sqrt{2/\pi}$-Lipschitz continuous, where $\widetilde{F}_\pi \left( \oplus_{t=0}^{H-1} \delta_t \right) := \mathbb{E}_{\Delta \sim \mathcal{N}(0, \sigma^2 I_{H \times N})} F_\pi \left( \oplus_{t=0}^{H-1} (\delta_t + \Delta_t) \right)$.*

*Proof.* The proof can be done in a similar fashion as the proof for Lemma 1 in Appendix A.1. Compared with the smoothed value network where the expectation is taken over the sampled states, here, similarly, the smoothed perturbed return function is derived by taking the expectation over sampled $\sigma$-randomized trajectories. The difference is that the output range of the Q-network is $[V_{\min}, V_{\max}]$, while the output range of the perturbed return function is $[J_{\min}, J_{\max}]$. Thus, the smoothed perturbed return function $\widetilde{F}$ is $\frac{(J_{\max} - J_{\min})}{\sigma} \sqrt{2/\pi}$-Lipschitz continuous. □

### A.4 Proof of Theorem 2

We recall the definition of $\widetilde{F}_\pi$ as well as Theorem 2:

$$\widetilde{F}_\pi\left(\oplus_{t=0}^{H-1}\delta_t\right) := \underset{\zeta\sim\mathcal{N}(0,\sigma^2 I_{H\times N})}{\mathbb{E}} F_\pi\left(\oplus_{t=0}^{H-1}(\delta_t+\zeta_t)\right).$$

**Theorem 2** (Expectation bound). *Let* $\underline{J_E} = \widetilde{F}_\pi\left(\oplus_{t=0}^{H-1}\mathbf{0}\right) - L\varepsilon\sqrt{H}$, *where* $L = \frac{(J_{\max}-J_{\min})}{\sigma}\sqrt{2/\pi}$. *Then* $\underline{J_E} \leq \mathbb{E}\left[J_\varepsilon(\pi')\right]$.

*Proof.* We note the following equality

$$\widetilde{F}_\pi(\oplus_{t=0}^{H-1}\delta_t) \overset{(a)}{=} \mathbb{E}\left[F_\pi\left(\oplus_{t=0}^{H-1}(\delta_t+\zeta_t)\right)\right] \overset{(b)}{=} \mathbb{E}\left[F_{\pi'}\left(\oplus_{t=0}^{H-1}\delta_t\right)\right] \overset{(c)}{=} \mathbb{E}\left[J_\varepsilon^H(\pi')\right], \tag{10}$$

where (a) comes from the definition of the smoothed perturbed return function $\widetilde{F}_\pi$, (b) is due to the definition of the $\sigma$-randomized policy $\pi'$, and (c) arises from the definition of the perturbed cumulative reward $J_\varepsilon^H$. Thus, the expected perturbed cumulative reward $\mathbb{E}\left[J_\varepsilon^H(\pi')\right]$ is equivalent to the smoothed perturbed return function $\widetilde{F}_\pi\left(\oplus_{t=0}^{H-1}\delta_t\right)$. Furthermore, since the distance between the all-zero $\oplus_{t=0}^{H-1}\mathbf{0}$ and the adversarial perturbations $\oplus_{t=0}^{H-1}\delta_t$ is bounded by $\varepsilon\sqrt{H}$, leveraging the Lipschitz smoothness of $\widetilde{F}$ in Lemma 2, we obtain the lower bound of the expected perturbed cumulative reward $\mathbb{E}\left[J_\varepsilon^H(\pi')\right]$ as $\widetilde{F}_\pi\left(\oplus_{t=0}^{H-1}\mathbf{0}\right) - L\varepsilon\sqrt{H}$. $\qquad\square$

### A.5 Proof of Theorem 3

We recall the definition of $\widetilde{F}_\pi^p$ as well as Theorem 3:

$$\widetilde{F}_\pi^p\left(\oplus_{t=0}^{H-1}\delta_t\right) := \sup_y\left\{y\in\mathbb{R}\mid\mathbb{P}\left[F_\pi\left(\oplus_{t=0}^{H-1}(\delta_t+\zeta_t)\right)\leq y\right]\leq p\right\}. \tag{11}$$

**Theorem 3** (Percentile bound). *Let* $\underline{J_p} = \widetilde{F}_\pi^{p'}\left(\oplus_{t=0}^{H-1}\mathbf{0}\right)$, *where* $p' := \Phi\left(\Phi^{-1}(p) - \varepsilon\sqrt{H}/\sigma\right)$. *Then* $\underline{J_p} \leq$ *the* $p$-*th percentile of* $J_\varepsilon(\pi')$.

*Proof.* To prove Theorem 3, we leverage the technique in the proof for Lemma 2 of Chiang et al. (2020) in their Appendix B.

For brevity, we abbreviate $\delta := \oplus_{t=0}^{H-1}\delta_t$ and redefine the plus operator such that $\delta+\zeta := \oplus_{t=0}^{H-1}(\delta_t+\zeta_t)$. Then, similar to Lemma 3, we have the conclusion that

$$\delta \mapsto \sigma\cdot\Phi^{-1}\left(\mathbb{P}\left[F_\pi(\delta+\zeta)\leq\widetilde{F}_\pi^{p'}(\mathbf{0})\right]\right)$$

is 1-Lipschitz continuous, where $\zeta\sim\mathcal{N}(0,\sigma^2 I_{H\times N})$.

Thus, under the perturbations $\delta_t\in\mathcal{B}^\varepsilon$ for $t=0\ldots H-1$, we have

$$
\begin{aligned}
\Phi^{-1}\left(\mathbb{P}\left[F_\pi(\delta+\zeta)\leq\widetilde{F}_\pi^{p'}(\mathbf{0})\right]\right) &\leq \Phi^{-1}\left(\mathbb{P}\left[F_\pi(\zeta)\leq\widetilde{F}_\pi^{p'}(\mathbf{0})\right]\right) + \frac{\|\delta\|_2}{\sigma} \\
&\leq \Phi^{-1}\left(\mathbb{P}\left[F_\pi(\zeta)\leq\widetilde{F}_\pi^{p'}(\mathbf{0})\right]\right) + \frac{\varepsilon\sqrt{H}}{\sigma} \quad \textit{(Since } \|\delta\|_2\leq\varepsilon\sqrt{H}\textit{)} \\
&= \Phi^{-1}(p') + \frac{\varepsilon\sqrt{H}}{\sigma} \quad\quad\quad\quad\ \textit{(By definition of } \widetilde{F}_\pi^p\textit{)} \\
&= \Phi^{-1}(p) \quad\quad\quad\quad\quad\quad\quad\ \textit{(By definition of } p'\textit{)}.
\end{aligned}
$$

Since $\Phi^{-1}$ monotonically increase, this implies that $\mathbb{P}\left[F_\pi(\delta+\zeta)\leq\widetilde{F}_\pi^{p'}(\mathbf{0})\right]\leq p$. According to the definition of $\widetilde{F}_\pi^p$ in (11), we see that $\widetilde{F}_\pi^{p'}(\mathbf{0})\leq\widetilde{F}_\pi^p(\delta)$, *i.e.*, $\underline{J_p}\leq$ the $p$-th percentile of $J_\varepsilon(\pi')$. Hence, the theorem is proved. $\qquad\square$

### A.6 Proof of Theorem 4

We recall Theorem 4:

**Theorem 4.** *Let $(r_t^1, \ldots, r_t^{|\mathcal{A}|-1})$ be a sequence of certified radii for state $s_t$ at time step $t$, where $r_t^k$ denotes the radius such that if $\varepsilon < r_t^k$, the possible action at time step $t$ will belong to the actions corresponding to top $k$ action values of $\widetilde{Q}$ at state $s_t$. The definition of $r_t$ in Theorem 1 is equivalent to $r_t^1$ here. The radii can be computed similarly as follows:*

$$r_t^k = \frac{\sigma}{2}\left(\Phi^{-1}\left(\frac{\widetilde{Q}^\pi(s_t, a_1) - V_{\min}}{V_{\max} - V_{\min}}\right) - \Phi^{-1}\left(\frac{\widetilde{Q}^\pi(s_t, a_{k+1}) - V_{\min}}{V_{\max} - V_{\min}}\right)\right), \quad 1 \le k < |\mathcal{A}|,$$

*where $a_1$ is the action of the highest $\widetilde{Q}$ value at state $s_t$ and $a_{k+1}$ is the $(k+1)$-th best action. We additionally define $r_t^0(s_t) = 0$, which is also compatible with the definition above.*

*Proof.* Replacing $a_2$ with $a_{k+1}$ in the proof for Theorem 1 in Appendix A.2 directly leads to Theorem 4. $\qquad\square$

## B    ADDITIONAL DETAILS OF CERTIFICATION STRATEGIES

In this section, we cover the concrete details regarding the implementation of our three certification strategies, as a complement to the high-level ideas introduced back in Section 4 and Section 5.

### B.1    DETAILED ALGORITHM OF CROP-LOACT

---

**Algorithm 1:** CROP-LOACT: Local smoothing for certifying per-state action

---

**Input:** state $s$, trained value network $Q^\pi$ with range $[V_{\min}, V_{\max}]$; parameters for smoothing: sampling times $m$, smoothing variance $\sigma^2$, one-sided confidence parameter $\alpha$

**Output:** smoothed value network $\widetilde{Q}^\pi$, selected action $a$, certification indicator $cert$, certified radius $r$ constant $L$

▷ Step 1:    smoothing

1 Generate noise samples $\delta_i \sim \mathcal{N}(0, \sigma^2 I)$ for $1 \le i \le m$

2 **for** *each action $a \in \mathcal{A}$* **do**

    ▷ clipping and averaging

3     $\widetilde{Q}^\pi(s, a) \leftarrow \frac{1}{m}\sum_{i=1}^m \texttt{clip}(Q^\pi(s + \delta_i, a), \min = V_{\min}, \max = V_{\max})$

4 $a_1, a_2 \leftarrow$ best action and runner-up action given by $\widetilde{Q}^\pi$

▷ Step 2:    certification

5 $\Delta = (V_{\max} - V_{\min})\sqrt{\frac{1}{2m}\ln\frac{1}{\alpha}}$

    ▷ confidence interval

6 **if** $\widetilde{Q}^\pi(s, a_1) \ge \widetilde{Q}^\pi(s, a_2) + 2\Delta$ **then**

    ▷ certification success

7     $cert \leftarrow \texttt{True}$

8     $r \leftarrow \frac{\sigma}{2}\big(\Phi^{-1}\big(\frac{\widetilde{Q}^\pi(s,a_1)-\Delta-V_{\min}}{V_{\max}-V_{\min}}\big) - \Phi^{-1}\big(\frac{\widetilde{Q}^\pi(s,a_2)+\Delta-V_{\min}}{V_{\max}-V_{\min}}\big)\big)$

9 **else**

    ▷ certification failure

10     $cert \leftarrow \texttt{False}$

11     $r \leftarrow \texttt{undefined}$

12 **return** $\widetilde{Q}^\pi, a_1, cert, r$

---

We present the concrete algorithm of CROP-LOACT in Algorithm 1 for the procedures introduced in Section 4.2. For each given state $s_t$, we first perform Monte Carlo sampling (Cohen et al., 2019; Lecuyer et al., 2019) to achieve local smoothing. Based on the smoothed value function $\widetilde{Q}^\pi$, we then compute the robustness certification for per-state action, *i.e.*, the certified radius $r_t$ at the given state $s_t$, following Theorem 1.

**Detailed Inference Procedure.**    During inference, we invoke the model with $m$ samples of Gaussian noise at each time step, and use the averaged Q value on these $m$ noisy samples to obtain the greedy action selection and compute the certification. This procedure is similar to PREDICT in Cohen et al. (2019), but since RL involves multiple step decisions, we do not take the "abstain" decision as in PREDICT in Cohen et al. (2019); instead, CROP-LOACT will take the greedy action at all steps no matter whether the action can be certified or not.

**Estimation of the Algorithm Parameters** $V_{\min}, V_{\max}$**.**    Our estimate is obtained via sampling the trajectories and calculating the Q values associated with the state-action pairs along these trajectories. Since we perform clipping using the obtained bounds, the certification is sound. The cost for estimating $V_{\min}$ and $V_{\max}$ is essentially associated with the number of sampled trajectories, the number of steps in each trajectory, the number of sampled Gaussian noise $m$ per step, and the cost of doing one forward pass of the Q network; thus, the estimation can be done with low cost. Furthermore,

---

**Algorithm 2:** CROP-GRE: Global smoothing for certifying cumulative reward

---

**Input:** initial state distribution $d_0$, trained value network $Q^\pi$, game cumulative reward range $[J_{\min}, J_{\max}]$, number of steps in an episode $H$, perturbation magnitude at each state $\varepsilon$, percentile $p$; parameters for smoothing: sampling times $m$, smoothing variance $\sigma^2$, one-sided confidence parameter $\alpha$

**Output:** Expectation bound $\underline{J_E}$, $p$-th percentile bound $\underline{J_p}$

▷ Step 1: smoothing

1  **for** $i = 1$ *to* $m$ **do**

2      $s_0 \sim d_0, J_i^C \leftarrow 0$     ▷ initialization

3      Generate macro-state noise $\delta_t \sim \mathcal{N}(0, \sigma^2 I)$ for $0 \le t < H$

4      **for** $t = 0$ *to* $H - 1$ **do**

5         $a_t \leftarrow \operatorname{argmax}_a Q(s_t + \delta_t, a)$

6         Execute action $a_t$ and observe reward $re_t$ and next state $s_{t+1}$     ▷ take a step

7         $J_i^C \leftarrow J_i^C + re_t$     ▷ accumulate the reward

▷ Step 2.1: certifying expectation bound

8  $\widetilde{F} \leftarrow \frac{1}{m} \sum_{i=1}^m J_i^C$     ▷ smoothed actual reward

9  $\Delta_{\text{conf}} \leftarrow (R_{\max} - R_{\min}) \sqrt{\frac{\ln(1/\alpha)}{2m}}$     ▷ confidence interval

10  $\Delta_{\text{lip}} \leftarrow \frac{R_{\max} - R_{\min}}{\sigma} \sqrt{\frac{2}{\pi}} \cdot \varepsilon \sqrt{H}$     ▷ bound given by Lipschitz continuity

11  $\underline{J_E} \leftarrow \widetilde{F} - \Delta_{\text{conf}} - \Delta_{\text{lip}}$

12

▷ Step 2.2: certifying percentile bound

13  $k \leftarrow \textsc{ComputeOrderStats}(\varepsilon, \sigma, p, m, H)$

14  $\underline{J_p} \leftarrow k$-th smallest value in $\{J_i^C\}_{i=1}^m$

15

16  **return** $\underline{J_E}, \underline{J_p}$

---

we can balance the trade-off between the accuracy and efficiency of the estimation, and for any configuration of $V_{\min}$ and $V_{\max}$, the certification will invariably be sound (reason above).

### B.2 DETAILED ALGORITHM OF CROP-GRE

We present the concrete algorithm of CROP-GRE in Algorithm 2 for the procedures introduced in Section 5.1. Similarly to CROP-LOACT, the algorithm also consists of two parts: performing smoothing and computing certification, where we compute both the expectation bound $\underline{J_E}$ and the percentile bound $\underline{J_p}$.

**Step 1: Global smoothing.** We adopt Monte Carlo sampling (Cohen et al., 2019; Lecuyer et al., 2019) to estimate the smoothed perturbed return function $\widetilde{F}$ by sampling multiple $\sigma$-randomized trajectories via drawing $m$ noise sequences. For each noise sequence $\zeta \sim \mathcal{N}(0, \sigma^2 I_{H \times N})$, we apply noise $\zeta_t$ to the input state $s_t$ sequentially, and obtain the sum of the reward $J_i^C = \sum_{t=0}^{H-1} re_t$ as the return for this $\sigma$-randomized trajectory. We then aggregate the smoothed perturbed return values $\{J_i^C\}_{i=1}^m$ via mean smoothing and percentile smoothing.

**Step 2: Certification for perturbed cumulative reward.** First, we compute the *expectation bound* $\underline{J_E}$ using Theorem 2. Since the smoothed perturbed return function $\widetilde{F}$ is obtained based on $m$ sampled noise sequences, we use Hoeffding's inequality (Hoeffding, 1994) to compute the lower bound of the random variable $\widetilde{F} \left( \oplus_{t=0}^{H-1} \mathbf{0} \right)$ with a confidence level $\alpha$. We then calculate the lower bound of $\widetilde{F} \left( \oplus_{t=0}^{H-1} \delta_t \right)$ under all possible $\ell_2$-bounded perturbations $\delta_t \in \mathcal{B}^\varepsilon$ leveraging the smoothness of $\widetilde{F}$.

We then compute the *percentile bound* $\underline{J_p}$ using Theorem 3. We let $J_i^C$ be sorted increasingly, and perform normal approximations (Stein et al., 1972) to compute the largest empirical order statistic $J_k^C$ such that $\mathbb{P}\left[ \underline{J_p} \ge J_k^C \right] \ge 1 - \alpha$. The empirical order statistic $J_k^C$ is then used as the proxy of $\underline{J_p}$ under $\alpha$ confidence level. We next provide detailed explanations for $\textsc{ComputeOrderStats}$, which aims to compute the order $k$ using binomial formula plus normal approximation.

**Estimation of $\widetilde{F}_\pi$ (defined in Lemma 2).** For computing the expectation bound (*i.e.*, the lower bound of $\mathbb{E}_\Delta \left[ J_\varepsilon(\pi') \right]$), an intermediate step is to estimate $\widetilde{F}_\pi$ (as shown in line 8 of Algorithm 2).

We emphasize that the *accuracy* of the estimation does not influence the *soundness* of the lower bound calculation. The reason is given below. The number of sampled randomized trajectories $m$ controls

the trade-off between the estimation *efficiency* and *accuracy*, as well as the *tightness* of the derived lower bound. Concretely, a small $m$ would provide high efficiency, low estimation accuracy, and loose lower bound; but the lower bound is always *sound*, since our algorithm explicitly accounts for the *inaccuracy* associated with $m$ via leveraging the Hoeffding's inequality in line 9 in Algorithm 2.

**Details of COMPUTEORDERSTATS.** We consider the sorted sequence $J_1^C \leq J_2^C \leq \cdots \leq J_m^C$. We additionally set $J_0^C = -\infty$ and $J_{m+1}^C = \infty$. Our goal is to find the largest $k$ such that $\mathbb{P}\left[J_{\underline{p}} \geq J_k^C\right] \geq 1 - \alpha$. We evaluate the probability explicitly as follows:

$$\mathbb{P}\left[J_{\underline{p}} \geq J_k^C\right] = \sum_{i=k}^{m} \mathbb{P}\left[J_i^C \leq J_{\underline{p}} < J_{i+1}^C\right] = \sum_{i=k}^{m} \binom{m}{i}(p')^i(1-p')^{m-i}.$$

Thus, the condition $\mathbb{P}\left[J_{\underline{p}} \geq J_k^C\right] \geq 1 - \alpha$ is equivalent to

$$\sum_{i=0}^{k-1} \binom{m}{i}(p')^i(1-p')^{m-i} \leq \alpha. \tag{12}$$

Given large enough $m$, the LHS of (12) can be approximated via a normal distribution with mean equal to $mp'$ and variance equal to $mp'(1-p')$. Concretely, we perform binary search to find the largest $k$ that satisfies the constraint.

We finally explain the upper bound of $\varepsilon$ that can be certified for each given smoothing variance. In practical implementation, for a given sampling number $m$ and confidence level parameter $\alpha$, if $p'$ is too small, then the condition (12) may not be satisfied even for $k = 1$. This implies that the existence of an upper bound of $\varepsilon$ that can be certified for each smoothing parameter $\sigma$, recalling that $p' := \Phi\left(\Phi^{-1}(p) - \varepsilon\sqrt{H}/\sigma\right)$.

**Detailed Inference and Certification Procedures.** We next provide detailed descriptions of the inference and certification procedures, respectively.

*Inference procedure.* We deploy the $\sigma$-randomized policy $\pi'$ as defined in Definition 4 in Section 5.1. That is, for each observed state, we sample one time of Gaussian noise $\Delta_t$ to add to $s_t$, and take action according to this single randomized observation $a_t = \pi(s_t + \Delta_t)$. Thus, in deployment time, the $\Delta$-randomized policy $\pi'$ executes on *one* rollout—at each step it takes one observation and chooses one action (following the procedure described above).

*Certification procedure.* We sample $m$ randomized trajectories in CROP-GRE instead of sampling $m$ noisy states per time step as in CROP-LOACT. For each sampled randomized trajectory, at each time step in the trajectory, we invoke the model with 1 sample of Gaussian noise (as shown in (4)). Given the $m$ randomized trajectories and the cumulative reward for each of them, we compute the certification using Theorem 2 and Theorem 3.

**The Algorithm Parameters $J_{\min}, J_{\max}$.** We use the default values of $J_{\min}, J_{\max}$ in the game specifications rather than estimate them, which are independent with the trained models. Thus there is no computational cost at all for obtaining the two parameters. As mentioned under Theorem 2 in Section 5.1, using these default values may induce a loose bound in practice, so we further propose the percentile smoothing that aims to eliminate the dependency of our certification on $J_{\min}$ and $J_{\max}$, thus achieving a tighter bound.

### B.3 DETAILED ALGORITHMS OF CROP-LORE

In the following, we will explain the *trajectory exploration and expansion*, the *growth of perturbation magnitude*, and *optimization tricks* in details.

**Trajectory Exploration and Expansion.** CROP-LORE organizes all possible trajectories in the form of a search tree and progressively grows it. Each node of the tree represents a state, and the depth of the node is equal to the time step of the corresponding state in the trajectory. The root node (at depth 0) represents the initial state $s_0$. For each node, leveraging Theorem 4, we compute a non-decreasing sequence $\{r^k(s)\}_{k=0}^{|\mathcal{A}|-1}$ corresponding to required perturbation radii for $\pi$ to choose each alternative action (the subscript $t$ is omitted for brevity). Suppose the current $\varepsilon$ satisfies $r^i(s) \leq \varepsilon < r^{i+1}(s)$. We grow $(i + 1)$ branches from current state $s$ corresponding to the original action and $i$ alternative actions since $\varepsilon \geq r^j(s)$ for $1 \leq j \leq i$. For nodes on the newly expanded branch, we repeat the same

---

**Algorithm 3:** CROP-LORE: Adaptive search for certifying cumulative reward

---

**Input:** Enivronment $\mathcal{E} = (\mathcal{S}, \mathcal{A}, R, \Gamma, d_0)$, trained value network $Q^\pi$ with range $[V_{\min}, V_{\max}]$; parameters for randomized smoothing: sampling times $m$, smoothing variance $\sigma^2$, one-sided confidence parameter $\alpha$

**Output:** a map $M$ that maps an attack magnitude $\varepsilon$ to the corresponding certified lower bound of reward $J$

▷ Initialize global variables

1  $p\_que \leftarrow \emptyset$       ▷ initialize an empty priority queue containing tuples of (state $s$, action $a$, radius $r$, reward $J$), sorted by increasing $r$

2  $M \leftarrow \emptyset$

3  $J_{\text{global}} \leftarrow \infty$       ▷ initialize global minimum reward

4  $\Delta = (V_{\max} - V_{\min}) \sqrt{\frac{1}{2m} \ln \frac{1}{\alpha}}$       ▷ confidence bound

5

6  **Function** GETACTIONS $(s, \varepsilon_{\lim}, J_{\text{cur}})$:

7      Generate noise samples $\delta_i \sim \mathcal{N}(0, \sigma^2 I)$ for $1 \leq i \leq m$

8      **for** *each action $a \in \mathcal{A}$* **do**

9          $\widetilde{Q}^\pi(s, a) \leftarrow \frac{1}{m} \sum_{i=1}^{m} \text{clip}(Q^\pi(s + \delta_i, a), \min = V_{\min}, \max = V_{\max})$

10     $a^\star \leftarrow \text{argmax}_{a \in \mathcal{A}} \widetilde{Q}^\pi(s, a)$

11     $a\_list \leftarrow \emptyset$

12     **for** *each action $a \in \mathcal{A}$* **do**

13         **if** $\Gamma(s, a) = \perp$ **then**

14             **continue**

15         **if** $\widetilde{Q}^\pi(s, a^\star) \geq \widetilde{Q}^\pi(s, a) + 2\Delta$ **then**

16             $r \leftarrow \frac{\sigma}{2}(\Phi^{-1}(\frac{\widetilde{Q}^\pi(s, a^\star) - \Delta - V_{\min}}{V_{\max} - V_{\min}}) - \Phi^{-1}(\frac{\widetilde{Q}^\pi(s, a) + \Delta - V_{\min}}{V_{\max} - V_{\min}}))$

17         **else**

18             $r \leftarrow 0$

19         **if** $r \leq \varepsilon_{\lim}$ **then**

                ▷ take possible actions

20             $a\_list \leftarrow a\_list \cup \{a\}$

21         **else**

            ▷ store impossible actions in queue for later expansion

22             $p\_que.\text{push}((s, a, r, J_{\text{cur}}))$

23     **return** $a\_list$

24

25 **Procedure** EXPAND $(s, \varepsilon_{\lim}, J_{\text{cur}})$:

26     **if** $J_{\text{cur}} \geq J_{\text{global}}$ **then**

27         **return** 0       ▷ pruning

28     $a\_list \leftarrow$ GETACTIONS $(s, \varepsilon_{\lim}, J_{\text{cur}})$

29     **if** $a\_list = \emptyset$ **then**

30         $J_{\text{global}} \leftarrow \min(J_{\text{global}}, J_{\text{cur}})$

31         **return** 0

32     **for** $a \in a\_list$ **do**

33         $s' \leftarrow \Gamma(s, a)$

34         $ret \leftarrow$ EXPAND $(s', \varepsilon_{\lim}, J_{\text{cur}} + R(s, a))$

35

36 $s_0 \sim d_0$       ▷ initialize initial state

37 EXPAND $(s_0, \varepsilon_{\lim} = 0, J_{\text{cur}} = 0)$       ▷ expand initial trajectory

38 **while** *True* **do**

39     **if** $p\_que = \emptyset$ **then**

40         **break**

            ▷ pop out the first element

41     $(s, a, r, J) \leftarrow p\_que.\text{pop}()$

            ▷ examine the next first element

42     $(\_, \_, r', \_) \leftarrow p\_que.\text{top}()$

            ▷ derive the critical $\varepsilon$'s

43     $\varepsilon \leftarrow r, \varepsilon' \leftarrow r'$

            ▷ obtain a pair of mapping

44     $M[\varepsilon] \leftarrow J_{\text{global}}$

            ▷ expand the tree from the new node

45     EXPAND $(\Gamma(s, a), \varepsilon', J + R(s, a))$

---

procedure to expand the tree with depth-first search (Tarjan, 1972) until the terminal state of the game is reached or the node depth reaches $H$. As we expand, we keep the record of cumulative reward for each trajectory and update the lower bound $\underline{J}$ when reaching the end of the trajectory if necessary.

**Perturbation Magnitude Growth.** When all trajectories for perturbation magnitude $\varepsilon$ are explored, we need to increase $\varepsilon$ to seek for certification under larger perturbations. Luckily, since the action space is discrete, we do not need to examine every $\varepsilon \in \mathbb{R}^+$ (which is infeasible) but only need to examine the next $\varepsilon$ where the chosen action in some step may change. We leverage *priority queue* (van Emde Boas, 1977) to effectively find out such next "critical" $\varepsilon$. Concretely, along the trajectory exploration, at each tree node, we search for the possible actions and store actions corresponding to $\{r^k(s)\}_{k=i+1}^{|\mathcal{A}|-1}$ into the priority queue, since these actions are exactly those need to be explored when $\varepsilon$ grows. After all trajectories for $\varepsilon$ are fully explored, we pop out the head element from the priority queue as the next node to expand and the next perturbation magnitude $\varepsilon$ to grow. We repeat this process until the priority queue becomes empty or the perturbation magnitude $\varepsilon$ reaches the predefined threshold.

**Additional Optimization.** We adopt some additional optimization tricks to reduce the complexity of the algorithm. First, for environments with no negative reward, we perform pruning to limit the tree size—if the cumulative reward leading to the current node already reaches the recorded lower bound, we can perform pruning, since the tree that follows will not serve to update the lower bound. This largely reduces the potential search space. We additionally adopt the memorization (Michie, 1968) technique which is commonly applied in search algorithms.

We point out a few more potential improvements. *First*, with more specific knowledge of the game mechanisms, the search algorithm can be further optimized. Take the Pong game as an example, given the horizontal speed of the ball, we can compress the time steps where the ball is flying between the two paddles, thus reducing the computation. *Second*, empirical attacks may be efficiently incorporated into the algorithm framework to provide upper bounds that help with pruning.

**Time Complexity.** The time complexity of CROP-LoRE is $O(H|S_{\text{explored}}| \times (\log |S_{\text{explored}}| + |A|T))$, where $|S_{\text{explored}}|$ is the number of explored states throughout the search procedure, which is no larger than cardinality of state set, $H$ is the horizon length, $|A|$ is the cardinality of action set, and $T$ is the time complexity of performing local smoothing. The main bottleneck of the algorithm is the large number of possible states, which is in the worst case exponential to state dimension. However, to provide a sound worst-case certification agnostic to game properties, exploring all possible states may be inevitable.

**Detailed Inference and Certification Procedures and Important Modules in Algorithm 3.** We next provide detailed descriptions of the inference and certification procedures, respectively, along with other important modules.

*Inference procedure.* In CROP-LoRE, we deploy the locally smoothed policy $\tilde{\pi}$ as defined in (2) in Section 4.1. Concretely, for each observed state $s_t$, we sample a batch of Gaussian noise to add to $s_t$, and take action according to the computed mean Q value on the batch of randomized observations. (Actually, the *inference* procedure per step is exactly the same as in CROP-LoAct described in Appendix B.1.)

*Certification procedure.* We obtain the *certification* via Theorem 4 and the adaptive search algorithm, which outputs a collection of pairs $\{(\varepsilon_i, \underline{J}_{\varepsilon_i})\}_{i=1}^{|C|}$ sorted in ascending order of $\varepsilon_i$, where $|C|$ is the length of the collection. The interpretation for this collection of pairs is provided below. For all $\varepsilon'$, let $i$ be the largest integer such that $\varepsilon_i \leq \varepsilon' < \varepsilon_{i+1}$, then as long as the perturbation magnitude $\varepsilon \leq \varepsilon'$, the cumulative reward $J_\varepsilon(\tilde{\pi}) \geq \underline{J}_{\varepsilon_i}$. This is supported by the fact that all certified radii associated with all nodes in the entire expanded tree compose a discrete set of finite cardinality; and the perturbation value between two adjacent certified radii in the returned collection will not lead to a different tree from the tree corresponding to the largest smaller perturbation magnitude. This actually is also the inspiration for the certification design.

*Important modules.* The function GetAction computes the possible actions at a given state $s$ under the limit $\varepsilon$, while the procedure Expand accomplishes the task of expanding upon a given node/state. The main part of the algorithm involves a loop that repeatedly selects the next element from the priority queue, *i.e.*, a node associated with an $\varepsilon$ value, to expand upon.

**Additional Clarifications.** We clarify that the algorithm only requires access to the environment such that it can obtain the reward and next state via *interacting* with the environment $\mathcal{E}$ (*i.e.*, taking action $a$ at state $s$ and obtain next action $s'$ and reward $r$), but does not require access to an oracle transition function $\Gamma$. We further emphasize that access to the environment that supports back-tracking is already sufficient for conducting our adaptive search algorithm.

## C    Discussion on the Certification Methods

We discuss the *advantages* and *limitations* of our certification methods, as well as possible direct *extensions*, hoping to pave the way for future research along similar directions. We also provide more *detailed analysis* that help with the understanding of our algorithms.

**CROP-LoAct.** The algorithm provides state-wise robustness certification in terms of the stability/consistency of the per-state action. It treats each time step *independently*, smooths the given state at the given time step, and provides the corresponding certification. Thus, one potential *extension* is to expand the time window from one time step to a consecutive sequence of several time steps, and provide certification for the sequences of actions for the given window of states.

**CROP-GRe.** As explained in Section 5, the expectation bound $\underline{J_E}$ is too loose to have any practical usage. In comparison, percentile bound $\underline{J_p}$ is much tighter and practical. However, one limitation of $\underline{J_p}$ is that there exists an upper bound of the attack magnitude $\varepsilon$ that can be certified for each $\sigma$,

as explained in Appendix B.2. For attack magnitudes that exceed the upper bound, we can obtain no useful information via this certification (though the upper bound is usually sufficiently large).

*One limitation of* CROP-GRE. Lemma 2 requires knowing the noise added to each step beforehand so as to generate $m$ randomized trajectories given the known noise sequence $\{\delta_t\}_{t=0}^{H-1}$. Thus, it cannot be applied against an adaptive attacker. We clarify that our CROP-LOACT and CROP-LORE does not have such limitation.

**CROP-LORE.** The algorithm provides the absolute lower bound $\underline{J}$ of the cumulative reward for any finite-horizon trajectory with a given initial state. The advantage of the algorithm is that $\underline{J}$ is an absolute lower bound that bounds the worst-case situation (apart from the exceptions due to the probabilistic confidence $\alpha$), rather than the statistical lower bounds $\underline{J_E}$ and $\underline{J_p}$ that characterize the statistical properties of the random variable $J_\varepsilon$.

*One potential pitfall in understanding* $\underline{J}$. What CROP-LORE certifies is the lower bound for the trajectory with *a given initial state* $s_0 \sim d_0$, rather than all possible states in $d_0$. This is because our search starts from a root node of the tree, which is set to the fixed given state $s_0$.

*Confidence of* CROP-LORE. Regarding the *confidence* of our probabilistic certification, we clarify that we consider the independent multiple-test. Concretely, due to the independence of decision making errors, the confidence is $(1 - \alpha)^N$, where $N$ is the maximum number of possible attacked states explored by CROP-LORE. Formally,

$$\Pr[\text{CROP-LORE certification holds}] = \prod_{\substack{\text{all } s \text{ explored} \\ \text{by CROP-LORE}}} \Pr[\text{not make error on } s | \text{not make error on } s_{pre}]$$

$$\stackrel{\text{(i)}}{=} \prod_{\text{all attackable } s} (1 - \alpha) \times \prod_{\text{all unattackable } s} 1$$

$$\stackrel{\text{(ii)}}{=} (1 - \alpha)^N. \tag{13}$$

In the above equation, we can see that (i) leverages the independence which in turn gives a bound of form $(1 - \alpha)^N$. This is because in each step, the event of "certification does not hold" is *independent* since we sample Gaussian noise *independently*. Leveraging such independence, we obtain a confidence lower bound of $(1 - \alpha)^N$. From (ii), we see that the confidence is only related to the number of possible attacked states $N$. This is because for unattackable states, the certification deterministically holds since there is no attack at current step. Therefore, we only need to count the confidence intervals from all possibly attackable steps instead of all states explored. We remark that in practice, the attacker usually has the ability to perturb only a limited number of steps, *i.e.*, $N$ is typically small. Therefore, the confidence is non-trivial.

*Main limitation of* CROP-LORE. The main *limitation* is the high time complexity of the algorithm (details see Appendix B.3). The algorithm has exponential time complexity in worst case despite the existence of several optimizations. Therefore, it is not suitable for environments with a large action set, or when the horizon length is set too long.

**Extension to Policy-based Methods.** Though we specifically study the robustness certification in Q-learning in this paper, our two certification criteria and three certification strategies can be readily extended to policy-based methods. The intuition is that, instead of smoothing the value function in the Q-learning setting, we directly smooth the policy function in policy-based methods. With the smoothing module replaced and the theorems updated, other technical details in the algorithms for certifying the per-state action and the cumulative reward would then be similar.

# D ADDITIONAL EXPERIMENTAL DETAILS

## D.1 DETAILS OF THE ATARI GAME ENVIRONMENT

We experiment with two Atari-2600 environments in OpenAI Gym (Brockman et al., 2016) on top of the Arcade Learning Environment (Bellemare et al., 2013). The *states* in the environments are high dimensional color images ($210 \times 160 \times 3$) and the *actions* are discrete actions that control the agent to accomplish certain tasks. Concretely, we use the `NoFrameskip-v4` version for our experiments,

where the randomness that influences the environment dynamics can be fully controlled by setting the random seed of the environment at the beginning of an episode.

## D.2 DETAILS OF THE RL METHODS

We introduce the details of the nine RL methods we evaluated. Specifically, for each method, we first discuss the algorithm, and then present the implementation details for it.

**StdTrain** (Mnih et al., 2013): StdTrain (naturally trained DQN model) is an algorithm based on Q-learning, in which DQN is used to reprent Q-value function and TD-loss is used to optimize DQN. Our StdTrain model is implemented with Double DQN (Van Hasselt et al., 2016) and Prioritized Experience Replay (Schaul et al., 2015).

**GaussAug** (Behzadan & Munir, 2017b): The algorithm of GaussAug is similar with AtdTrain, in which appropriate Gaussian random noises added to states during training. As for implementation, our GaussAug model is based on the same architecture with the same set of *basic* training techniques of StdTrain, and adds Gaussian random noise with $\sigma = 0.005$, which has the best testing performance under attacks compared with other $\sigma$, to all the frames during training.

**AdvTrain** (Behzadan & Munir, 2017b): Instead of using the original observation to train Double DQN (Van Hasselt et al., 2016), AdvTrain generates adversarial perturbations and applies the perturbations to part of or all of the frames when training. In our case, we add the perturbations generated by 5-step PGD attack to 50% of the frames to make a balance between between stable training and effectiveness of adversarial training.

**SA-MDP (PGD) and SA-MDP (CVX)** (Zhang et al., 2020): Instead of utilizing adversarial noise to train models directly like AdvTrain, SA-MDP (PGD) and SA-MDP (CVX) regularize the loss function with the help of adversarial noise during training in order to train empirically robust RL agents. In our experiment settings, which is the same as SA-MDP (PGD) and SA-MDP (CVX) (Zhang et al., 2020), models consider adversarial noise with $\ell_\infty$-norm $\varepsilon = 1/255$ when solving the maximization for the regularizer and minimize the original TD-loss concurrently.

**RadialRL** (Oikarinen et al., 2020): With the similar idea of adding regularization during training of SA-MDP (PGD) and SA-MDP (CVX), RadialRL compute the worst-case loss instead of regularizing of the bound of the difference of policy distribution under original states and adversarial states. In our case, the RadialRL model uses a linearly increasing perturbation magnitude $\varepsilon$ (from 0 to $1/255$) to compute the adversarial loss.

**CARRL** (Everett et al., 2021): Unlike other methods which try to improve robustness of RL agents during training, CARRL enhances RL agents' robustness during testing time. It computes the lower bound of each action's Q value at each step and take the one with the highest lower bound conservatively. Given that the lower bound is derived via neural network verification methods (Gowal et al., 2018; Weng et al., 2018), CARRL can only be applied to low dimensional environments (Everett et al., 2021). Thus, we only evaluate CARRL on CartPole and Highway, where we set the bound of the $\ell_2$ norm of the perturbation be $\varepsilon = 0.1$ and $\varepsilon = 0.05$ respectively when computing the lower bound of Q value, which demonstrate the best empirical performance under noise and empirical attacks.

**NoisyNet** (Fortunato et al., 2017): Instead of the conventional exploration heuristics for DQN and Dueling agents which is $\varepsilon$-greedy algorithms, NoisyNet adds parametric noise to DQN's weights, which is claimed to aid efficient exploration and yield higher scores than StdTrain. Concretely, our NoisyNet shares the same basic architecture with StdTrain, and samples the network weights during training and testing, where the weights are sampled from a Gaussian distribution with $\sigma = 0.5$.

**GradDQN** (Pattanaik et al., 2018): GradDQN is a variation of AdvTrain, which utilizes conditional value of risk (CVaR) as optimization criteria for robust control instead standard expected long term return. This method maximizes the expected return over worst $\alpha$ percentile of returns, which can prevent the adversary from possible bad states. In our case, we generate the adversarial states with 10-step attack by calculating the direction of gradient and sampling in that direction from $beta1, 1$ distribution in each step.

For StdTrain, SA-MDP (PGD), SA-MDP (CVX), RadialRL, we directly evaluate the trained models provided by the authors[1].

## D.3 RATIONALES FOR SETTINGS AND EXPERIMENTAL DESIGNS

**Finite Horizon Setting.** In this paper, we focus on a finite horizon setting, mainly for the evaluation purpose—we would need to compute the (certified and empirical) cumulative rewards from finite rollouts. We clarify that our criteria, theorem, and algorithm are applicable to the infinite horizon case as well.

**The Role of Practical Attacks in Our Evaluation.** The goal of this paper is to provide certification that comes with theoretical guarantees, and therefore following the standard certified robustness literatures (Cohen et al., 2019; Li et al., 2020; Salman et al., 2019; Jeong & Shin, 2020), there is no need to evaluate the empirical attacks, since the certification always holds as long as the attacks satisfy certain conditions regardless of the actual attack algorithms. The attacks we evaluated in our paper were only for the demonstration purpose and in the hope of providing an example.

## D.4 DETAILED EVALUATION SETUP FOR ATARI GAMES

We introduce the detailed evaluation setups for the three certification methods corresponding to the experiments in Section 6.

**Evaluation Setup for CROP-LOACT.** We report results averaged over 10 episodes and set the length of the horizon $H = 500$. At each time step, we sample $m = 10,000$ noisy states for smoothing. When applying Hoeffding's inequality, we adopt the confidence level parameter $\alpha = 0.05$. Since the input state observations for the two Atari games are in image space, we rescale the input states such that each pixel falls into the range $[0, 1]$. When adding Gaussian noise to the rescaled states, we sample Gaussian noise of zero mean and different variances. Concretely, the standard deviation $\sigma$ is selected among $\{0.001, 0.005, 0.01, 0.03, 0.05, 0.1, 0.5, 0.75, 1.0, 1.5, 2.0, 4.0\}$. We evaluate different parameters for different environments.

**Evaluation Setup for CROP-GRE.** We sample $m = 10,000$ $\sigma$-randomized trajectories, each of which has length $H = 500$, and conduct experiments with the same set of smoothing parameters as in the setup of CROP-LOACT. When accumulating the reward, we set the discount factor $\gamma = 1.0$ with no discounting. We take $\alpha = 0.05$ as the confidence level when applying Hoeffding's inequality in the expectation bound $J_E$, and $\alpha = 0.05$ as the confidence level for COMPUTEORDERSTATS in the percentile bound $J_p$. For the validation of tightness, concretely, we carry out a 10-step PGD attack with $\ell_2$ radius within the perturbation bound at all time steps during testing to evaluate the tightness of our certification.

**Evalution Setup for CROP-LORE.** We set the horizon length as $H = 200$ and sample $m = 10,000$ noisy states with the same set of smoothing variance as in the setup of CROP-LOACT. Similarly, we adopt $\gamma = 1.0$ as the discount factor and $\alpha = 0.05$ as the confidence level when applying Hoeffding's inequality. When evaluating Freeway, we modify the reward mechanism so that losing one ball incurs a zero score rather than a negative score. This modification enables the pruning operation and therefore facilitates the search algorithm, yet still respects the goal of the game. We also empirically attack the policy $\tilde{\pi}$ to validate the tightness of the certification via the same PGD attack as described above. For each set of experiments, we run the attack one time with the same initial state as used for computing the lower bound in Algorithm 3.

## D.5 EXPERIMENTAL SETUP FOR CARTPOLE

We additionally experiment with CartPole-v0 in OpenAI Gym (Brockman et al., 2016) on top of the Arcade Learning Environment (Bellemare et al., 2013). We introduce the experimental setup below.

**Details of the CartPole Environment.** The *state* in the CartPole game is a low dimensional vector of length $4$ and the *action* $\in \{move\ right, move\ left\}$. The goal of the game is to balance the rod on the cart, and one reward point is earned at each timestep when the rod is upright.

---

[1] StdTrain, SA-MDP (PGD), SA-MDP (CVX) from `https://github.com/chenhongge/SA_DQN` and RadialRL from `https://github.com/tuomaso/radial_rl` under Apache-2.0 License

**Implementation Details of the RL Methods on CartPole.** In addition to the eight RL Methods evaluated on Pong and Freeway as shown in Section 6, we evaluate another RL algorithm CARRL (Everett et al., 2021), which is claimed to be robust in low dimensional games like CartPole. This method relies on linear bounds output by the Q network and selects the action whose lower bound of the Q value is the highest. More detailed introduction to the algorithm and description of the implementation details are provided in Appendix D.2.

**Evaluation Setup for CROP-LOACT on CartPole.** We report results averaged over 10 episodes and set the length of the horizon $H = 200$. At each time step, we sample $m = 10,000$ noisy states for smoothing. When applying Hoeffding's inequality, we adopt the confidence level parameter $\alpha = 0.05$. We do not perform rescaling on the state observations. When adding Gaussian noise to states, we sample Gaussian noise of zero mean and different variances. Concretely, the standard deviation $\sigma$ is selected among $\{0.001, 0.005, 0.01, 0.03, 0.05, 0.1\}$.

**Evaluation Setup for CROP-GRE on CartPole.** We sample $m = 10,000$ $\sigma$-randomized trajectories, each of which has length $H = 200$, and conduct experiments with the same set of smoothing parameters as in the setup of CROP-LOACT. All other experiment settings are the same as Pong and Freeway in Appendix D.4.

**Evalution Setup for CROP-LORE on CartPole.** Since the reward of CartPole game is quite dense, we set the horizon length as $H = 10$ and sample $m = 10,000$ noisy states with the same set of smoothing variance as in the setup of CROP-LOACT. All other experiment settings are the same as Pong and Freeway in Appendix D.4.

### D.6    EXPERIMENTAL SETUP FOR HIGHWAY

In addition to the previous environments in OpenAI Gym (Brockman et al., 2016) on the Arcade Learning Environment (Bellemare et al., 2013), we also evaluate our methods in the Highway environment (Leurent, 2018). Specifically, we test out algorithms in the highway-fast-v0 environment.

**Details of the highway Environment.** The *state* in the highway-fast-v0 environment is a $5 \times 5$ matrix, where each line represents a feature vectors of either the ego vehicle or other vehicles closest to the ego vehicle. The feature vector for each vehicle has the form of $[x, y, vx, vy, 1]$, corresponding to the two-dimensional positions and velocities and an additional indicator value. The *actions* are high-level control actions among {*lane left*, *idle*, *lane right*, *faster*, *slower*}. The *reward* mechanism in this environment is associated with the status of the ego vehicle; more concretely, it gives higher reward for the vehicle when it stays at the rightmost lane, moves at high speed, and does not crash into other vehicles.

**Evaluation Setup for CROP-LOACT on highway.** We report results averaged over 10 episodes and set the length of the horizon $H = 30$, which is the maximum length of an episode in the environment configuration. At each time step, we sample $m = 10,000$ noisy states for smoothing. When applying Hoeffding's inequality, we adopt the confidence level parameter $\alpha = 0.05$. We do not perform rescaling on the state observations. When adding Gaussian noise to states, we sample Gaussian noise of zero mean and different variances. Concretely, the standard deviation $\sigma$ is selected among $\{0.005, 0.05, 0.1, 0.5, 0.75, 1.0\}$.

**Evaluation Setup for CROP-GRE on highway.** We sample $m = 10,000$ $\sigma$-randomized trajectories (each of which has length $H = 30$) and conduct experiments with the same set of smoothing parameters as in the setup of CROP-LOACT. All other experiment settings are the same as Pong and Freeway in Appendix D.4.

**Evalution Setup for CROP-LORE on highway.** Since the reward of highway game is also quite dense, we set the horizon length as $H = 10$ as in the CartPole environment and sample $m = 10,000$ noisy states with the same set of smoothing variance as in the setup of CROP-LOACT. All other experiment settings are the same as Pong and Freeway in Appendix D.4.

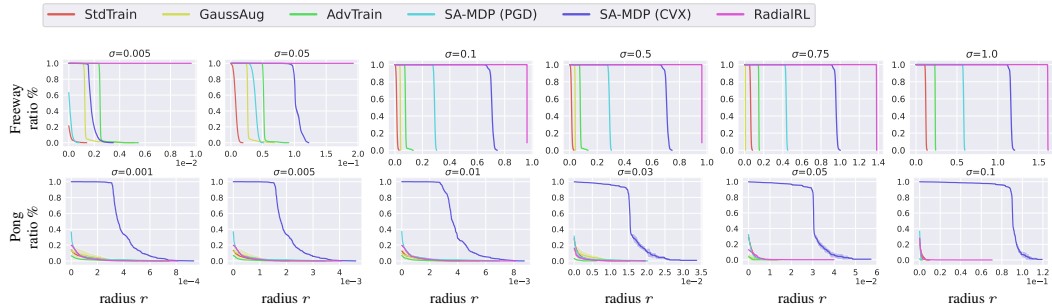

Figure 4: Robustness certification for *per-state action* in terms of certified ratio $\eta_r$ w.r.t. certified radius $r$. Each column corresponds to one smoothing parameter $\sigma$. The shaded area represents the standard deviation.

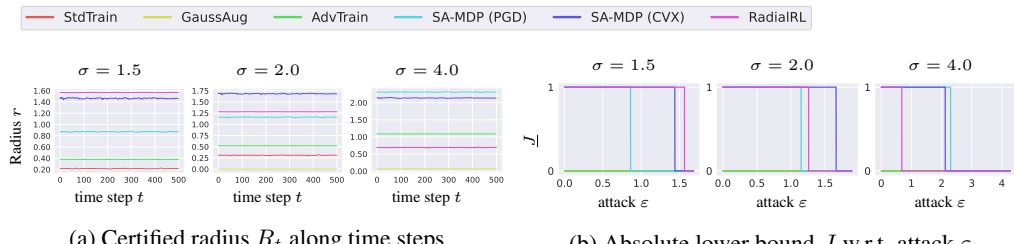

(a) Certified radius $R_t$ along time steps       (b) Absolute lower bound $\underline{J}$ w.r.t. attack $\varepsilon$

Figure 5: Robustness certification for (a) *per-state action* in terms of certified radius $r$ at all time steps, and (b) *cumulative reward* in terms of absolute lower bound $\underline{J}$ w.r.t. the attack magnitude $\varepsilon$. The results are reported under large smoothing parameter $\sigma$ compared with those evaluated in Figure 1 and Figure 3 in the main paper.

# E  ADDITIONAL EVALUATION RESULTS AND DISCUSSIONS

## E.1  ROBUSTNESS CERTIFICATION FOR PER-STATE ACTION – CERTIFIED RATIO

We present the robustness certification for per-state action in terms of the certified ratio $\eta_r$ w.r.t. the certified radius $r$ in Figure 4. From the figure, we see that RadialRL is the most certifiably robust method on Freeway, followed by SA-MDP (CVX) and SA-MDP (PGD); while on Pong, SA-MDP (CVX) is the most robust. These conclusions are highly consistent with those in Section 6.1 as observed from Figure 1. Comparing the curves for Freeway and Pong, we note that Freeway not only achieves larger certified radius *overall*, it also more *frequently* attains large certified radius.

## E.2  RESULTS OF FREEWAY UNDER LARGER SMOOTHING PARAMETER $\sigma$

Since the robustness of the methods RadialRL, SA-MDP (CVX), and SA-MDP (PGD) on Freeway have been shown to improve with the increase of the smoothing parameter $\sigma$ in Section 6, we subsequently evaluate their performance under even larger $\sigma$ values. We present the certification results in Figure 5 and their benign performance in Figure 6.

First of all, as Figure 6 reveals, RadialRL, SA-MDP (CVX), and SA-MDP (PGD) can tolerate quite large noise—the benign performance of these methods does not drop even for $\sigma$ up to $4.0$. For the remaining three methods, the magnitude of noise they can tolerate is much smaller, as already presented in Figure 2 in the main paper. On the other hand, although all the three methods RadialRL, SA-MDP (CVX), and SA-MDP (PGD) invariably attain good benign performance as $\sigma$ grows, their certified robustness does not equally increase. As Figure 5 shows, the certified robustness of SA-MDP (PGD) increases the fastest, while that of RadialRL drops a little. The observa-

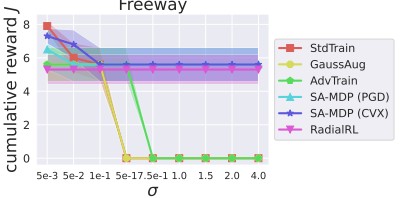

Figure 6: Benign performance of locally smoothed policy $\tilde{\pi}$ under a larger range of smoothing parameter $\sigma$ with clean state observations.

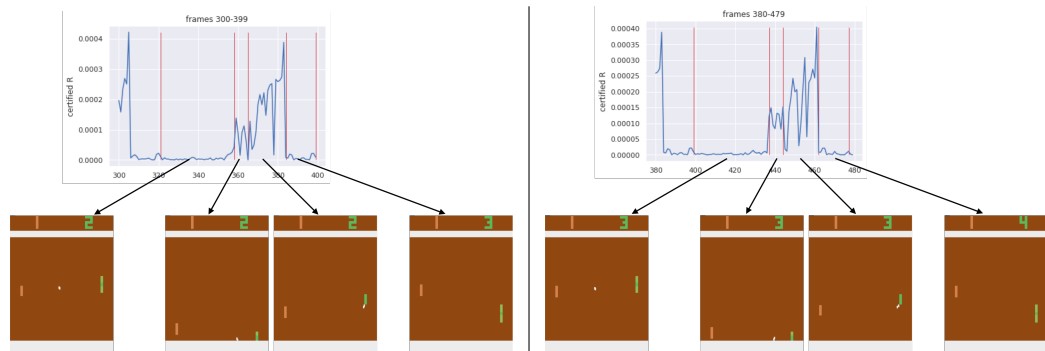

Figure 7: Periodic patterns in Pong. The figure shows two periods (left and right), including the certified radius $r$ w.r.t. the time steps (above), and the selected game frames corresponding to different stages in each period (below). Different periods are highly similar.

tion corresponds exactly to our discussion regarding the tradeoff between value function smoothness and the margin between the values to the top two actions in Section 4.

### E.3 PERIODIC PATTERNS FOR PER-STATE ROBUSTNESS

We specifically study the periodic patterns for per-state robustness in Pong and present the results in Figure 7. In the game frames, our agent controls the green paddle on the right. Whenever the opponent (the orange paddle on the left) misses catching a ball, we earn a point. We present two periods (frame 320-400 and frame 400-480) where our agent each earns a point in the figure above. The total points increase from 2 to 3, and from 3 to 4, respectively.

We note that Pong is a highly periodic game, with each period consisting of the following four game stages corresponding to distinct stages of the *certified robustness*: 1) A ball is initially fired and set flying towards the opponent. In this stage, the certified radius remains low, since our agent does not need to take any critical action. 2) The opponent catches the ball and reverses the direction of the ball. In this stage, the certified radius gradually increases, since the moves of our agent serve as the preparation to catch the ball. 3) The ball bounces from the ground, flies towards our agent, gets caught, and bounces back. In this stage, the certified radius increases rapidly, since every move is critical in determining whether we will be able to catch the ball, *i.e.*, whether our point (action value) will increase. 4) The opponent misses the ball, and the ball disappears from the game view. In this stage, the certified radius drastically plummets to a low value, since there is no clear signal (*i.e.*, ball) in the game for the agent to take any determined action.

The illustration not only helps us understand the semantic meaning of the certified radius, but can further provide guidance on designing empirically robust RL algorithms that leverage the information regarding the high/low certified radius of the states.

### E.4 DETAILED DISCUSSIONS AND SELECTION OF SMOOTHING PARAMETER $\sigma$

We first take Freeway as an example to concretely explain the impact of smoothing variance from the perspective of benign performance and certified results, and then introduce the guidelines for the selection of $\sigma$ based on different certification criteria.

On *Freeway*, as $\sigma$ increases, the *benign performance* of the locally smoothed policy $\tilde{\pi}$ of StdTrain, GaussAug, and AdvTrain decrease rapidly in Figure 2. Also, the *certified radius* of these methods barely gets improved and remains low even as $\sigma$ increases. For the percentile bound $J_p$, these three methods similarly suffer a drastic decrease of *cumulative reward* as $\sigma$ grows. This is however not the case for SA-MDP (PGD), SA-MDP (CVX), and RadialRL, for which not only the *benign performance* do not experience much degradation for $\sigma$ as large as 1.0, but the *certified radius* even steadily increases. In addition, in terms of the percentile bound $J_p$, as $\sigma$ grows, even we increase the attack $\varepsilon$ simultaneously, $J_p$ almost does not decrease. This indicates that larger smoothing variance can bring more robustness to SA-MDP (PGD), SA-MDP (CVX), and RadialRL.

**CROP-LoAct.**  We discuss how to find the sweet spot that enables high benign performance and robustness simultaneously. As $\sigma$ increases, the benign performance of different methods generally decreases due to the noise added, while not all methods decrease at the same rate. On *Freeway*, StdTrain and GaussAug decrease the fastest, while SA-MDP (PGD), SA-MDP (CVX), and $\overline{\text{RadialRL}}$ do not degrade much even for $\sigma$ as large as $4.0$. Referring back to Figure 1 and Figure 5, we see that the certified radius of StdTrain and GaussAug remains low even though $\sigma$ increases, thus the best parameters for these two methods are both around $0.1$.

For SA-MDP (PGD) and SA-MDP (CVX), their certified radius instead steadily increases as $\sigma$ increases to $4.0$, indicating that larger smoothing variance can bring more robustness to these two methods without sacrificing benign performance, offering certified radius larger than $2.0$. As to RadialRL, its certified radius reaches the peak at $\sigma = 1.0$ and then decreases under larger $\sigma$. Thus, $1.0$ is the best smoothing parameter for RadialRL which offers a certified radius of $1.6$. Generally speaking, it is feasible to select an appropriate smoothing parameter $\sigma$ for all methods to increase their robustness, and robust methods will benefit more from this by admitting larger smoothing variances and achieving larger certified radius.

**CROP-GRe.**  We next discuss how to leverage Figure 3 to assist with the selection of $\sigma$. For a given $\varepsilon$ or a known range of $\varepsilon$, we can compare between the lower bounds corresponding to different $\sigma$ and select the one that gives the highest lower bound. For example, if we know that $\varepsilon$ is small in advance, we can go for a relatively small $\sigma$ which retains a quite high lower bound. If we know that $\varepsilon$ will be large, we will instead choose among the larger $\sigma$'s, since a larger $\varepsilon$ will not fall in the certifiable ranges of smaller $\sigma$'s. According to this guideline, we are able to obtain lower bounds of quite high value for RadialRL, SA-MDP (CVX), and SA-MDP (PGD) on Freeway under a larger range of attack $\varepsilon$.

**CROP-LoRe.**  Apart from the conclusion that larger smoothing parameter $\sigma$ often secures higher lower bound $\underline{J}$, we point out that smaller smoothing variances may be able to lead to a higher lower bound than larger smoothing variances for a certain range of $\varepsilon$. This is almost always true for very small $\varepsilon$ since smaller $\sigma$ is sufficient to deal with the weak attack without sacrificing much empirical performance, *e.g.*, in Figure 3, when $\varepsilon = 0.001$, SA-MDP (CVX) on Freeway can achieve $\underline{J} = 3$ at $\sigma = 0.005$ while only $\underline{J} = 1$ for large $\sigma$. This can also happen to robust methods at large $\varepsilon$, *e.g.*, when $\varepsilon = 1.2$, RadialRL on Freeway achieves $\underline{J} = 2$ at $\sigma = 0.75$ while only $\underline{J} = 1$ at $\sigma = 1.0$. Another case is when $\sigma$ is large enough such that further increasing $\sigma$ will not bring additional robustness, *e.g.*, in Figure 5, when $\varepsilon = 1.5$, RadialRL on Freeway achieves $\underline{J} = 1$ at $\sigma = 1.5$ while only $\underline{J} = 0$ at $\sigma = 4.0$.

### E.5 Computational Cost

Our experiments are conducted on GPU machines, including GeForce RTX 3090, GeForce RTX 2080 Ti, and GeForce RTX 1080 Ti. The running time of $m = 10000$ forward passes with sampled states for per-state smoothing ranges from 2 seconds to 9 seconds depending on the server load. Thus, for one experiment of CROP-LoAct with trajectory length $H = 500$ and 10 repeated runs, the running time ranges from 2.5 hours to 12.5 hours. CROP-GRe is a little more time-consuming than CROP-LoAct, but the running time is still in the same magnitude. For trajectory length $H = 500$ and sampling number $m = 10000$, most of our experiments finish within 3 hours. For CROP-LoRe specifically, the running time depends on the eventual size of the search tree, and therefore differs tremendously for different methods and different smoothing variances, ranging from a few hours to 4 to 5 days.

### E.6 Discussion on Evaluation Results of CartPole

We provide the evaluation results for CROP on CartPole, comparing nine RL algorithms: the six algorithms evaluated in our main paper (StdTrain, GaussAug, AdvTrain, SA-MDP (PGD), SA-MDP (CVX), and RadialRL), as well as three additional algorithms (CARRL, NoisyNet, and GradDQN). The detailed descriptions of these algorithms are provided in Appendix D.2. We present the evaluation results in Figure 8.

**Impact of Smoothing Parameter $\sigma$.**  In CartPole game, we draw similar conclusions regarding the impact of smoothing variance from the results given by CROP-LoAct, CROP-GRe and CROP-

LORE. For small $\sigma$, different methods are indistinguishable. As $\sigma$ increases, CARRL demonstrates its advantage as $\sigma$ increases, but AdvTrain and GradDQN tolerate large noise. One of differences when evaluating with CROP-GRE and CROP-LORE is that RadialRL can hold a high performance and even sometimes can be better than CARRL. Another one is that the lower bound for GradDQN is lower than all other methods in CROP-LORE.

**Tightness of the certification $\underline{J_E}$, $\underline{J_p}$ and $\underline{J}$.** In CartPole, we also compare the empirical cumulative rewards achieved under PGD attacks with our certified lower bounds $\underline{J_E}$, $\underline{J_p}$ and $\underline{J}$. Demonstrated by Figure 8c, first of all, the correctness of our bounds is validated because the empirical results are consistently lower bounded by our certifications. Compared with the loose expectation bound $\underline{J_E}$, the improved percentile bound $\underline{J_p}$ is much tighter. Compared with these two methods, the absolute lower bound $\underline{J}$ is even tighter, especially noticing the zero gap between the certification and the empirical result when the attack magnitude is not too large. Finally, methods with the same empirical results under attack may achieve very different certified lower bounds $\underline{J_E}$, $\underline{J_p}$ and $\underline{J}$, showing the importance of certification.

**Environment Properties.** Different from Pong and Freeway, CartPole is an environment with low dimension *states*, in contrast to the high dimensional Atari games evaluated in Section 6. This gives rise to several outcomes. First, StdTrain can tolerate noise well in CartPole. Second, CARRL demonstrates its advantage in low dimensional game, which is consistent with the empirical observations in Everett et al. (2021).

### E.7  DISCUSSION ON EVALUATION RESULTS OF HIGHWAY

We next present the evaluation results for CROP on an autonomous driving environment Highway, whose state dimension is larger than CartPole but smaller than Atari games. We compare nine RL algorithms as introduced previously. We present the evaluation results in Figure 9.

**Impact of Smoothing Parameter $\sigma$.** In highway, CARRL demonstrates its advantage when $\sigma$ is small, which is supported by results in all three certifications CROP-LOACT, CROP-GRE, and CROP-LORE. For action certification (*i.e.*, CROP-LOACT), SA-MDP (PGD) can tolerate a large range of $\sigma$, meaning that the action selection of model trained with SA-MDP (PGD) algorithm is more consistent than other methods. For reward certification (*i.e.*, CROP-GRE and CROP-LORE), we derive the conclusion that RadialRL outperforms than other methods when $\sigma$ is large.

**Tightness of the certification $\underline{J_E}$, $\underline{J_p}$ and $\underline{J}$.** In highway, we compare the empirical cumulative rewards achieved under PGD attacks with our certified lower bounds $\underline{J_E}$, $\underline{J_p}$, and $\underline{J}$. The correctness of our bounds is validated because the empirical results are consistently lower bounded by our certifications, as shown in Figure 9c. As for the tightness of reward certification methods, we draw similar conclusion as other environments—$\underline{J}$ is the most tight, followed by $\underline{J_p}$; and $\underline{J_E}$ is much looser than $\underline{J_p}$ and $\underline{J_E}$. We also notice the non-negligible gap between certified lower bounds and the empirical rewards demonstrated by RadialRL and CARRL under large smoothing parameters. This may imply that there exists large scopes for improving the attack method, or there is potential to further tighten the certified lower bounds.

## F  A BROADER DISCUSSION ON RELATED WORK

### F.1  EVASION ATTACKS IN RL

We consider the adversarial attacks on state observations, where the attacker aims to add perturbations to the state during test time, so as to achieve certain **adversarial goals**, such as misleading the *action* selection and minimizing the *cumulative reward*. We discuss the related works that fall into these two categories below.

**Misleading the Action Selection.** It has been shown that different methods—applying *random noise* to simply interfere with the action selection (Kos & Song, 2017), or adopting *adversarial attacks* (*e.g.*, FGSM (Goodfellow et al., 2014) and CW attacks (Carlini & Wagner, 2017)) to deliberately alter the probability of action selection—are effective to different degrees. More concretely, when manipulating the action selection probability, some works aim to reduce the probability of selecting

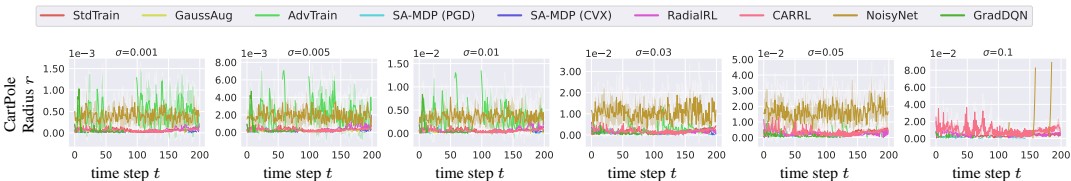

(a) Robustness certification for *per-state action* in terms of certified radius $r$ at all time steps. The shaded area represents the standard deviation.

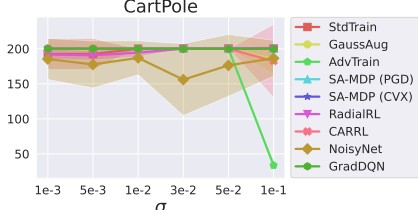

(b) Benign performance of locally smoothed policy $\tilde{\pi}$ under different smoothing variance $\sigma$.

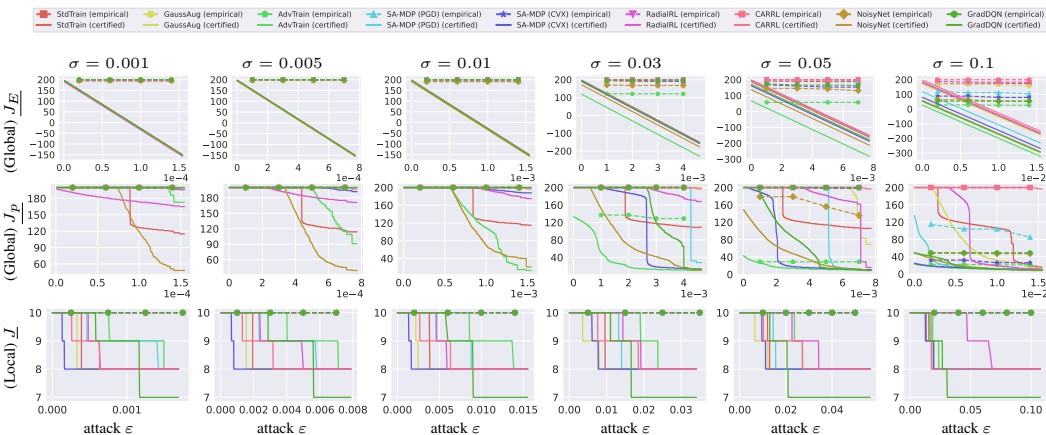

(c) Robustness certification as cumulative reward, including *expectation bound* $\underline{J_E}$, *percentile bound* $\underline{J_p}$ ($p = 50\%$), and *absolute lower bound* $\underline{J}$. Solid lines represent the certified reward bounds of different methods, and dashed lines show the empirical performance under PGD attacks.

Figure 8: **Robustness certification on CartPole** in terms of (a-b): *robustness of per-state action* and (c): *lower bound of cumulative rewards*. In (a) and (c), each column corresponds to one smoothing variance.

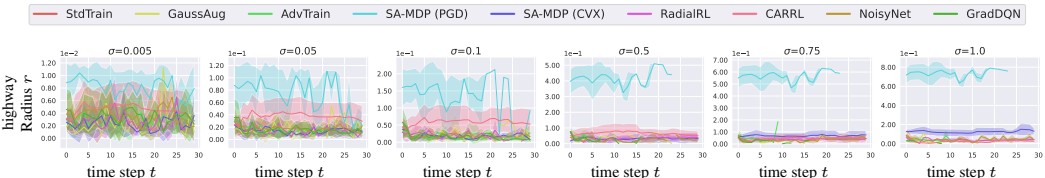

(a) Robustness certification for *per-state action* in terms of certified radius $r$ at all time steps. The shaded area represents the standard deviation.

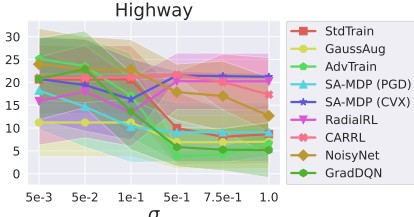

(b) Benign performance of locally smoothed policy $\tilde{\pi}$ under different smoothing variance $\sigma$.

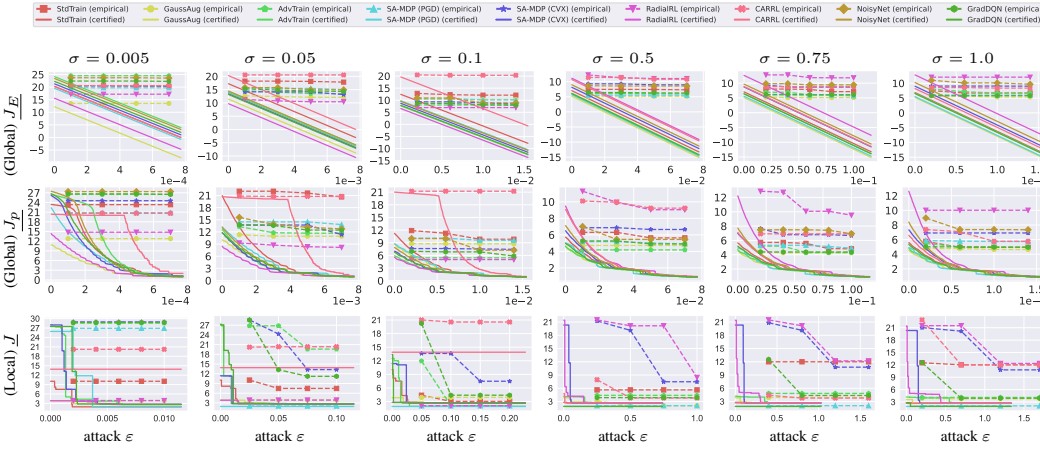

(c) Robustness certification as cumulative reward, including *expectation bound* $\underline{J_E}$, *percentile bound* $\underline{J_p}$ ($p = 50\%$), and *absolute lower bound* $\underline{J}$. Solid lines represent the certified reward bounds of different methods, and dashed lines show the empirical performance under PGD attacks.

Figure 9: **Robustness certification on highway** in terms of (a-b): *robustness of per-state action* and (c): *lower bound of cumulative rewards*. In (a) and (c), each column corresponds to one smoothing variance.

the optimal action (Huang et al., 2017; Kos & Song, 2017; Behzadan & Munir, 2017b), while others try to increase the probability of selecting the worst action (Lin et al., 2017; Pattanaik et al., 2018).

**Minimizing the Cumulative Reward.** ATLA (Zhang et al., 2021) and PA-AD (Sun et al., 2021) consider an *optimal adversary* under the SA-MDP framework (Zhang et al., 2020), which aims to lead to minimal value functions under bounded state perturbations. To find this *optimal adversary* (*i.e.*, the optimal adversarial state perturbation), ATLA (Zhang et al., 2021) proposes to train an adversary whose action space is the perturbation set in the state space, while PA-AD (Sun et al., 2021) further decouples the problem of finding state perturbations into finding policy perturbations plus finding the state that achieves the lowest value policy, thus addressing the challenge of large state space.

## F.2 ROBUST RL

**Distributionally Robust RL.** Nilim & El Ghaoui (2005) and Iyengar (2005) consider the problem of distributionally robust RL, where there is normally an uncertainty set with prior information regarding the distribution of the uncertain parameters. Iyengar (2005) directly put constraints on environmental dynamics and reward functions by assuming that they take values in an uncertainty set. Another line of research assumes the environmental dynamics and reward function as the uncertain parameters and are sampled from an uncertain distribution in the uncertain set. The uncertainty set is in general state-wise independent, *i.e.*, "s-rectangularity" in Nilim & El Ghaoui (2005). Both types aim to derive a policy by solving a max-min problem, with the former taking the minimum directly over the parameters, while the latter taking the minimum over the distribution.

**Empirically Robust RL against Evasion Attacks.** We have briefly introduced related work on empirically robust RL in Section 7 in the main paper. We emphasize two main differences between the contributions of these works and our CROP. **First**, most of these robust RL methods only provide empirical robustness against perturbed state inputs during test time, but cannot provide theoretical guarantees for the performance of the trained models under any bounded perturbations; while our CROP framework can provide practically computable *certified robustness* w.r.t. two robustness certification criteria (per-state action stability and cumulative reward lower bound), *i.e.*, for a given RL algorithm, we can compute certifications for it that indicate its robustness. (We will discuss a few more related work that can provide certified robustness in Appendix F.3.) **Second**, most of these Robust RL methods focus on only the per-state decision, while we additionally consider the *certification of trajectories* to obtain the lower bound of cumulative rewards. Below, we provide a more comprehensive review of the categories of RL methods that demonstrate empirical robustness against evasion attacks.

*Randomization methods* (Tobin et al., 2017; Akkaya et al., 2019) were first proposed to encourage exploration. This type of method was later systematically studied for its potential to improve model robustness. NoisyNet (Fortunato et al., 2017) adds parametric noise to the network's weight during training, providing better resilience to both training-time and test-time attacks (Behzadan & Munir, 2017b; 2018), also reducing the transferability of adversarial examples, and enabling quicker recovery with fewer number of transitions during phase transition.

Under the *adversarial training* framework, Kos & Song (2017) and Behzadan & Munir (2017b) show that re-training with random noise and FGSM perturbations increases the resilience against adversarial examples. Pattanaik et al. (2018) leverage attacks using an engineered loss function specifically designed for RL to significant increase the robustness to parameter variations. RS-DQN (Fischer et al., 2019) is an *imitation learning* based approach that trains a robust student-DQN in parallel with a standard DQN in order to incorporate the constrains such as SOTA adversarial defenses (Madry et al., 2017; Mirman et al., 2018).

SA-DQN (Zhang et al., 2020) is a *regularization* based method that adds regularizers to the training loss function to encourage the top-1 action to stay unchanged under perturbation.

Built on top of the *neural network verification* algorithms (Gowal et al., 2018; Weng et al., 2018), Radial-RL (Oikarinen et al., 2020) proposes to minimize an adversarial loss function that incorporates the upper bound of the perturbed loss, computed using certified bounds from verification algorithms. CARRL (Everett et al., 2021) aims to compute the lower bounds of action-values under potential perturbation and select actions according to the worst-case bound, but it relies on linear bounds (Weng et al., 2018) and is only suitable for low-dimensional environments.

### F.3 ROBUSTNESS CERTIFICATION FOR RL

Despite the abundant literature in robustness certification in supervised learning, there is almost no work on robustness certification for RL, given the unclear criteria and the intrinsic difficulty of the task. In this part, we first introduce another two works that can achieve robustness certification at state-level, and then another concurrent work Kumar et al. (2021) which also aims to provide provable robustness regarding the cumulative reward for RL.

**Robustness Certification on State Level.** Fischer et al. (2019) and Zhang et al. (2020) have discussed the state level robustness certification as well, but the way they obtain the certification is different from our CROP. Concretely, we achieve robustness by probabilistic certification via randomized smoothing (Cohen et al., 2019), while Fischer et al. (2019) and Zhang et al. (2020) directly achieve robustness by deterministic certification via leveraging the neural network verification techniques (Zhang et al., 2018; Xu et al., 2020; Mirman et al., 2018).

Despite these works on state-level robustness certification for RL, we are the *first* to provide the certification of cumulative rewards. We emphasize that the certification of lower bound of the cumulative reward in RL is more challenging than the per-state certification considering the dynamic nature of RL. Our main contribution of the paper indeed mainly focuses on providing an efficient adaptive search based algorithm CROP-LoRE to certify the lower bound of the cumulative reward together with rigorous analysis of the certification.

**Robustness Certification for Cumulative Rewards.** We compare our robustness certification for cumulative rewards with another concurrent work Kumar et al. (2021).

In terms of the *threat model*, both Kumar et al. (2021) and CROP consider the type of adversary that can perturb the state observations of the agent during test time. In our CROP, we consider a perturbation budget per time step following previous works (Huang et al., 2017; Behzadan & Munir, 2017b; Kos & Song, 2017; Pattanaik et al., 2018; Zhang et al., 2020), while they assume a budget for the entire episode. The two perspectives are closely related.

Regarding the *certification criteria*, we certify both *per-state action stability* and *cumulative reward lower bound*. Although they also consider the cumulative reward in their certification goal, they formulate the certification as a classification problem—certifying whether the cumulative reward is above a threshold or not, in contrast to directly certifying the lower bound as in our case.

For the *certification technique*, both works are developed based on randomized smoothing proposed in supervised learning (Cohen et al., 2019). We propose a global smoothing technique (CROP-GRE), as well as an adaptive search algorithm coupled with local smoothing (CROP-LoRE) to achieve the certification of cumulative reward; while they propose an adaptive version of the Neyman-Pearson lemma, which is similar with our global smoothing. Among the three algorithms (CROP-GRE, CROP-LoRE, and Kumar et al. (2020)), CROP-GRE cannot defend against an adaptive adversary (as concretely stated in Appendix C) while the other two can; specifically, CROP-LoRE is much more sophisticated and tight than the global smoothing ones, which is an important contribution in our work.

