# OpenReview forum: "CROP: Certifying Robust Policies for Reinforcement Learning through Functional Smoothing"
_ICLR.cc/2022/Conference — ICLR 2022 Poster_

### Official Review · Reviewer_PEk6 · 2021-10-26

**Correctness:** 3
**Technical Novelty And Significance:** 3
**Empirical Novelty And Significance:** 3
**Recommendation:** 6
**Confidence:** 2

**Main Review:**

I believe works that certifies robust behaviours of a trained RL policy under adversarial perturbation to be helpful, and the ability to validate different RL-algorithms as more robust or less robust w.r.t. these perturbations (in the experimental section) is important and compelling. Overall I find the premise of the paper solid, and its empirical evidence well documented and solid.

I cannot vouch for the correctness of the formal proofs, as I am not too good at analysis. However, I would like to ask a few questions:

1) modeling of transition function perturbations : how would this work adapt to the setting where the adversary changes the transition function itself, rather than merely applying a noise on the current observation? for instance, imagine we train a maze-world agent on slip-free environment, and want to certify some of its properties in real-life where the world might be slippery?

2) modeling structured perturbations : perturbation does not occur uniformly at random, typically there is a generative procedure which applies perturbation under some kind of distribution, allowing one to give tighter bounds if this generative procedure is modeled. for instance, in pong one might randomly generate hallucinated balls around the screen, or to fake multiple copies of the enemy paddle. how would the current approach extend to these structured perturbations?

**Summary Of The Paper:**

This paper gives certification of RL models under worse-case state perturbations in two respects: 1) behaving the same way w.r.t. state-action pairs and 2) achieving similar rewards at the end of a sampled trajectory. The paper gives theoretical grounding on the proposed certifications and demonstrate empirically the certifications are sound.

**Summary Of The Review:**

overall I am convinced this paper is solid based on its problem statement and empirical evidence, but I will defer to someone who can formally verify the math better.

---

> ### Author Response · Authors · 2021-11-22
> **Official Response to Reviewer PEk6**
>
> We thank the reviewer for the insightful comments and provide responses below.
>
> > 1. Modeling of transition function perturbations
>
> We thank the reviewer for pointing out the interesting setting, which is a different threat model considered in our paper. Our CROP framework considers the threat model of test-time perturbations on state observations which has been well studied in multiple empirically robust RL papers [1,2,3,4,5]. The threat model proposed by the reviewer can be subsumed to the category of perturbing the environmental dynamics. Although there are a few empirical works on the attacks [6] or robustness [7,8,9] in this scenario, there is currently no such work that provides robustness certification against this type of adversary. Our CROP framework cannot be directly adapted to this setting, but we can also adopt the intuition from randomized smoothing [10] to devise smoothing methods to smooth the transition function, and it would be interesting future work. Thanks for the suggestion!
>
> > 2. Modeling structured perturbations
>
> We thank the reviewer for the insightful comment. We clarify that the goal of our work is to provide a genetic robustness certification against *any* potential adversary that perturbs the state observations, rather than certain types of known adversaries. Indeed, more knowledge on the adversary structure or the game structure would allow a tighter certification, since the knowledge introduces more information that leads to simplifications of the original problem. For example, [11] explicitly considers such an adversary for Pong that shifts the ball downward a few pixels within the allowed magnitude (their Section V.C). Leveraging the knowledge, they devise a more targeted defense strategy by enumerating the finite set of the perturbed outcome and finding the worst one to defend against. Similarly, the knowledge and the corresponding method to leverage the knowledge can be incorporated into our certification framework as a plug-in module. To summarize, we propose a framework that provides generic robustness certification under minimal knowledge/assumption of the adversary/game structure, and our framework would readily enable a tighter certification by incorporating methods that specifically leverage the additional knowledge, which would lead to tighter certification results and therefore very interesting future work.
>
>
> **References**
>
> [1] Huang, Sandy, et al. "Adversarial attacks on neural network policies." arXiv preprint arXiv:1702.02284 (2017).
>
> [2] Behzadan, Vahid, and Arslan Munir. "Whatever does not kill deep reinforcement learning, makes it stronger." arXiv preprint arXiv:1712.09344 (2017).
>
> [3] Kos, Jernej, and Dawn Song. "Delving into adversarial attacks on deep policies." arXiv preprint arXiv:1705.06452 (2017).
>
> [4] Pattanaik, Anay, et al. "Robust Deep Reinforcement Learning with Adversarial Attacks." Proceedings of the 17th International Conference on Autonomous Agents and MultiAgent Systems. 2018.
>
> [5] Zhang, Huan, et al. "Robust Deep Reinforcement Learning against Adversarial Perturbations on State Observations." Advances in Neural Information Processing Systems 33 (2020): 21024-21037.
>
> [6] Rakhsha, Amin, et al. "Policy teaching in reinforcement learning via environment poisoning attacks." Journal of Machine Learning Research 22.210 (2021): 1-45.
>
> [7] Rajeswaran, Aravind, et al. "Epopt: Learning robust neural network policies using model ensembles." International Conference on Learning Representations. 2017.
>
> [8] Mandlekar, Ajay, et al. "Adversarially robust policy learning: Active construction of physically-plausible perturbations." 2017 IEEE/RSJ International Conference on Intelligent Robots and Systems (IROS). IEEE, 2017.
>
> [9] Pinto, Lerrel, et al. "Robust adversarial reinforcement learning." International Conference on Machine Learning. PMLR, 2017.
>
> [10] Cohen, Jeremy, Elan Rosenfeld, and Zico Kolter. "Certified adversarial robustness via randomized smoothing." International Conference on Machine Learning. PMLR, 2019.
>
> [11] Everett, Michael, Björn Lütjens, and Jonathan P. How. "Certifiable Robustness to Adversarial State Uncertainty in Deep Reinforcement Learning." IEEE Transactions on Neural Networks and Learning Systems (2021).

---

> > ### Comment · Reviewer_PEk6 · 2021-11-22
> > **thanks! my questions have been answered**
> >
> > I think the paper would benefit (maybe for a future revision as the other reviews do not look too promising at this point) from a stand point of "we choose this problem because it is on first principle a good problem" rather than relying/deferring to prior accepted works too heavily in a way of justification.
> >
> > I also quite agree with R1's comment of "the problem isn't verifying if a policy is robust, but to synthesize a policy that is robust by construction". Because RL policies are automatically trained, then it really makes a lot of sense to automatically come up with something that is robust.
> >
> > Again, I do not have too much expertise in this specific area, so take my suggestions with a grain of salt, and I hope to see a more structured form of adversarial attacks in the future, as making some assumptions (about structure) would allow you to perform richer inferences, and it is entirely reasonable that these kind of human interventions (by proving a grounding of these structures) are common in practice.

---

### Official Review · Reviewer_tzpV · 2021-11-01

**Correctness:** 4
**Technical Novelty And Significance:** 3
**Empirical Novelty And Significance:** 3
**Recommendation:** 6
**Confidence:** 4

**Details Of Ethics Concerns:**

I do not have ethics concerns.

**Main Review:**

Strengths:

1. This paper is well-written. The formulations are clear and easy-to-follow. The algorithm and experiment settings are discussed in detail in Appendix.
2. The certification makes intuitive sense, and seems to be correct, although I have not checked the detailed proof.
3. Both per-state action certification and cumulative reward certification are provided, while most related works do not discuss the reward certification.
4. The experimental results are interesting. The author provide a leaderboard that shows the comparison of many robust RL algorithms in different settings, which could be helpful for the evaluation of robust RL methods.


Weaknesses:

1. The certifications are straightforward application of the randomized smoothing certification used in supervised learning, which is not surprisingly novel. But different from supervised learning, the certified radii in this paper involve $V_{min}, V_{max}$ and $J_{min}, J_{max}$ which can be hard to obtain in practice, and may make the bound vacuous. The authors have provided methods to address the challenge of estimating these terms, but it also seems to be expensive.
2. The certifications on cumulative reward seem to be a little naive, and too expensive. The global smoothing method estimate the reward of many smoothed trajectories. But the noise sequence $\zeta$ is of dimension $H\times N$. As a result, it is hard to accurately estimate $\tilde{F}$. For the local smoothing method, an exhaustive search is needed, which could require exponential sample and computation cost.
3. For the cumulative reward, the "certification" is estimated by sampling methods. But as point 2 above states, it is sampling a very high-dimensional random vector, which may result in very inaccurate estimations. Then the certification can not serve as a strict lower bound of the cumulative reward. It is possible that a strong attack (e.g. [1]) can reduce the cumulative reward to be lower than the estimated certification. But this paper only tests the reward under a simple PGD attack.
4. Some related works are missing. This paper claims to be the first certification work for RL, but another paper[2] also proposes a smoothing-based certification for cumulative rewards. Although [2] is a very recent paper and can be regarded as concurrent work, I would suggest the authors to mention this paper and discuss the relation and differences between two papers.
5. Pong and Freeway are relatively simple compared to other Atari games. And the reward scales in these two games are small, thus it is hard to justify the tightness of the bound. In many literature[3], BankHeist and RoadRunner are used in addition to Pong and Freeway. So I would be happy to see more experimental results in these two games in order to better evaluate the proposed method.

Minor issues that do not affect my score:
1. The proposed certification only works for a greedy policy based on a Q-network. It will be better if the authors can provide some discussions on policy-based methods such as PPO.
2. This paper focuses on certifying an existing policy, and does not provide a method for adversarial training.


Refs:

[1] Sun et al. "Who Is the Strongest Enemy? Towards Optimal and Efficient Evasion Attacks in Deep RL."

[2] Kumar et al. "Policy Smoothing for Provably Robust Reinforcement Learning."

[3] Zhang et al. "Robust deep reinforcement learning against adversarial perturbations on state observations."

**Summary Of The Paper:**

This paper proposes two robustness certification criteria for Q-learning based RL policies under adversarial state perturbations: per-state action robustness and lower bound of cumulative rewards. The certification is mainly based on the randomized smoothing technique. For certifying the cumulative reward, two smoothing methods (global smoothing and local smoothing) are presented with different lower bound formulations. With the proposed smoothing methods, the authors empirically evaluate and compare 6 robust RL approaches in 3 environments, showing the applicability of the proposed method.

**Summary Of The Review:**

This paper focuses on an important problem of certifying RL policies against state perturbations. The proposed method makes intuitive sense, but as a direct application of the randomized smoothing technique, the novelty is a little limited. Although theoretical analysis of the lower bound of cumulative rewards is provided under 2 smoothing settings, the estimation of the lower bound could be either too loose or too expensive. The empirical evaluation is interesting and helpful for understanding recent robust RL works, but there is no enough evidence for the effectiveness of the proposed method (the tested environments are relatively simple). Therefore, I think this paper is around the borderline, and has the potential to be accepted. But at the current stage, I tend to reject it, unless more improvements are made (e.g. a more practical smoothing method, theoretical complexity analysis, or experiments in more complicated environments).


---
After rebuttal: the authors have provided more implementation details and experimental results, which I find interesting. So I increased my score.

---

> ### Author Response · Authors · 2021-11-22
> **Official Response to Reviewer tzpV (4/4)**
>
> **References**
>
> [1] Cohen, Jeremy, Elan Rosenfeld, and Zico Kolter. "Certified adversarial robustness via randomized smoothing." International Conference on Machine Learning. PMLR, 2019.
>
> [2] Sun, Yanchao, et al. "Who Is the Strongest Enemy? Towards Optimal and Efficient Evasion Attacks in Deep RL." arXiv preprint arXiv:2106.05087 (2021).
>
> [3] Kumar, Aounon, Alexander Levine, and Soheil Feizi. "Policy Smoothing for Provably Robust Reinforcement Learning." arXiv preprint arXiv:2106.11420 (2021).
>
> [4] Huang, Sandy, et al. "Adversarial attacks on neural network policies." arXiv preprint arXiv:1702.02284 (2017).
>
> [5] Behzadan, Vahid, and Arslan Munir. "Whatever does not kill deep reinforcement learning, makes it stronger." arXiv preprint arXiv:1712.09344 (2017).
>
> [6] Kos, Jernej, and Dawn Song. "Delving into adversarial attacks on deep policies." arXiv preprint arXiv:1705.06452 (2017).
>
> [7] Pattanaik, Anay, et al. "Robust Deep Reinforcement Learning with Adversarial Attacks." Proceedings of the 17th International Conference on Autonomous Agents and MultiAgent Systems. 2018.
>
> [8] Zhang, Huan, et al. "Robust Deep Reinforcement Learning against Adversarial Perturbations on State Observations." Advances in Neural Information Processing Systems 33 (2020): 21024-21037.
>
> [9] Li, Linyi, et al. "Sok: Certified robustness for deep neural networks." arXiv preprint arXiv:2009.04131 (2020).
>
> [10] ​​Salman, Hadi, et al. "Provably robust deep learning via adversarially trained smoothed classifiers." Proceedings of the 33rd International Conference on Neural Information Processing Systems. 2019.
>
> [11] Jeong, Jongheon, and Jinwoo Shin. "Consistency regularization for certified robustness of smoothed classifiers." arXiv preprint arXiv:2006.04062 (2020).

---

> ### Author Response · Authors · 2021-11-22
> **Official Response to Reviewer tzpV (3/4)**
>
> > 4. Missing related work Kumar et al. [3]
>
> We thank the reviewer for pointing out the interesting concurrent related work Kumar et al. [3]. We provide comparisons of their paper and ours below, and we have also updated the discussions into Appendix F.3 in our revision.
>
> In terms of the **threat model**, both works consider the type of adversary that can perturb the state observations of the agent during test time. In our work, we consider a perturbation budget per time step following previous works [4,5,6,7,8], while they assume a budget for the entire episode. The two perspectives are closely related.
>
> Regarding the **certification criteria**, we certify both *per-state action stability* and *cumulative reward lower bound*. Although they also consider the cumulative reward in their certification goal, they formulate the certification as a classification problem---certifying whether the cumulative reward is above a threshold or not, in contrast to directly certifying the lower bound as in our case.
>
> For the **certification technique**, both works are developed based on randomized smoothing proposed in supervised learning [1]. We propose a global smoothing technique (CROP-GRe), as well as an adaptive search algorithm coupled with local smoothing (CROP-LoRe) to achieve the certification of cumulative reward; while they propose an adaptive version of the Neyman-Pearson lemma, which is similar with our global smoothing. Note that we have analyzed in our paper and show that our local smoothing approach is much more sophisticated and tight than the global smoothing one.
>
> > 5. Evaluation on more complicated games.
>
> We thank the reviewer for the thoughtful suggestion and we provide more evaluations on a more complicated game Breakout following the suggestion, and we have provided the results in our revision Appendix E.7. Concretely, we present the *certified* cumulative reward lower bound computed via our CROP-LoRe algorithm in Figure 9, as well as the *empirical* cumulative reward obtained under practical attacks in Table 1.
>
> We obtain similar conclusions based on the new game and will describe the detailed observations as below. **First**, Figure 9 shows that GaussAug is the most certifiably robust, followed by AdvTrain; while StdTrain and RegPGD are shown to be non-robust for this game, whose lower bounds remain 0 for all smoothing parameters $\sigma$ we evaluate. **Second**, the robustness ranking given by our certification is aligned with the ranking under practical attacks. From Table 1, we see that GaussAug can achieve much higher cumulative reward under larger perturbation magnitudes $\varepsilon$ compared with other RL methods, which is consistent with the conclusion drawn from Figure 9. **Third**, our certification is relatively tight, especially for larger perturbation magnitudes. Notably, when $\sigma=0.05$, our lower bounds exactly match the attack results.
>
> > Minor
>
> **Discussions on policy-based methods such as PPO**
>
> Thanks for the helpful suggestion. We note that all our methods can be readily extended to policy-based methods. The intuition is that, instead of smoothing the value function in the Q-learning setting, we directly smooth the policy function in policy-based methods. With the smoothing module replaced and the theorems updated, other technical details in the algorithms for certifying the per-state action and the cumulative reward would then be similar. We have made this more clear in Appendix C in our revision.
>
> **Certifying an adversarial policy vs adversarial training**
>
> We thank the reviewer for the interesting question. We clarify that our goal is to propose a certification framework with robustness guarantees, rather than propose empirically robust RL algorithms. We point to [9] for a more detailed discussion on robustness certification and robust training, where the latter is related to or built on the former, and its main purpose is to enhance the robustness certification. We believe our certification will provide an in-depth understanding of the RL algorithms from the certified robustness perspective, which will inspire interesting future work on designing robust RL algorithms.

---

> ### Author Response · Authors · 2021-11-22
> **Official Response to Reviewer tzpV (2/4)**
>
> > 2. Naive and expensive certification for cumulative reward
>
> We thank the reviewer for the insightful comment and provide responses for CROP-GRe and CROP-LoRe separately below.
>
> **CROP-GRe**: We first clarify that our goal is to compute the *lower bound* of the expectation of $F_\pi$ or the $p$-th percentile of $F_\pi$. The estimation of $\widetilde F$ is only an intermediate step, and its *accuracy* does not influence the *soundness* of the lower bound calculation. The reason is given below. The number of sampled randomized trajectories $m$ controls the trade-off between the estimation *efficiency* and *accuracy*, as well as the *tightness* of the derived lower bound. Concretely, a small $m$ would provide high efficiency, low estimation accuracy, and loose lower bound; but the lower bound is always *sound*, since our algorithm explicitly accounts for the *inaccuracy* associated with $m$ via leveraging the Hoeffding’s inequality (Algorithm 2 line 9, Appendix B.2) in the case of Expectation bound (or the binomial formula in the case of Percentile bound). We acknowledge that the high dimension of noise will inevitably lead to a loose bound, so we further propose CROP-LoRe which leverages the adaptive search to obtain a tighter bound. We have updated the discussions in Appendix B.2 in our revision.
>
> **CROP-LoRe**: The reviewer is correct that CROP-LoRe performs an exhaustive search which has high computation cost, but in practical implementation, we adopt several optimization techniques (such as pruning and memorization, as described in Section B.3), which largely reduces the potential search space and saves unnecessary computation. Thus, the search is practically feasible. Furthermore, with more specific knowledge of the game mechanisms, the search algorithm can be further optimized. (Take the Pong game as an example, given the horizontal speed of the ball, we can compress the time steps where the ball is flying between the two paddles, thus reducing the computation.) We additionally emphasize that CROP is the first framework that achieves the robustness certification for RL algorithms, and we hope our framework can inspire more empirically robust RL algorithms, as well as more efficient certification algorithms that achieve tighter bounds.
>
> In terms of the time complexity of CROP-LoRe, it is $O(H |S_{\mathrm{explored}}| \times (\log |S_{\mathrm{explored}}| + |A| T))$, where $|S_{\mathrm{explored}}|$ is the number of explored states throughout the search procedure, which is no larger than cardinality of state set, $H$ is the horizon length, $|A|$ is the cardinality of action set, and $T$ is the time complexity of performing local smoothing. The main bottleneck of the algorithm is the large number of possible states, which is in the worst case exponential to state dimension. However, to provide a sound worst-case certification agnostic to game properties, exploring all possible states may be inevitable.
>
> We have involved the discussions above in Appendix B.3 in our revision.
>
> > 3. Soundness of the lower bound; evaluation under a strong attack Sun et al. [2]
>
> We thank the reviewer for the insightful question. Regarding the soundness of the lower bound, as explained in the previous question, our algorithm has explicitly accounted for the *inaccuracy* associated with $m$ via leveraging Hoeffding’s inequality. Thus, the obtained lower bound is sound (i.e., holds with confidence $1-\alpha$), and even a strong attack would not be able to lead to a lower cumulative reward.
>
> We further emphasize that the goal of this paper is to provide certification that comes with theoretical guarantees, and therefore following the standard certified robustness literatures [1,9,10,11] there is no need to evaluate the empirical attacks, since the certification always holds as long as the attacks satisfy certain conditions regardless of the actual attack algorithms. The attacks we evaluated in our paper were only for the demonstration purpose and in the hope of providing an example, and we have made this motivation more clear in Appendix D.3 in our revision. We thank the reviewer for pointing out the interesting concurrent work Sun et al. [2] which conducts a strong empirical attack, and have involved the paper in the broader discussions of the related work in Appendix F.1 in the revision.

---

> ### Author Response · Authors · 2021-11-22
> **Official Response to Reviewer tzpV (1/4)**
>
> We thank the reviewer for the valuable comments and suggestions, and we provide the corresponding answers as below.
>
> > 1. Novelty beyond randomized smoothing in supervised learning; estimation of $V_{\rm min}, V_{\rm max}, J_{\rm min}, J_{\rm max}$
>
> We thank the reviewer for the thoughtful comment. We first note that this is the first work that is able to provide the certification for RL. Although it is based on the randomized smoothing approach in supervised learning [1], we additionally address the challenges that are specific to RL (i.e., the two challenges discussed in Section 4.2).
>
> We next provide more concrete discussions on the estimation of the variables $V_{\rm min}, V_{\rm max}, J_{\rm min}, J_{\rm max}$, as well as the cost for estimating them.
>
> - $V_{\rm min}, V_{\rm max}$: Our estimate is obtained via sampling the trajectories and calculating the Q values associated with the state-action pairs along these trajectories. Since we perform clipping using the obtained bounds, the certification is sound. The cost for estimating $V_{\rm min}$ and $V_{\rm max}$ is not high given that they can be directly obtained via sampling as concretely described above. Furthermore, we can balance the trade-off between the accuracy and efficiency of the estimation, since for any configuration of $V_{\rm min}$ and $V_{\rm max}$, the certification will invariantly be sound (reason above).
>
> - $J_{\rm min}, J_{\rm max}$: We use the default values of  $J_{\rm min}, J_{\rm max}$ in the game specifications rather than estimate them, which are independent with the trained models. Thus there is no cost at all for the estimation. As mentioned under Theorem 2 in Section 5.1, using these default values may induce a loose bound in practice, so we further propose the percentile smoothing that aims to eliminate the dependency of our certification on $J_{\rm min}$ and $J_{\rm max}$, thus achieving a tighter bound.
>
> We have emphasized our contribution in Section 4.2 and provided more details w.r.t. the variables in Appendix B.1 and Appendix B.2 in our revision.

---

> > ### Comment · Reviewer_tzpV · 2021-11-29
> > **Thank you for the detailed clarification**
> >
> > After reading the response and other reviews, I think this paper has some good contributions. There are still some flaws, such as the efficiency of the computation. But I agree that there are intrinsic difficulties in certifying RL methods with uncertain environments. The additional empirical study is also appreciated. Based on the above reasons, I have increased my score.

---

### Official Review · Reviewer_zGtv · 2021-11-02

**Correctness:** 2
**Technical Novelty And Significance:** 3
**Empirical Novelty And Significance:** 2
**Recommendation:** 5
**Confidence:** 4

**Main Review:**

The paper provides a comprehensive study of the RL robustness in the per-state setting as well as a per-trace setting. I am delighted to see interesting work beyond per-state certification in the RL setting.
Furthermore, the mathematical parts are well written and can be followed easily.

However, I am wondering about several aspects of the submission, which I hope the authors can clarify:
- Can you justify the claim to be the "first framework for certifying the robustness of Q-learning algorithms", given works like [2,3] and others exist? ([3] is even discussed in the text.) Granted, with the exception of parallel work, this is the first paper to look at the certification of traces rather than just per-state decisions.
- Can you elaborate on the inference/prediction procedure for the agents in all CROP settings? The text outlines only the certification. Is it sufficient to invoke the model with a single sample of Gaussian Noise at each step in inference, or is a procedure like Predict in [1] used?
  As I understand (2), at least for CROP-LoAct this is the case.
- I am generally confused whether CROP aims to be a testing-framework or a certified defense. Mathematically, sections 3-5 present a certified defense (e.g. something that is a applied to a base model) to obtain certifiable guarantee.
  Yet, sentences like "... we find that (1) RegCVX, RegPGD and RadialRL achieve high certified robustness..." or "As the first work providing robustness certification for RL, we expect more RL algorithms and games will be certified under our framework in future work." make CROP look like a testing/certification framework. This compounds with the fact that inference is not discussed (see previous question), as it seems one can just run a RadialRL model without further considerations to obtain the robustness guarantees outlined here.
- Does CROP-LoRe account for multiple-testing? Since the individual local certificates only hold with confidence $1-\alpha$ (which according to E.4 you choose as 0.05), a CROP-Lore certificate invoking N local certificates holds with at most confidence $1-N\alpha$.
- Why does CROP-LoAct not rephrase picking the action as a classification problem and directly apply the approach from [1], thus sidestepping the challenges outlined in 4.2? Is it because you rely on this form of certificate in CROP-LoRe?
- Does CROP-LoRe need access to an oracle transition function $\Gamma(s, a)$ which can execute the transition for a given $s$? If this is the case this seems a departure for the standard RL-setting where the environment can only be queried in a fixed temporal order. While I think this is perfectly fine for the considered purpose, such a departure from standard should be clearly indicated.
- Does Lemma 2 hold if the adversary can arbitrarily choose $\zeta_{t}$ based on the smoothed actions at steps $0, \dots, t$? It seems to me the presented Lemma 2 only holds if the attacker commits to the values of $\zeta_{t}$ beforehand.

While I like the high-level approach and I think its a valuable area of investigation, I am not convinced by the presentation and potentially technical correctness (see questions) of the current submission. Further, it appears that recently a pre-print [4] was introduced, which concerns a similar topic. Could you briefly comment on the difference in approach.

Minor:
- There is a hyperlink (from a citation) crossing a page boundary at the end of the first page, which leads unpleasant artifacts.
- The symbol used for smoothing noise seems to change between $\zeta$ and $\Delta$. If this is deliberate, I may have missed why.

[1] Certified Adversarial Robustness via Randomized Smoothing, Cohen et al.; ICML 2019

[2] Online Robustness Training for Deep Reinforcement Learning, Fischer et al.; arXiv 2019

[3] Robust Deep Reinforcement Learning against Adversarial Perturbations on State Observations, Zhang et al., NeurIPS 2020

[4] Policy Smoothing For Provably Robust Reinforcement Learning, Kumar et al., arXiv 2021

**Summary Of The Paper:**

This paper introduces CROP, a framework for the certification of reinforcement learning (RL) agents against (adversarial) observation/state perturbations based on randomized smoothing (RS). Formally, certificates for individual per-state decision as well as cumulative scores are distinguished.
In order to obtain a per-state certificate the authors introduce CROP-LoAct, which applies RS to the individual decision of the agent on each state.
For cumulative-score certificates two approaches are considered:
First, CROP-GRe which applies RS to the return function of a whole trajectory and obtains a bound either via mean smoothing or percentile smoothing.
Second, CROP-LoRe, which combines per-state certificates for a whole trace to obtain a score certificate. Via tree-search a lower bound for a given start state and base policy can be provided.



**Summary Of The Review:**

A conceptually interesting approach to the verification of reinforcement learning in the per-state setting and beyond. However, the current manuscript leaves open questions about conceptual decisions and formal correctness.

---

> ### Author Response · Authors · 2021-11-22
> **Official Response to Reviewer zGtv (3/3)**
>
> > 9. Minor
>
> **unpleasant hyperlink**
>
> We thank the reviewer for pointing this out and have solved the problem in our revision.
>
> **both $\zeta$ and $\Delta$ for noise**
>
> We are sorry for the confusion. The reason for using different symbols is to distinguish the scenarios. Concretely, we used $\zeta$ to denote the sample of $1$ noise at each step in the randomized trajectory (in CROP-GRe), and $\Delta$ to denote the samples of $m$ noise at each step for smoothing the current step prediction (in CROP-LoAct and CROP-LoRe). Indeed, both $\zeta$ and $\Delta$ simply denote the noise sampled from a Gaussian distribution in essence, and we have updated the symbols in Section 5.1 in our revision to ensure consistency with Section 4.
>
> **References**
>
> [1] Cohen, Jeremy, Elan Rosenfeld, and Zico Kolter. "Certified adversarial robustness via randomized smoothing." International Conference on Machine Learning. PMLR, 2019.
>
> [2] Fischer, Marc, et al. "Online robustness training for deep reinforcement learning." arXiv preprint arXiv:1911.00887 (2019).
>
> [3] Zhang, Huan, et al. "Robust Deep Reinforcement Learning against Adversarial Perturbations on State Observations." NeurIPS. 2020.
>
> [4] Kumar, Aounon, Alexander Levine, and Soheil Feizi. "Policy Smoothing for Provably Robust Reinforcement Learning." arXiv preprint arXiv:2106.11420 (2021).
>
> [5] Li, Linyi, et al. "Sok: Certified robustness for deep neural networks." arXiv preprint arXiv:2009.04131 (2020).
>
> [6] Watkins, Christopher JCH, and Peter Dayan. "Q-learning." Machine learning 8.3-4 (1992): 279-292.
>
> [7] Mnih, Volodymyr, et al. "Playing atari with deep reinforcement learning." arXiv preprint arXiv:1312.5602 (2013).
>
> [8] Huang, Sandy, et al. "Adversarial attacks on neural network policies." arXiv preprint arXiv:1702.02284 (2017).
>
> [9] Behzadan, Vahid, and Arslan Munir. "Whatever does not kill deep reinforcement learning, makes it stronger." arXiv preprint arXiv:1712.09344 (2017).
>
> [10] Kos, Jernej, and Dawn Song. "Delving into adversarial attacks on deep policies." arXiv preprint arXiv:1705.06452 (2017).
>
> [11] Pattanaik, Anay, et al. "Robust Deep Reinforcement Learning with Adversarial Attacks." Proceedings of the 17th International Conference on Autonomous Agents and MultiAgent Systems. 2018.
>
> [12] Zhang, Huan, et al. "Robust Deep Reinforcement Learning against Adversarial Perturbations on State Observations." Advances in Neural Information Processing Systems 33 (2020): 21024-21037.

---

> > ### Comment · Reviewer_zGtv · 2021-11-22
> > **Follow-up**
> >
> > I thank the authors for their response. However, I still have some follow-up questions:
> >
> > *Regarding 1:*
> > The claim about [2,3] seems wrong.
> > See Table 3 and the last paragraph in [3].
> > While they do not provide certificates on the obtained reward, they do certify robustness certificates similar to CROP-LoACT. Thus statements like the first sentence of the abstract, still seem factually wrong.
> >
> > *Regarding 2:*
> > Thanks for the clarification.
> > Does that mean that in order to run a CROP-GRe or CORP-LoRe policy in practice, I would need multiple rollouts even when deployed? This seems to be a severe limitation for an RL agent.
> >
> > *Regarding 3:*
> > I agree that you build certified agents from differently trained base agents and evaluate them in regards to score and certificate. While I that ranking base-models made based on the performance of the smoothed model, I find the term "testing framework" quite misleading and as a reader found it hard to follow.
> >
> > *Regarding 4:*
> > Wouldn't $N$ be the number of times line 15 is invoked in Algorithm 3?
> > As each time you do this, you generate a certificate that hold with confidence $1-\alpha$.
> > The overall output of Algorithm 3 depends on all of these certificates to hold simultaneously, and thus with probability $1-min(N\alpha, 1)$ by the union bound.
> >
> > *Regarding 5:*
> > As you don't perform training I don't see why you need the absolute value of the $Q$ values.
> > You only need to know which action has the largest $Q$ value in each state. This should be a classification setting. Could you elaborate on your answer?
> >
> > *Regarding 6:*
> > Thanks for the clarification.
> >
> > *Regarding 7/8:*
> > Thanks for the clarification.
> > Wouldn't then your answer to question 7 be the main distinction between your work and [4]?
> >
> > *Regarding 9:*
> > Great! Thank you.
> >
> > Best Regards,
> > Reviewer zGtv

---

> > > ### Author Response · Authors · 2021-11-23
> > > **Response to the Follow-up of Reviewer zGtv (2/2)**
> > >
> > > > 5. Differences from the classification setting
> > >
> > > We thank the reviewer for the thoughtful question. The reviewer is correct that we only need the information regarding which action has the largest Q value in each state, but this is not exactly the classification setting where the network learns and outputs a probability distribution over the classes. In the Q-learning setup [8,9], the network learns the Q values for different actions which are real numbers (not the probability distribution). That is why we claim *we cannot directly apply the approach from [1] without further tweaking the procedure*---there exist non-negligible differences between the Q-learning setting and the classification setting during certification, and we provide our workaround in Sec. 4.2 to address the challenges. Another potential way to tweak the procedure is to additionally apply softmax on the Q values to obtain a probability distribution, then address the problem in the **classification** setting. It would be an interesting future attempt to also look into this approach and perform a comparison with our current solution.
> > >
> > > > 7/8. Failure against an adaptive attacker; comparison with Kumar et al. [4]
> > >
> > > Thank the reviewer for pointing this out. We acknowledge that this is indeed a main distinction between our work and Kumar et al. [4], and we have updated the discussions in Appendix F.3 in our revision accordingly. In addition, we note that our adaptive search algorithm CROP-LoRe does not have such limitation on the adaptive attacker as the global smoothing CROP-GRe does, and we show that CROP-LoRe is much more sophisticated and tight than the global smoothing as one of our main contributions.
> > >
> > > We hope our answers can address the reviewer’s concerns, and we are happy to provide more clarifications should the reviewer have further questions. Thank you for the insightful detailed questions and suggestions again!
> > >
> > >
> > >
> > > **References**
> > >
> > > [1] Cohen, Jeremy, Elan Rosenfeld, and Zico Kolter. "Certified adversarial robustness via randomized smoothing." International Conference on Machine Learning. PMLR, 2019.
> > >
> > > [2] Fischer, Marc, et al. "Online robustness training for deep reinforcement learning." arXiv preprint arXiv:1911.00887 (2019).
> > >
> > > [3] Zhang, Huan, et al. "Robust Deep Reinforcement Learning against Adversarial Perturbations on State Observations." NeurIPS. 2020.
> > >
> > > [4] Kumar, Aounon, Alexander Levine, and Soheil Feizi. "Policy Smoothing for Provably Robust Reinforcement Learning." arXiv preprint arXiv:2106.11420 (2021).
> > >
> > > [5] Zhang, Huan, et al. "Efficient Neural Network Robustness Certification with General Activation Functions." Advances in Neural Information Processing Systems 31 (2018): 4939-4948.
> > >
> > > [6] Xu, Kaidi, et al. "Automatic Perturbation Analysis on General Computational Graphs." arXiv preprint arXiv:2002.12920 (2020).
> > >
> > > [7] Mirman, Matthew, Timon Gehr, and Martin Vechev. "Differentiable abstract interpretation for provably robust neural networks." International Conference on Machine Learning. PMLR, 2018.
> > >
> > > [8] Watkins, Christopher JCH, and Peter Dayan. "Q-learning." Machine learning 8.3-4 (1992): 279-292.
> > >
> > > [9] Mnih, Volodymyr, et al. "Playing atari with deep reinforcement learning." arXiv preprint arXiv:1312.5602 (2013).
> > >
> > > [10] Tjeng, Vincent, Kai Y. Xiao, and Russ Tedrake. "Evaluating Robustness of Neural Networks with Mixed Integer Programming." International Conference on Learning Representations. 2018.
> > >
> > > [11] ​​Weng, Lily, et al. "Towards fast computation of certified robustness for relu networks." International Conference on Machine Learning. PMLR, 2018.
> > >
> > > [12] Singh, Gagandeep, et al. "An abstract domain for certifying neural networks." Proceedings of the ACM on Programming Languages 3.POPL (2019): 1-30.

---

> > > > ### Comment · Reviewer_zGtv · 2021-11-23
> > > > **Further follow-up**
> > > >
> > > > Thank you for your swift reply.
> > > > Again I have a few follow-up questions:
> > > >
> > > > > 2. Multiple rollouts for CROP-GRe and CROP-LoRe.
> > > >
> > > > As you point out in your previous reply, you apply randomized smoothing to create a provably robust policy from a base policy.
> > > > As test/certification time you sample to get the score bounds of your methods in order to evaluate them.
> > > >
> > > > However, usually to invoke the robust policy at inference time you will also need to perform some sampling there.
> > > > (This is essentially the difference between Certify and Predict in [1].)
> > > > This is what my original question 2 was asking about, my apologies if this was not clear. Could you clarify what how the inference procedure for the robust policies looks like at deployment time?
> > > >
> > > > The need data augmentation at inference time (in order to evaluate the right model), makes randomized smoothing fundamentally different than the verification setting in [10,11,12], which you have cited.
> > > >
> > > > > 4. Confidence of CROP-LoRE
> > > >
> > > > Why does this depend on the number of possibly attacked states?
> > > > On a high level your algorithm performs a statistical test with confidence $1-\alpha$ in every iteration of line 15.
> > > > The correctness of this algorithm relies on all tests being true, thus all tests must hold jointly.
> > > > Furthermore just because your Gaussian noise is independent you may still introduce a dependence via the inputs or policy, and independence would need to be mathematically shown.
> > > >
> > > > > 5. Differences from the classification setting
> > > >
> > > > I fully agree with your statements about learning.
> > > > However, when certifying a given policy couldn't you treat the Q-values as logits and *directly* apply [1]?
> > > >
> > > >
> > > >
> > > > Thanks for your quick replies.
> > > > I realize that you can't update the paper anymore (or at least can't very soon in the future) and don't expect this.=
> > > >
> > > > Best Regards,
> > > > Reviewer zGtv
> > > >
> > > > **References**
> > > > All references refer to those introduced by the authors in the above post

---

> > > > > ### Author Response · Authors · 2021-11-24
> > > > > **Response to the Further Follow-up of Reviewer zGtv**
> > > > >
> > > > > Thank the reviewer for the further follow-up! We provide the explanations below:
> > > > >
> > > > > > 2. Multiple rollouts for CROP-GRe and CROP-LoRe.
> > > > >
> > > > > We thank the reviewer for further clarifying the question! We are sorry about the previous confusion. We clarify that at deployment time, the inference procedure does not need *multiple rollouts*, and we provide detailed inference procedures below and will make it clear in the revision.
> > > > >
> > > > > **CROP-GRe**: At deployment time, we deploy the $\sigma$-randomized policy $\pi’$ as defined in Definition 4 in Section 5.1. That is, for each observed state $s_t$, we sample **one time** of Gaussian noise $\Delta_t$ to add to $s_t$, and take action according to this single randomized observation $a=\pi(s_t+\Delta_t)$. Thus, in deployment time, the $\sigma$-randomized policy $\pi’$ executes on **one** rollout - at each step it takes one observation and chooses one action (following the procedure described above).
> > > > >
> > > > > **CROP-LoRe**: At deployment time, We deploy the locally smoothed policy $\tilde \pi$ as defined in Equation (2) in Section 4.1. Concretely, for each observed state $s_t$, we sample a batch of Gaussian noise to add to $s_t$, and take action according to the computed mean Q value on the batch of randomized observations. Thus, in deployment time, the locally smoothed policy $\tilde\pi$ executes on **one** rollout - at each step it takes one observation and chooses one action (following the procedure described above).
> > > > >
> > > > > > 4. Confidence of CROP-LoRE
> > > > >
> > > > > *"Why does this depend on the number of possibly attacked states? On a high level your algorithm performs a statistical test with confidence $1-\alpha$ in every iteration of line 15. The correctness of this algorithm relies on all tests being true, thus all tests must hold jointly."*
> > > > >
> > > > > This is because for non-attacked states the sampling error does not exist as we show below:
> > > > > For the smoothed policy, suppose at current state $s$ which is not attacked, the action that will be taken is $a^*$ defined in line 10. Note that in line 15, for $a = a^*$, the condition can never be satisfied since $\tilde{Q}^\pi (s,a^*) = \tilde{Q}^\pi (s,a) < \tilde{Q}^\pi(s,a) + 2\Delta$. Thus, $r = 0$ and we will always include $a^*$ in $a\\_list$, i.e., deterministically we will explore the action that will be chosen for this step. Therefore, the sampling error does not exist for non-attacked states, and the reviewer is correct that $N$ could be the number of times line 15 is invoked in the worst case.
> > > > >
> > > > > *"Furthermore just because your Gaussian noise is independent you may still introduce a dependence via the inputs or policy, and independence would need to be mathematically shown."*
> > > > >
> > > > > Sorry for the confusion. Here we refer to the independence of random variables $\tilde{Q}^\pi(s,a)$ for different states $s$. Now we formally show such independence:
> > > > > Given two distinct states $s_i, s_j$, from line 9 of Algorithm 3, we have
> > > > > $$ \tilde{Q}^\pi(s_i,a) = \dfrac{1}{m} \sum_{i=1}^m \texttt{clip}(Q^\pi(s_i+\delta_i, a),\min=V_{\min}, \max=V_{\max}), $$ where the randomness only comes from Gaussian noise $\\{\delta\_i\\}\_{i=1}^m$.
> > > > > $$ \tilde{Q}^\pi(s_j,a) = \dfrac{1}{m} \sum_{j=1}^m \texttt{clip}(Q^\pi(s_j+\delta_j, a),\min=V_{\min}, \max=V_{\max}), $$ where the randomness only comes from Gaussian noise $\\{\delta\_j\\}\_{j=1}^m$.
> > > > > Since $\\{\delta\_i\\}\_{i=1}^m$ and $\\{\delta\_j\\}\_{j=1}^m$ are independently sampled, the random variables $\tilde{Q}^\pi(s_i,a)$ and $\tilde{Q}^\pi(s_j,a)$ are independent. Thus, the event $\| \tilde{Q}^\pi(s_i,a) - \mathbb{E} \tilde{Q}^\pi(s_i,a) \| \ge \Delta$ is dependent of the event $\| \tilde{Q}^\pi(s_j,a) - \mathbb{E} \tilde{Q}^\pi(s_j,a) \| \ge \Delta$. Same argument generalizes: for any distinct states, as long as $\tilde{Q}^\pi(s,a)$ is independently estimated, the random variables are independent.
> > > > >
> > > > > > 5. Difference from the classification setting
> > > > >
> > > > > We thank the reviewer’s follow-up question. The reviewer is correct: in our last response, we mentioned that “Another potential way to tweak the procedure is to additionally apply softmax on the Q values to obtain a probability distribution, then address the problem in the **classification** setting. It would be an interesting future attempt to also look into this approach and perform a comparison with our current solution.” In this case, the Q function can be modeled similarly to the classification setting. We hope this clarifies the reviewer’s concern, and we will also add this discussion to our revision later.

---

> > > > > > ### Comment · Reviewer_zGtv · 2021-11-24
> > > > > > **Response**
> > > > > >
> > > > > > Thank you very much for the clarification.
> > > > > > This indeed clears up most of my questions.
> > > > > >
> > > > > > I was very impressed by the speed and technical depth in which the authors addressed reviewer questions.
> > > > > > The discussion phase so far has strongly increased my understanding of the submission and the paper itself was strongly improved.
> > > > > > To reflect this I have updated my score.
> > > > > >
> > > > > > Despite this, I still believe that further presentation updates are required to prevent misunderstandings in the first place.
> > > > > > (As updates to the paper are not possible anymore, I want to point out that I am not sure whether such updates would fit within the context of the discussion phase.)
> > > > > >
> > > > > > To summarize, after my discussion I believe that the paper contains many interesting and correct technical contributions, which could be presented in a more detailed and appealing manner.

---

> > > > > > > ### Author Response · Authors · 2021-11-24
> > > > > > > **Thank you! We will keep improving.**
> > > > > > >
> > > > > > > We thank the reviewer for all the insightful discussion and valuable suggestions! We really appreciate it and will reflect it in our acknowledgement as well.
> > > > > > >
> > > > > > > We have updated our paper accordingly and will keep improving our paper based on the new comments and please let us know if the reviewer has further suggestions, and we really look forward to improving our paper based on the reviewer’s suggestions!

---

> > > ### Author Response · Authors · 2021-11-23
> > > **Response to the Follow-up of Reviewer zGtv (1/2)**
> > >
> > > We thank the reviewer for the timely feedback and provide our responses below.
> > >
> > > > 1. Robustness certification in [2,3]
> > >
> > > We thank the reviewer for the thoughtful question and kind reminder. We agree with the reviewer that our paper is the first work to provide the certification of traces (i.e., lower bound of the cumulative reward) rather than just per-state decisions, given that [2,3] have discussed the per-state certification as well. We have updated our revision in abstract, introduction, related work, and Appendix F.3 to make this clear. Thank you for the helpful suggestions.
> > >
> > > We also want to emphasize that the certification of lower bounds of the cumulative reward in RL is more challenging than the per-state certification considering the dynamic nature of RL. Our main contribution of the paper indeed mainly focuses on providing an efficient tree search based algorithm CROP-LoRe to certify the lower bound of the cumulative reward together with rigorous analysis of the certification. Concretely, we achieve robustness by probabilistic certification via randomized smoothing [1], while [2,3] directly achieve robustness by deterministic certification via leveraging the neural network verification techniques [5,6,7].
> > >
> > > > 2. Multiple rollouts for CROP-GRe and CROP-LoRe.
> > >
> > > We thank the reviewer for the question. The reviewer understands our procedures of CROP-GRe and CROP-LoRe correctly. Indeed, we need to run multiple rollouts during certification (i.e., $m$ randomized trajectories in the case of CROP-GRe), and explore different actions within the search tree for CROP-LoRe.
> > >
> > > However, we would like to clarify that our CROP-GRe and CROP-LoRe are **robustness certification algorithms** for a given deployed RL algorithm rather than a RL algorithm itself, and the certification process incurs its own computation but does not alter the time complexity of the deployed RL algorithm. In general, the robustness certification algorithms will often induce additional computational cost, according to the literature of robustness certification [10,11,12].
> > >
> > > > 3. Testing framework
> > >
> > > We thank the reviewer for the insightful discussion and sorry for the confusion. We have updated our revision to reflect that our certification framework focuses on a “smoothed” version of a given trained RL model. Specifically,  our certification framework is based on the randomized smoothing technique which is only capable of certifying a “smoothed” version of a given trained model. It would be a very challenging and interesting future work to directly certify a given RL model.
> > >
> > > > 4. Confidence for CROP-LoRe
> > >
> > > Thanks for the insightful question. Here $N$ is the maximum number of possible attacked states explored by CROP-LoRE rather than the number of times line 15 is involved (i.e., the total number of explored states), and thus the CROP-LoRe confidence is $(1-\alpha)^N$.
> > >
> > > Formally, $$\begin{aligned} \Pr[\text{CROP-LoRE certification holds}] & = \prod_{\text{all } s \text{ explored by CROP-LoRE}} \Pr[\text{not make error on }s | \text{not make error on }s_{pre}] \\\\ & \overset{\text{(i)}}{=} \prod_{\text{all attackable } s} (1-\alpha) \times \prod_{\text{all unattackable } s} 1 \\\\ & \overset{\text{(ii)}}{=} (1-\alpha)^N. \end{aligned}$$ In the above equation, we can see that (i) leverages the independence which in turn gives a bound of form $(1-\alpha)^N$. This is because in each step, the event of “certification does not hold” is **independent** since we sample Gaussian noise **independently**. We agree that union bound also gives confidence $1-\min(N\alpha,1)$, but leveraging the independence we can obtain a tighter confidence lower bound of $(1-\alpha)^N$.
> > > (ii) shows $N$ should be the number of possible attacked states. This is because for unattackable states, the certification deterministically holds since there is no attack at current step. Therefore, we only need to count the confidence intervals from all possibly attackable steps instead of “the number of times line 15 is involved”.
> > >
> > > We have updated the above discussions in Appendix C in our revision to help readers understand our algorithm better.

---

> ### Author Response · Authors · 2021-11-22
> **Official Response to Reviewer zGtv (2/3)**
>
> > 4. Multiple-testing: confidence $1-N\alpha$ in CROP-LoRE
>
> We thank the reviewer for the thoughtful question. We acknowledge that the reviewer is correct and we consider the independent multiple-test. Concretely, due to the independence of decision making errors, the confidence is $(1-\alpha)^N$ instead of $1-N\alpha$, where $N$ is the maximum number of possible attacked states explored by CROP-LoRE. We have made this explicit in Appendix C in revision. We remark that in practice, the attacker usually has the ability to perturb only a limited number of steps, i.e., $N$ is typically small. Therefore, the confidence is still non-trivial.
>
> > 5. Why does CROP-LoAct not rephrase the action selection as a classification problem?
>
> We thank the reviewer for the valuable comment and we provide the explanations as follows. In Q-learning [6,7], the Q value function is first learned, and then leveraged to perform action selection according to criteria such as the greedy action selection criterion. This procedure is unlike performing action classification directly, and the differences between the two are exactly the two challenges we outlined. Thus, we cannot directly apply the approach from [1] without further tweaking the procedure. The reviewer’s intuition is correct that CROP-LoRe shares the similar challenge with CROP-LoAct (and thus can be addressed using the same approach), while the key reason for this question is that the procedures are different.
>
> > 6. Access to an oracle function $\Gamma(s,a)$
>
> We are sorry for the confusion. We clarify that the algorithm only requires access to the environment such that it can obtain the reward and next state via interacting with the environment (i.e., taking action $a$ at state $s$ and obtain next action $s’$ and reward $r$), but does not require access to an oracle transition function. We further point out that access to the environment is already sufficient for conducting our adaptive search algorithm. Thus, our setting is exactly the same as the standard RL setting. We have update Appendix B.3 and Algorithm 3 in our revision and to make this more clear.
>
> > 7. Lemma 2 under adaptive attack
>
> We thank the reviewer for the thoughtful question. The reviewer is correct in that our Lemma 2 requires knowing the noise added to each step beforehand, so as to generate $m$ randomized trajectories given the known noise sequence {$\zeta_t$}$_{t=0}^{H-1}$. Thus it cannot be applied against an adaptive attacker, which is indeed a drawback of the CROP-GRe certification. We have added the discussion of this drawback to Appendix C in our revision, but we also note that our CROP-LoAct and CROP-LoRe can work fine against an adaptive attacker.
>
> > 8. Comparison with the related work Kumar et al. [4]
>
> We thank the reviewer for pointing out the interesting concurrent related work Kumar et al. [4]. We provide comparisons of their paper and ours below, and we have also updated the discussions into Appendix F.3 in our revision.
>
> In terms of the **threat model**, both works consider the type of adversary that can perturb the state observations of the agent during test time. In our work, we consider a perturbation budget per time step following previous works [8,9,10,11,12], while they assume a budget for the entire episode. The two perspectives are closely related.
>
> Regarding the **certification criteria**, we certify both *per-state action stability* and *cumulative reward lower bound*. Although they also consider the cumulative reward in their certification goal, they formulate the certification as a classification problem---certifying whether the cumulative reward is above a threshold or not, in contrast to directly certifying the lower bound as in our case.
>
> For the **certification technique**, both works are developed based on randomized smoothing proposed in supervised learning [1]. We propose a global smoothing technique (CROP-GRe), as well as an adaptive search algorithm coupled with local smoothing (CROP-LoRe) to achieve the certification of cumulative reward; while they propose an adaptive version of the Neyman-Pearson lemma, which is similar with our global smoothing. Note that we have analyzed in our paper and show that our local smoothing approach is much more sophisticated and tight than the global smoothing one.

---

> ### Author Response · Authors · 2021-11-22
> **Official Response to Reviewer zGtv (1/3)**
>
> We thank the reviewer for the insightful questions and comments, and we provide responses below.
>
> > 1. Justification for “the first framework for certifying the robustness of Q-learning algorithms”
>
> We thank the reviewer for the question. We clarify that although there are concurrent works on robust RL like [2,3], they only provide *empirical robustness* which does not come with theoretical guarantees, i.e., they propose RL algorithms that are empirically shown to improve the robustness; while our CROP framework can provide practically computable *certified robustness* w.r.t. two robustness certification criteria (per-state action stability and cumulative reward lower bound), i.e., for a given RL algorithm, we can compute certifications for it that indicate its robustness. Thus, we believe that this is “the first framework for certifying the robustness of Q-learning algorithms”. We have provided the clarifications in Section 7 Related Work and have made the points more clear in Appendix F.2 in the revision.
>
> We are also delighted that the reviewer noticed the difference that beyond the per-state decisions considered in [2,3], we additionally consider the “certification of traces” to obtain the lower bound of cumulative rewards. We have included the discussions above in Appendix F.2 in the revision.
>
> > 2. Elaboration on the inference/prediction procedures
>
> We thank the reviewer for the thoughtful question. We next provide a more concrete description regarding how we conduct the inference/prediction in each of our CROP algorithms. These details as well as the algorithm boxes are all contained in our Appendix B, and we have updated our revision to make them more clear. Below we provide the pointers to the relevant sections.
>
> In CROP-LoAct, during inference, we invoke the model with $m$ samples of Gaussian noise at each time step, and use the averaged Q value on these $m$ noisy samples to obtain the greedy action selection, and accordingly compute the certification for it using Theorem 1. The reviewer is right that this procedure is similar to Predict in [1], but since RL involves multiple step decisions, we do not have the “abstain” decision as in Predict in [1], i.e., CROP-LoAct will take the greedy action at all steps no matter whether the action can be certified or not. We have included the details in Appendix B.1 in our revision.
>
> In CROP-GRe, instead of sampling $m$ noisy states per time step, we sample $m$ randomized trajectories. For each sampled randomized trajectory, at each time step in the trajectory, we invoke the model with $1$ sample of Gaussian noise. Given the $m$ randomized trajectories and the cumulative reward for each of them, we compute the certification using Theorem 2 and Theorem 3. We have included the details in Appendix B.2 in our revision.
>
> In CROP-LoRe, the inference procedure per step is exactly the same as in CROP-LoAct. We obtain the certification using the adaptive search algorithm and Theorem 4. We have included the details in Appendix B.3 in our revision.
>
> > 3. Is CROP a testing framework or a certified defense?
>
> We thank the reviewer for the valuable question. We clarify that CROP is a testing framework with certified robustness guarantees, rather than an empirical testing framework. In particular, our certification (sections 3-5) is obtained via adding sampled smoothing noise during the testing time given a trained policy without modifying it, and then providing certification for such a smoothed policy. Though the smoothed policy is not exactly the same as the base policy, the smoothing step aims to *enable the probabilistic certification* based on randomly sampled smoothing noises rather than *improving the robustness* of the base policy. We additionally point to a general survey paper for supervised learning [5] for a more detailed discussion on robustness certification and robust training, where the latter is related to or built on the former, and its main purpose is to enhance the robustness certification (similar to the certified defense mentioned by the reviewer). Based on this background, the purpose of our work is to propose a robustness certification method for existing empirically robust RL algorithms.
>
> Concretely, in this paper, we provide the certification results for several existing trained policies such as the vanilla trained policy (StdTrain), policy trained with Gaussian noise added during training time (GaussAug [1]), and policies trained with other empirically robust RL algorithms (e.g., RegCVX [2] and RadialRL [3]). We believe our certification of these algorithms on different games will provide an in-depth understanding of them from the certified robustness perspective, which will inspire interesting future work on designing robust RL algorithms.

---

### Official Review · Reviewer_ZBtu · 2021-11-02

**Correctness:** 3
**Technical Novelty And Significance:** 1
**Empirical Novelty And Significance:** 2
**Recommendation:** 3
**Confidence:** 3

**Main Review:**

*Originality*

This work essentially extends robustness certification methods existing in supervised learning, to reinforcement learning. Although the idea of certification is new, the essence of robustness guarantees together with function perturbation is not new in the RL literature. As such, some significant literature review on robust MDPs and robust RL is cruelly missing.

*Quality*

Essentially, the two proposed certificates are based on a state perturbation to which either the Q-function or the policy is applied. As far as I understand, this does not affect the model itself but only the ordering in either the Q-values or the policy. As a result, this discredits the certificate.
l. 47-53: the results obtained from CROP are problem-dependent. This may say something about the domain, but also about the certificate itself and how reliable it is. It is not clear which one of the two options applies.
CROP designs a certification for the robustness of other previous algorithms, but the very method is not challenged in and of itself. Instead, in Sec. 6, it is shown how robust are different methods but the reliability of CROP itself is not tested.

*Significance*

Overall, I find several problematic issues in this work. On a higher level, the fundamental RL question is not "how to certify robustness" but rather "how to attain robustness". Thus, I disagree with the leading direction of that paper and am not sure how relevant it would be for RL researchers. Also, a comprehensive RL review and robustness in RL is strongly missing.

*Clarity: Minor comments*

- l. 5 "smoothed with Gaussian noise": these are "perturbed" rather than "smoothed" ("smoothing" is used for making a function continuous or the like)
- l. 14 "RegPGD, RegCVX, RadialRL": those methods have never been described before, nor are they explained
- l. 120, 121, 131 "per-state action": you may just use "policy" instead
- l. 128 : Notation never introduced for those bounds. If these are reward bounds, then a  factor seems to be missing
- l. 113-114: I do not understand the justification for focusing on a finite horizon.
- l. 101 the maximum perturbation magnitude is uniform over the state-action space. Would not it be tiny then? How reliable would the certificate be if that radius gets close to 0? Also, a state-dependent  appears in Thm 1. This is confusing, or it should be discussed whether/when one would consider uniform VS state-dependent magnitude.
- Thm 1- Notation is confusing. It refers both to the immediate reward (l. 72) and a radius ball in Thm 1.

*English corrections*

- l. 1 "Certifying Robust Policies for reinforcement learning (CROP)" --> "Certifying Robust Policies (CROP) for reinforcement learning (RL)"
- l. 5 "which uses" --> "that uses/using"; l. 9 "which makes use" --> "that uses/using"
- l. 10 "for reward" --> "for the reward"/ "for performance"
- l. 15 "we demonstrate that ... by evaluating ..." --> "by evaluating ..., we demonstrate that ..." (revert)
- Remove "the" in l. 29, 30, 31, 38, 83, 398
- l. 25 "In particular, adversarial training ... by enforcing the smoothness of the trained models"--> "In particular, by enforcing the smoothness of the trained models, adversarial training... " (revert)
- l. 27-28 "have been proposed", "it is" - time concordance
- l. 31 "deterministic approaches" --> remove "approaches" (appears in l. 32)
- l. 32 "the lower bound" --> "a lower bound"
- l. 34 "compared with classification model" --> "compared to classification"
- l. 37 "Compared with" --> "Differently than"
- l. 37 "classification model" --> "classification"
- l. 37 "which only has one-step predictions" --> "which involves one-step prediction only"
- l. 37-38 "both ... as well as ..." --> "both ... and" or just "as well as"
- l. 39 remove "of RL"
- l. 48 "game properties" --> "the game properties"
- l. 84 "will be" --> "is"
- l. 94 "following" redundant
- l. 105, 399 "reinforcement learning" --> "RL" (use abbreviation consistently -- or never use it)
- l. 121 "to calculate the lower bound of maximum perturbation magnitude in Def 1" --> "to calculate a lower bound of the maximum perturbation magnitude from Def 1"
- l. 126 "the Gaussian distribution" --> "a Gaussian distribution"
- l. 140 "the tradeoff" --> "a tradeoff"
- l. 400-403 A 3-line section is a bit weird

*Missing references on robust RL (to name a few)*

[1] Iyengar, Garud N. "Robust dynamic programming." Mathematics of Operations Research 30.2 (2005): 257-280.

[2] Nilim, Arnab, and Laurent El Ghaoui. "Robust control of Markov decision processes with uncertain transition matrices." Operations Research 53.5 (2005): 780-798.

[3] Tessler, Chen, Yonathan Efroni, and Shie Mannor. "Action robust reinforcement learning and applications in continuous control." International Conference on Machine Learning. PMLR, 2019.

**Summary Of The Paper:**

This work proposes a certification (CROP) attesting that a policy is robust against adversarial state perturbation. It finds its inspiration in the supervised learning literature and extends it to the sequential decision-making setting. Experiments on two Atari games test the performance of CROP.

**Summary Of The Review:**

By construction, robust policies (i.e., those that solve max-min objective in robust MDPs) provide a robustness certificate that is more general than the two criteria provided in Def. 1 and 3. As such, I see this study as a specific case of prior work.

---

> ### Author Response · Authors · 2021-11-22
> **Official Response to Reviewer ZBtu (4/4)**
>
> **References**
>
> [1] Weng, Lily, et al. "Towards fast computation of certified robustness for relu networks." International Conference on Machine Learning. PMLR, 2018.
>
> [2] Gowal, Sven, et al. "Scalable verified training for provably robust image classification." Proceedings of the IEEE/CVF International Conference on Computer Vision. 2019.
>
> [3] Cohen, Jeremy, Elan Rosenfeld, and Zico Kolter. "Certified adversarial robustness via randomized smoothing." International Conference on Machine Learning. PMLR, 2019.
>
> [4] Zhang, Huan, et al. "Towards stable and efficient training of verifiably robust neural networks." arXiv preprint arXiv:1906.06316 (2019).
>
> [5] Everett, Michael, Björn Lütjens, and Jonathan P. How. "Certifiable Robustness to Adversarial State Uncertainty in Deep Reinforcement Learning." IEEE Transactions on Neural Networks and Learning Systems (2021).
>
> [6] Fortunato, Meire, et al. "Noisy networks for exploration." arXiv preprint arXiv:1706.10295 (2017).
>
> [7] Pattanaik, Anay, et al. "Robust Deep Reinforcement Learning with Adversarial Attacks." Proceedings of the 17th International Conference on Autonomous Agents and MultiAgent Systems. 2018.
>
> [8] ​​Oikarinen, Tuomas, Tsui-Wei Weng, and Luca Daniel. "Robust deep reinforcement learning through adversarial loss." arXiv preprint arXiv:2008.01976 (2020).
>
> [9] ​​Li, Linyi, et al. "Sok: Certified robustness for deep neural networks." arXiv preprint arXiv:2009.04131 (2020).
>
> [10] Behzadan, Vahid, and Arslan Munir. "Whatever does not kill deep reinforcement learning, makes it stronger." arXiv preprint arXiv:1712.09344 (2017).
>
> [11] Zhang, Huan, et al. "Robust Deep Reinforcement Learning against Adversarial Perturbations on State Observations." Advances in Neural Information Processing Systems 33 (2020): 21024-21037.
>
> [12] Iyengar, Garud N. "Robust dynamic programming." Mathematics of Operations Research 30.2 (2005): 257-280.
>
> [13] Nilim, Arnab, and Laurent El Ghaoui. "Robust control of Markov decision processes with uncertain transition matrices." Operations Research 53.5 (2005): 780-798.
>
> [14] Tessler, Chen, Yonathan Efroni, and Shie Mannor. "Action robust reinforcement learning and applications in continuous control." International Conference on Machine Learning. PMLR, 2019.

---

> ### Author Response · Authors · 2021-11-22
> **Official Response to Reviewer ZBtu (3/4)**
>
> > 4. Clarity.
>
> We thank the reviewer for the detailed editing suggestions and we have updated them accordingly in our revision. We will provide clarifications for the unclear parts below and provide pointers to our revision.
>
>
> **l. 14: Details of evaluated RL methods RegPGD, RegCVX, RadialRL, etc.**
>
> The introduction of these methods can be found in our experimental setup in Section 6, with the more detailed description deferred to Appendix D.2.
>
>
> **l. 128: $[V_{\rm min}, V_{\rm max}]$.**
>
> We are sorry about the confusion. We meant to use $[V_{\rm min}, V_{\rm max}]$ to define the range of the Q values as in our Lemma 1 and Thm 1. In particular, $V_{\rm min}$ denotes the minimum Q value of the *given* Q network over the state-action space, and $V_{\rm max}$ denotes the maximum Q value. They are not reward bounds. Rather, they are specifically associated with each trained model. In practical implementation, the values are obtained via computing the minimum and maximum value for the *given* trained Q network over a sampled set of state-action pairs. We have made this more clear in Appendix B.1 in our revision.
>
> **l. 113-114: finite horizon.**
>
> We clarify that our criteria, theorem, and algorithm are applicable to the infinite horizon case as well. The focus of a finite horizon in this paper is mainly for the evaluation purpose---we would need to compute the (certified and empirical) cumulative rewards from finite rollouts. We have made this more clear in Appendix D.3 in our revision.
>
> **l. 101: maximum perturbation magnitude.**
>
> We thank the reviewer for pointing this out, and we would clarify that the maximum perturbation magnitude is state-dependent (as shown in Theorem 1), rather than uniform, so for some states the certified radius could be large (not necessarily to be tiny) as shown in Fig. 1 in the paper.
>
> We believe the state-dependent perturbation certification is reasonable and interesting given that different states are of different sensitivity against perturbations of the same magnitude. For example, in the Pong game, let us consider two states: state A where the ball is flying towards the paddle, and state B where the ball is flying away from the paddle. It is obvious that the action taken at state A is much more critical than the action taken at state B, and thus even a little perturbation applied to state A can cause much more devastating consequences than perturbations of a similar magnitude on state B. This is interesting since a byproduct of our certification is to help identify the “vulnerable states”. For instance, in Fig. 1 in the main paper and Fig. 7 in Appendix, we can see that in the Pong game, certain states are more vulnerable (i.e., lower certified radius) as we discussed in Appendix E.3.
>
> In addition, when the certified radius is close to 0, it means that this state is very sensitive and a small perturbation could potentially lead to the attacks, which inspires further exploration on such states to improve the certified robustness. Note that our certification is calculated regarding the worst case attacks under the perturbation constraints, so the certification is always reliable based on our theoretical analysis, although such “worst case” certification bound could be conservative. Thus, we believe there is still room to further tighten the bound to certify a larger perturbation radius for some states, which is an interesting future work.
>
> > 5. Missing references on robust RL
>
> We thank the reviewer for pointing out the interesting related works [12,13,14] and have updated our Appendix F.2 to include the references and discussions.

---

> ### Author Response · Authors · 2021-11-22
> **Official Response to Reviewer ZBtu (2/4)**
>
> > 3. The problem-dependent results obtained from CROP, and how reliable the certification is.
>
> Thanks for pointing out this interesting observation. We provide discussions about our certification results and implications for different RL algorithms on different games in Sec. 6. We believe both the domain and certification contribute to the final certification results, and thus in our evaluation we control the variable of domain (i.e., different games) and evaluate the certification results of different RL algorithms within each game only to mitigate the domain dependency, so as to obtain in-depth understanding about the RL algorithms based on the certification.
>
> Next, we will illustrate the reliability of our certification from different perspectives:
>
> i) Theoretically, the correctness/reliability of certification and verification naturally follows from rigorous mathematical proofs (Theorem 1, 2, 3, 4), which is also commonly followed by rich literature on robustness certification/verification in supervised learning [1,2,3,4,9] as discussed in Sec. 1.
>
> ii) Empirically, one standard approach for showing the validity/reliability of certification is to compare the robustness certification with the empirical robustness. We follow such standard to provide this comparison (Fig. 3) and discuss the tightness of our certificates (Sec. 6.3), where we show that our certification is sound (e.g., the empirical attacks cannot decrease the reward to be lower than the certification which is the lower bound) and tight. In addition, although we do not have previous work on the robustness certification in RL to compare with, we do find that our certified results largely match some existing empirical observations [8,10,11] (Sec. 6.1).
>
> We will provide some more empirical observations demonstrating the reliability and intuition of our certification below.
>
> a) We observe that some algorithms are more certifiably robust for certain games while not for others, and intuitively it is reasonable since some RL algorithms are more suitable for certain games given different properties of specific games (Sec. 6.3). Thus, our quantitative certification indeed provides rigorous backup for our intuitions about RL.
>
> b) We note that some of our conclusions based on certification such as “the adversarially trained policies are more robust than the ones based on vanilla training” are consistent with existing empirical evaluations, which again demonstrates the effectiveness of our certification, and such certification clearly provides strong guarantees compared with the empirical evaluations.
>
> c) Although as the first work that provides certification for RL algorithms, we do not have other certification algorithms to compare with in terms of their consistency, we do note that our three certification approaches under two certification criteria give highly consistent conclusions regarding the robustness of each RL algorithm on each game (Sec. 6.3). This again shows the consistency and reliability of our certification results.
>
> Finally, regarding the reviewer’s suggestion to evaluate some more environments, we provide the certification results on an additional game CartPole in Appendix E.6, where CARRL [5], NoisyNet [6], and RadialRL [8] are shown to be the most certifiably robust on CartPole, which is consistent with the empirical observation in CARRL [5]. We follow the suggestion to evaluate one more complicated game Breakout and provide the results and discussions in Appendix E.7 in our revision.

---

> ### Author Response · Authors · 2021-11-22
> **Official Response to Reviewer ZBtu (1/4)**
>
> We thank the reviewer for the valuable comments and suggestions. Actually, our ICLR submission is an updated version which has addressed several concerns from the reviews such as the typos, concrete descriptions for the RL methods (in Appendix D.2), etc. In addition, in our current version, we have added a benchmark for different certification methods and criteria on different games as stated in the Abstract with the link https://crop-leaderboard.me/.
>
> Next we provide detailed answers to the questions.
>
> > 1. Robustness improvement vs. Robustness certification.
>
> Thanks for pointing this out and we strongly believe that both robustness improvement and robustness certification are important and of great interest to the community. For instance, there is extensive research on certifying the robustness for classifiers [1,2,3,4], while this question is open for RL given its complexity and dynamics. In this work, we make the first attempt towards providing such certification for a given RL algorithm by proposing different certification metrics/algorithms/results to bridge the gap.
>
> We believe our proposed robustness certification for RL will lead to a line of interesting future research on how to certify RL with tighter bounds (just like what happened in the classification literatures currently), provide new insightful understandings about different RL algorithms given their robustness guarantees (our certification results of RL algorithms will serve as a benchmark), and inspire new RL algorithms design to improve the certified robustness.
>
> In addition, current methods that improve RL robustness usually do not come with any certification (e.g., the robust RL algorithms we certified in our paper), which means they cannot guarantee that the trained policy is robust against potential future or adaptive attackers, while such guarantee is of great importance for safety-critical applications such as autonomous driving. Thus, the scalable and rigorous robustness certification methods are in great need for various existing RL algorithms.
>
> Finally, we have added the suggested related research (Iyengar et al. [12], Nilim et al. [13], Tessler et al. [14]) in Appendix F.2 in our revision; thanks for pointing them out. We want to emphasize that our CROP framework is the first work that provides sound, practical, and scalable robustness certification for modern RL methods without further assumptions. Considering these related works, we provide additional experiments to certify three more robust RL algorithms (CARRL [5], NoisyNet [6], and GradDQN [7]) in addition to the six RL algorithms included in the main paper on CartPole for further analysis in Appendix E.6, and also conducted additional experiments to provide evaluations on a more complicated game Breakout in Appendix E.7 in our revision. We hope these new results are useful and are happy to run more evaluations if you have other suggestions. This discussion is very valuable for improving our work.
>
>
> > 2. Certification does not affect the model.
>
> Thanks for the comments and we agree that the model itself will not be affected by our certification process, which is by design. Since the goal of our paper is to provide scalable, rigorous, and fair robustness certification for different RL algorithms, we aim to not change the trained RL model itself and therefore provide comparable robustness guarantees for existing RL algorithms.

---

### Author Response · Authors · 2021-11-23
**Paper Revision Summary**

We thank all the reviewers for their insightful feedback and suggestions for improving the paper. We are glad that the reviewers found our paper clear and easy to follow, focuses on a valuable area of investigation, makes intuitive sense at high level and is also mathematically solid, and contains interesting and well-documented evaluation results.

Below we summarize the updates in our revision:

1. **[Abstract, Section 1, Section 6, Appendix F.3]** Change the statement of “*the first* framework for certifying robustness in RL” to “a framework that certifies the robustness for RL which *first* provides the certification for cumulative rewards”, and add discussions for the related works (Appendix F.3), following Reviewer $\color{blue}\text{zGtv}$’s suggestions
2. **[Section 4.2, Section 5, Appendix B]** Add more details of certification methods (e.g., detailed inference procedures, estimation of the algorithm parameters, algorithm time complexity, etc.) following Reviewer $\color{purple}\text{ZBtu}$, Reviewer $\color{blue}\text{zGtv}$, and Reviewer $\color{green}\text{tzpV}$’s suggestions.
3. **[Section 5.1]** Change the notation for the noise sequence from $\zeta$ to $\Delta$, following Reviewer $\color{blue}\text{zGtv}$’s suggestion.
4. **[Section 6, Appendix E.7 - Figure 9 and Table 1]** Add the evaluation results for CROP-LoRe on a more complicated game Breakout, following Reviewer $\color{green}\text{tzpV}$’s suggestion.
5. **[Section 6.2, Section D.3]** Explain the rationales for the evaluation settings and experimental designs (e.g., finite horizon setting and evaluation of practical attacks), following $\color{purple}\text{ZBtu}$ and Reviewer $\color{green}\text{tzpV}$’s suggestions.
6. **[Section 6.3, Appendix C]** Provide more discussions and analysis to help with the understanding of our algorithm (e.g., limitation of CROP-GRe, confidence of CROP-LoRe, extension to policy-based methods), following Reviewer $\color{blue}\text{zGtv}$ and Reviewer $\color{green}\text{tzpV}$’s suggestions.
7. **[Section 7, Appendix F]** Include more related work and add a separate section (Appendix F) to discuss our relation with relative work in detail, following Reviewer $\color{purple}\text{ZBtu}$, Reviewer $\color{blue}\text{zGtv}$, and Reviewer $\color{green}\text{tzpV}$’s suggestions.

We hope our answers and updated revision can address the reviewers’ concerns, and we are more than happy to discuss more with the reviewers on any unaddressed questions. Thanks!

---

### Decision · Program_Chairs · 2022-01-20

**Decision:**

Accept (Poster)

**Comment:**

The authors propose a framework for for the certification of reinforcement learning agents against adversarial observation/state perturbations based on randomized smoothing. They develop the theory of the framework, demonstrating that the framework can be used to certify lower bounds on the worst-case cumulative reward of an agent. They validate their theoretical bounds experimentally.

The paper is well written and reviewers were mostly in agreement that the contributions are worthy of acceptance. The technical concerns from reviewer zGtv were addressed during the discussion phase, but I strongly encourage the authors to revise the manuscript to address the points raised in the discussion.